# Trends in atmospheric evaporative demand in Great Britain using high-resolution meteorological data

**Emma L. Robinson[1], Eleanor M. Blyth[1], Douglas B. Clark[1], Jon Finch[1] and Alison C. Rudd[1]**

[1]{Centre for Ecology and Hydrology, Maclean Building, Benson Lane, Crowmarsh Gifford, Wallingford OX10 8BB}

Correspondence to: Emma L. Robinson (emrobi@ceh.ac.uk)

**Abstract**
Observations of climate are often available on very different spatial scales from observations
of the natural environments and resources that are affected by climate change. In order to help
bridge the gap between these scales using modelling, a new dataset of daily meteorological
variables was created at 1 km resolution over Great Britain for the years 1961-2012, by
interpolating coarser resolution climate data and including the effects of local topography.
These variables were used to calculate atmospheric evaporative demand (AED) at the same
spatial and temporal resolution. Two functions that represent AED were chosen: one is a
standard form of Potential Evapotranspiration (PET) and the other is a derived PET measure
used by hydrologists that includes the effect of water intercepted by the canopy (PETI).
Temporal trends in these functions were calculated, with PET found to be increasing in all
regions, and at an overall rate of $0.021\pm0.021$ mm d$^{-1}$ decade$^{-1}$ in Great Britain. PETI was found
to be increasing at a rate of $0.019\pm0.020$ mm d$^{-1}$ decade$^{-1}$ in Great Britain, but this was not
statistically significant. However, there was a trend in PETI in England of $0.023\pm0.023$ mm d$^{-1}$
$^{1}$ decade$^{-1}$. The trends were found to vary by season, with spring PET increasing by $0.043\pm0.019$
mm d$^{-1}$ decade$^{-1}$ ($0.038\pm0.018$ mm d$^{-1}$ decade$^{-1}$ when the interception correction is included) in
Great Britain, while there is no statistically significant trend in other seasons. The trends were
attributed analytically to trends in the climate variables; the overall positive trend was
predominantly driven by rising air temperature, although rising specific humidity had a negative
effect on the trend. Recasting the analysis in terms of relative humidity revealed that the overall
effect is that falling relative humidity causes the PET to rise. Increasing downward short- and
longwave radiation made an overall positive contribution to the PET trend, while decreasing
wind speed made a negative contribution to the trend in PET. The trend in spring PET was
particularly strong due to a strong decrease in relative humidity and increase in downward
shortwave radiation in the spring.

# 1 Introduction

There are many studies showing the ways in which our living environment is changing over time: changing global temperatures (IPCC, 2013), radiation (Wild, 2009) and wind speeds (McVicar et al., 2012) can have significant impacts on ecosystems and human life (IPCC, 2014a). While there are overall global trends, the impacts can vary between regions (IPCC, 2014b). In the UK, wildlife surveys of both flora (Wood et al., 2015; Evans et al., 2008) and fauna (Pocock et al., 2015) show a shift in patterns and timing (Thackeray et al., 2010). In addition, the UK natural resources of freshwater (Watts et al., 2015), soils (Reynolds et al., 2013; Bellamy et al., 2005) and vegetation (Berry et al., 2002; Hickling et al., 2006; Norton et al., 2012) are changing. The UK is experiencing new environmental stresses on the land and water systems through changes in temperature and river flows (Crooks and Kay, 2015; Watts et al., 2015; Hannaford, 2015), which are part of a widespread global pattern of temperature increase and circulation changes (Watts et al., 2015).

To explain these changes in terms of climate drivers, there are several gridded meteorological datasets available at global and regional scales. Global datasets can be based on observations – for example the 0.5° resolution Climate Research Unit time series 3.21 (CRU TS 3.21) data (Jones and Harris, 2013; Harris et al., 2014) – while some are based on global meteorological reanalyses bias-corrected to observations – for example the WATCH Forcing Data (WFD, 0.5°; Weedon et al. (2011)), the WATCH Forcing Data methodology applied to ERA-Interim reanalysis product (WFDEI, 0.5°; Weedon et al. (2014)) and the Princeton Global Meteorological Forcing Dataset (0.25°–1°; Sheffield et al. (2006)). At the regional scale in Great Britain (GB), there are datasets that are derived directly from observations – for example the Met Office Rainfall and Evaporation Calculation System (MORECS) dataset at 40 km resolution (Thompson et al., 1981; Hough and Jones, 1997; Field, 1983) and the UKCP09 observed climate data at 5 km resolution (Jenkins et al., 2008).

However, while regional observations of carbon, methane and water emissions from the land (Baldocchi et al., 1996), the vegetation cover (Morton et al., 2011) and soil properties (FAO/IIASA/ISRIC/ISS-CAS/JRC, 2012) are typically made at the finer landscape scale of 100 m to 1000 m, most of these long-term gridded meteorological datasets are only available at a relatively coarse resolution of a few tens of km. These spatial scales may not be representative of the climate experienced by the flora and fauna being studied, and it has also been shown that input resolution can have a strong effect on the performance of hydrological

models (Kay et al., 2015). In addition, the coarse temporal resolution of some datasets, for
example the monthly CRU TS 3.21 data (Harris et al., 2014; Jones and Harris, 2013), can miss
important sub-monthly extremes.
Regional studies are important to identify drivers and impacts of changing meteorology that
may or may not be reflected in trends in global means. For example, in Canada (Vincent et al.,
2015) and Europe (Fleig et al., 2015), high resolution meteorological data have been used to
identify the impacts of changing circulation patterns, while in Australia wind speed data have
been used to quantify the effects of global stilling in the region (McVicar et al., 2008). While
there are datasets available at finer spatial and temporal resolutions for the UK (such as
UKCP09 (Jenkins et al., 2008)), these often do not provide all the variables needed to identify
the impacts of changing climate.
To address this, we have created a meteorological dataset for Great Britain at 1 km resolution:
the Climate Hydrology and Ecology research Support System meteorology dataset for Great
Britain (1961-2012) (CHESS-met; Robinson et al. (2015b)). It is derived from the observation-
based MORECS dataset (Thompson et al., 1981; Hough and Jones, 1997), and then downscaled
using information about topography. This is augmented by an independent precipitation dataset
– Gridded Estimates of daily and monthly Areal Rainfall for the United Kingdom (CEH-GEAR;
Tanguy et al. (2014); Keller et al. (2015)) – along with variables from two global datasets –
WFD and CRU TS 3.21 – to produce a comprehensive, observation-based, daily meteorological
dataset at 1 km × 1 km spatial resolution.
In order to understand the effect of meteorology on the water cycle, a key variable in
hydrological modelling is the atmospheric evaporative demand (AED), which is determined by
meteorological variables (Kay et al., 2013). It has been shown that water-resource and
hydrological model results are largely driven by how this property is defined and used
(Haddeland et al., 2011). The AED can be expressed in several ways, for instance the
evaporation from a wet surface, from a well-watered but dry uniform vegetated cover, or from
a hypothetical well-watered but dry version of the actual vegetation. Metrics such as the Palmer
Drought Severity Index (PDSI; Palmer (1965)) use potential evapotranspiration (PET) as an
input to represent AED, while many hydrological models such as Climate and Land use
Scenario Simulation in Catchments (CLASSIC; Crooks and Naden (2007)) or Grid-to-Grid
(G2G; Bell et al. (2009)), which also require an input representing AED, use a distinct form of
the PET which includes the intercepted water from rainfall (this is described later in the text)

which we hereby name PETI. While hydrological models can make use of high resolution topographic information and precipitation datasets, they are often driven with PET calculated at a coarser resolution (Bell et al., 2011; Bell et al., 2012; Kay et al., 2015). Therefore, we have also created a 1 km × 1 km resolution dataset, the Climate Hydrology and Ecology research Support System Potential Evapotranspiration dataset for Great Britain (1961-2012) (CHESS-PE; Robinson et al. (2015a)), consisting of estimates of PET and PETI, which can be used to run high-resolution hydrological models.

Other regional studies have created gridded estimates of AED in Austria (Haslinger and Bartsch, 2016) and Australia (Donohue et al., 2010). Regional studies of trends in AED have seen varied results, with increasing AED seen in Romania (Paltineanu et al., 2012), Serbia (Gocic and Trajkovic, 2013), Spain (Vicente-Serrano et al., 2014), some regions of China (Li and Zhou, 2014) and Iran (Azizzadeh and Javan, 2015; Hosseinzadeh Talaee et al., 2013; Tabari et al., 2012), decreasing AED in north east India (Jhajharia et al., 2012) and regions in China (Yin et al., 2009; Song, 2010; Shan et al., 2015; Zhao et al., 2015; Zhang et al., 2015; Lu et al., 2016) and regional variability in Australia (Donohue et al., 2010) and China (Li et al., 2015). In order to understand this variability, it is important to quantify the relative contributions of the changing meteorological variables to trends in AED and regional studies often find different drivers of changing AED (see McVicar et al. (2012) for a review). Relative humidity has been shown to drive AED in the Canary Islands (Vicente-Serrano et al., 2016),wind speed and air temperature were shown to have nearly equal but opposite effects in Australia (Donohue et al., 2010), while in China sunshine hours (Li et al., 2015), wind speed (Yin et al., 2009) or a combination of the two (Lu et al., 2016) have been shown to drive trends. Rudd and Kay (2015) investigated projected changes in PET using a regional climate model, but little has been done to investigate historical trends of AED in the UK.

The objectives of this paper are (i) to evaluate the trends in key meteorological variables in Great Britain over the years 1961-2012; (ii) to evaluate the AED in Great Britain over the same time period using PET; (iii) to investigate the effect of including interception in the formulation of PET called PETI; (iv) to evaluate trends in PET over the time period of interest; and (v) to attribute the trends in PET to trends in meteorological variables. To address these objectives, the paper is structured as follows. Section 2 presents the calculation of the meteorological variables. Section 3 presents the calculation of PET and PETI from the meteorological variables and assesses the difference between PET and PETI. In Section 4 the trends of the meteorological

variables and AED are calculated and the trends in PET are attributed to trends in meteorological variables. In Section 5 the results are discussed and conclusions are presented in Section 6.

## 2   Calculation of meteorological variables

The meteorological variables included in this new dataset (Robinson et al., 2015b) are daily mean values of air temperature, specific humidity, wind speed, downward longwave (LW) and shortwave (SW) radiation, precipitation and air pressure, plus daily temperature range (Table 1). These variables are important drivers of near-surface conditions, and, for instance, are the full set of variables required to drive the JULES land surface model (LSM) (Best et al., 2011; Clark et al., 2011), as well as other LSMs.

The data were derived primarily from MORECS, which is a long-term gridded dataset starting in 1961 and updated to the present (Thompson et al., 1981; Hough and Jones, 1997). It interpolates five variables from synoptic stations (daily mean values of air temperature, vapour pressure and wind speed, daily hours of bright sunshine and daily total precipitation) to a 40 km × 40 km resolution grid aligned with the Ordnance Survey National Grid. There are currently 270 stations reporting in real time, while a further 170 report the daily readings on a monthly basis, but numbers have varied throughout the run. The algorithm interpolates a varying number of stations (up to nine) for each square, depending on data availability (Hough and Jones, 1997). The interpolation is such that the value in each grid square is the effective measurement of a station positioned at the centre of the square and at the grid square mean elevation, averaged from 00:00 GMT to 00:00 GMT the next day. MORECS is a consistent, quality-controlled time series, which accounts for changing station coverage. The MORECS variables were used to derive the air temperature, specific humidity, wind speed, downward LW and SW radiation and air pressure in the new dataset. The WFD and CRU TS 3.21 datasets were used for surface air pressure and daily temperature range respectively, as they could not be calculated solely from MORECS. Additionally precipitation was obtained from the CEH-GEAR data, which is a product directly interpolated to 1 km from the station data (Keller et al., 2015).

The spatial coverage of the dataset was determined by the spatial coverage of MORECS, which covers the majority of Great Britain, but excludes some coastal regions and islands at the 1 km scale. For most of these points, the interpolation was extended from the nearest MORECS

squares, but some outlying islands (in particular Shetland and the Scilly Isles) were excluded
when the entire island was further than 40 km from the nearest MORECS square.

## 2.1  Air temperature

Air temperature, $T_a$ (K), was derived from the MORECS air temperature. The MORECS air
temperature was reduced to mean sea level, using a lapse rate of -0.006 K m$^{-1}$ (Hough and
Jones, 1997). A bicubic spline was used to interpolate from 40 km resolution to 1 km resolution,
then the temperatures were adjusted to the elevation of each 1 km square using the same lapse
rate. The 1 km resolution elevation data used were aggregated from the Integrated Hydrological
Digital Terrain Model (IHDTM) – a 50 m resolution digital terrain model (Morris and Flavin,
172  1990).

## 2.2  Specific humidity

Specific humidity, $q_a$ (kg kg$^{-1}$), was derived from the MORECS vapour pressure, $e_M$ (Pa), which
was first reduced to mean sea level, using the equation
$$e_{sea} = e_M \left( 1 - \frac{L_e}{100} h_M \right) \tag{1}$$
where $L_e$ is the lapse rate of -0.025 % m$^{-1}$ and $h$ is the elevation of the MORECS square
(Thompson et al., 1981). The actual lapse rate of humidity will, in general, vary according to
atmospheric conditions. However, calculating this would require more detailed information
than is available in the input data used. Any method of calculating the variation of specific
humidity with height will involve several assumptions, but the method used here is well-
established and is used by the Met Office in calculating MORECS (Thompson et al., 1981).
The value of the vapour pressure lapse rate is chosen to keep relative humidity approximately
constant with altitude, rather than assuming that the vapour pressure itself is constant.
A bicubic spline was used to interpolate vapour pressure to 1 km resolution then the values
were adjusted to the 1 km resolution elevation using the IHDTM elevations and using the same
lapse rate, such that
$$e = e_{sea,1km} \left( 1 + \frac{L_e}{100} h_{1km} \right), \tag{2}$$
where $e_{sea,1km}$ is the sea-level vapour pressure at 1 km resolution and $h_{1km}$ is the 1 km resolution
elevation.
Finally the specific humidity was calculated, using
$$q_a = \frac{\epsilon e}{p_* - (1-\epsilon)e},$$    (3)
where $e$ is the vapour pressure (Pa) and $\epsilon = 0.622$ is the mass ratio of water to dry air (Gill,
1982). The air pressure, $p_*$, in this calculation was assumed to have a constant value of 100000
Pa because this was prescribed in the computer code. It would be better to use a varying air
pressure, as calculated in Section 2.8, but this makes a negligible difference (of a few percent)
to the calculated specific humidity, and to the PET and PETI calculated in Section 3, and a
constant $p_*$ was retained.
**2.3  Downward shortwave radiation**
Downward SW radiation, $S_d$ (W m$^{-2}$), was derived from the MORECS hours of bright sunshine
(defined as the total number of hours in a day for which solar irradiation exceeds 120 W m$^{-2}$
(WMO, 2013)). The value calculated is the mean SW radiation over 24 hours. The sunshine
hours were used to calculate the cloud cover factor, $C_f = n/N$, where $n$ is the number of hours
of bright sunshine in a day, and $N$ is the total number of hours between sunrise and sunset
(Marthews et al., 2011). The cloud cover factor was interpolated to 1 km resolution using a
bicubic spline. The downward SW solar radiation for a horizontal plane at the Earth's surface
was then calculated using the solar angle equations of Iqbal (1983) and a form of the Ångström-
Prescott equation which relates hours of bright sunshine to solar irradiance (Ångström, 1918;
Prescott, 1940), with empirical coefficients calculated by Cowley (1978). They vary spatially
and seasonally and effectively account for reduction of irradiance with increasing solar zenith
angle, as well as implicitly accounting for spatially- and seasonally-varying aerosol effects.
However, they do not vary interannually and thus do not explicitly include long-term trends in
aerosol concentration.
The downward SW radiation was then corrected for the average inclination and aspect of the
surface, assuming that only the direct beam radiation is a function of the inclination and that
the diffuse radiation is homogeneous. It was also assumed that the cloud cover is the dominant
factor in determining the diffuse fraction (Muneer and Munawwar, 2006). The aspect and
inclination were calculated using the IHDTM elevation at 50 m resolution, following the
method of Horn (1981), and were then aggregated to 1 km resolution. The top of atmosphere
flux for horizontal and inclined surfaces was calculated following Allen et al. (2006) and the
ratio used to scale the direct beam radiation.

## 2.4 Downward longwave radiation

Downward LW radiation, $L_d$ (W m$^{-2}$), was derived from the 1 km resolution air temperature (Sect. 2.1), vapour pressure (Sect. 2.2) and cloud cover factor (Sect. 2.3). The downward LW radiation for clear sky conditions was calculated as a function of air temperature and precipitable water using the method of Dilley and O'Brien (1998), with precipitable water calculated from air temperature and humidity following Prata (1996). The additional component due to cloud cover was calculated using the equations of Kimball et al. (1982), assuming a constant cloud base height of 1000 m.

## 2.5 Wind speed

The wind speed at a height of 10 m, $u_{10}$ (m s$^{-1}$), was derived from the MORECS 10 m wind speed, which were interpolated to 1 km resolution using a bicubic spline and adjusted for topography using a 1 km resolution dataset of mean wind speeds produced by the UK Energy Technology Support Unit (ETSU; Newton and Burch (1985); Burch and Ravenscroft (1992)). This used Numerical Objective Analysis Boundary Layer (NOABL) methodology combined with station wind measurements over the period 1975-84 to produce a map of mean wind speed over the UK. To calculate the topographic correction, the ETSU wind speed was aggregated to 40 km resolution, then the difference between each 1 km value and the corresponding 40 km mean found. This difference was added to the interpolated daily wind speed. In cases where this would result in a negative wind speed, the wind speed was set to zero.

## 2.6 Precipitation

Precipitation rate, $P$ (kg m$^{-2}$ s$^{-1}$), is taken from the daily CEH-GEAR dataset (Tanguy et al., 2014; Keller et al., 2015), scaled to the appropriate units. The CEH-GEAR methodology uses natural neighbour interpolation (Gold, 1989) to interpolate synoptic station data to a 1 km resolution gridded daily dataset of the estimated precipitation in 24 hours between 09:00 GMT and 09:00 GMT the next day.

## 2.7 Daily temperature range

Daily temperature range (DTR), $D_T$ (K), was obtained from the CRU TS 3.21 monthly mean daily temperature range estimates on a 0.5° latitude × 0.5° longitude grid, which is interpolated from monthly climate observations (Harris et al., 2014; Jones and Harris, 2013). There is no

standard way to correct DTR for elevation, so these data were reprojected to the 1 km grid with
no interpolation and the monthly mean used to populate the daily values in each month.
Although DTR is not required in the calculation of AED, it is a required input of the JULES
LSM, in order to run at sub-daily timestep with daily input data.

## 2.8 Surface air pressure

Surface air pressure, $p_*$ (Pa), was derived from the WFD, an observation-corrected reanalysis
product, which provides 3 hourly meteorological data for 1958-2001 on a 0.5° latitude × 0.5°
longitude resolution grid (Weedon et al., 2011). Mean monthly values of WFD surface air
pressure and air temperature were calculated for each 0.5° grid box over the years 1961-2001.
These were reprojected to the 1 km grid with no interpolation, then the lapse rate of air
temperature (Sect. 2.1) used to calculate the integral of the hypsometric equation (Shuttleworth,
2012), in order to obtain the air pressure at the elevation of each 1 km grid. The mean monthly
values were used to populate the daily values in the full dataset, thus the surface air pressure in
the new dataset does not vary interanually, but does vary seasonally. This is reasonable as the
trend in surface air pressure in the WFD is negligible (Weedon et al., 2011).

## 2.9 Spatial and seasonal patterns of meteorological variables

Long-term mean values of the meteorological variables were calculated for each 1 km square
over the whole dataset, covering the years 1961-2012 (Fig. 1). Four sub-regions of interest were
defined (Fig. 2); three of these regions correspond to nations (England, Wales and Scotland),
while the fourth is the 'English lowlands', a subset of England, covering south-central and
south-east England, East Anglia and the East Midlands (Folland et al., 2015). Mean-monthly
climatologies were calculated over the whole of Great Britain (GB), and over these four regions
of interest (Fig. 3).
The maps clearly show the effect of topography on the variables (Fig. 1), with an inverse
correlation between elevation and temperature, specific humidity, downward LW radiation and
surface air pressure and a positive correlation with wind speed. The precipitation has an east-
west gradient due to prevailing weather systems and orography. The fine-scale structure of the
downward SW radiation is due to the aspect and elevation of each grid cell, with more spatial
variability in areas with more varying terrain. As no topographic correction has been applied to
DTR, it varies only on a larger spatial scale. Although specific humidity is inversely
proportional to elevation, relative humidity is not, as the saturated specific humidity will also
be inversely proportional to elevation due to the decrease in temperature with height. The strong
correlation between wind speed and elevation means that it is very variable over short spatial
scales, particularly in Scotland.
The mean-monthly climatologies (Fig. 3) demonstrate the differences between the regions, with
Scotland generally having lower temperatures and more precipitation than the average, and
England (particularly the English lowlands) being warmer and drier.
**2.10 Validation of meteorology**
The precipitation dataset, CEH-GEAR, has previously been validated against observations
(Keller et al., 2015). Other studies discuss the uncertainties in the CRU TS 3.21 daily
temperature range data (Harris et al., 2014) and WFDEI air pressure data (Weedon et al., 2014).
For the other variables, the MORECS data set is ultimately derived from the synoptic stations
around the UK which represent most of the available observed meteorological data for the
country. The only way to validate the gridded meteorology presented here is to compare it to
independently observed data, which are available at a few sites where meteorological
measurement stations that are not part of the synoptic network are located. Here we carry out a
validation exercise with data from four sites from the UK, which have meteorological
measurements available for between 5 and 10 years. Details of the sites and data are in
Appendix A. Fig. 4 shows the comparison of data set air temperature with the observed air
temperature at each of the four sites. This shows a strong correlation ($r^2$ between 0.94 and 0.97)
between the data set and the observations. Fig. 5 shows the mean-monthly climatology
calculated from both the data set and from the observations (only for times for which
observations were available) and demonstrates that the data set successfully captures the
seasonal cycle. This has been repeated for downward SW radiation and for an estimate of the
mixing ratio of water vapour, 10 m wind speed and surface air pressure (Appendix A). The air
temperature, downward SW radiation and mixing ratio all have high correlations and represent
the seasonal cycle well. The downward SW is overestimated at Auchencorth Moss, which may
be due to local factors (e.g. shading, or the siting of the station within the grid square). The
wind speed is overestimated by the derived data set at two sites, which is likely to be due to
land cover effects. The modelling which produced the ETSU dataset uses topography but not
land cover (Burch and Ravenscroft, 1992; Newton and Burch, 1985), so at sites with tall
vegetation the wind speed is likely to be less than the modelled value. The air pressure has a
low correlation because the data set contains a mean-monthly climatological value. However,
the mean bias is low and the RMSE is small, confirming that it is reasonable to use a
climatological value in place of daily data.

## 3   Calculation of potential evapotranspiration (PET)

There are several ways to assess the evaporative demand of the atmosphere. Pan evaporation
can be modelled using the Pen-Pan model (Rotstayn et al., 2006), or open-water evaporation
can be modelled with the Penman equation (Penman, 1948). However, neither of these account
for the fact that in general the evaporation is occurring from a vegetated surface. A widely used
model is the Penman-Monteith PET, $E_P$ (mm d$^{-1}$, equivalent to kg m$^{-2}$ d$^{-1}$), which is a physically-
based formulation of AED (Monteith, 1965), including the effect of stomatal resistance. It
provides an estimate of AED dependent on the atmospheric conditions but allowing for the fact
that the water is evaporating through the surface of leaves and thus the resistance is higher. It
can be calculated from the daily meteorological variables using the equation
$$E_P = \frac{t_d}{\lambda}\frac{\Delta A + \frac{c_p \rho_a}{r_a}(q_s - q_a)}{\Delta + \gamma\left(1 + \frac{r_s}{r_a}\right)} \,,\tag{4}$$
where $t_d = 86400$ s d$^{-1}$ is the length of a day, $\lambda = 2.5\times10^6$ J kg$^{-1}$ is the latent heat of evaporation,
$q_s$ is saturated specific humidity (kg kg$^{-1}$), $\Delta$ is the gradient of saturated specific humidity with
respect to temperature (kg kg$^{-1}$ K$^{-1}$), $A$ is the available energy (W m$^{-2}$), $c_p = 1010$ J kg$^{-1}$ K$^{-1}$ is the
specific heat capacity of air, $\rho_a$ is the density of air (kg m$^{-3}$), $q_a$ is specific humidity (kg kg$^{-1}$),
$\gamma = 0.004$ K$^{-1}$ is the psychrometric constant, $r_s$ is stomatal resistance (s m$^{-1}$) and $r_a$ is aerodynamic
resistance (s m$^{-1}$) (Stewart, 1989).
The saturated specific humidity, $q_s$ (kg kg$^{-1}$), is calculated from saturated vapour pressure, $e_s$
(Pa), using Eq. 3. The saturated vapour pressure is calculated using an empirical fit to air
temperature
$$e_s = p_{sp}exp\left(\sum_{i=1}^{4} a_i\left(1 - \frac{T_{sp}}{T_a}\right)^i\right),\tag{5}$$
where $p_{sp} = 101325$ Pa is the steam point pressure, $T_{sp} = 373.15$ K is the steam point temperature
and $a = (13.3185, -1.9760, -0.6445, -0.1299)$ are empirical coefficients (Richards, 1971).
The derivative of the saturated specific humidity with respect to temperature, $\Delta$ (kg kg$^{-1}$ K$^{-1}$),
is therefore
$$\Delta = \frac{T_{sp}}{T_a^2} \frac{p_* q_s}{p_* - (1-\epsilon)e_s} \sum_{i=1}^{4} i a_i \left(1 - \frac{T_{sp}}{T_a}\right)^{i-1} , \qquad (6)$$
where the air pressure used is the spatially varying air pressure calculated in Sect.2.8.
The available energy, $A$ (W m$^{-2}$), is the energy balance of the surface,
$$A = R_n - G , \qquad (7)$$
where $R_n$ is the net radiation (W m$^{-2}$) and $G$ is the soil heat flux (W m$^{-2}$). The net soil heat flux
is negligible at the daily timescale (Allen et al., 1998), so the available energy is equal to the
net radiation, such that
$$A = (1 - \alpha)S_d + \varepsilon(L_d - \sigma T_*^4) , \qquad (8)$$
where $\sigma$ is the Stefan-Boltzmann constant, $\alpha$ is the albedo and $\varepsilon$ the emissivity of the surface
and $T_*$ is the surface temperature (Shuttleworth, 2012). For this study we make the simplifying
assumption that the surface temperature is approximately equal to the air temperature, $T_a$ and
use the latter in Eq. 8.
The air density, $\rho_a$ (kg m$^{-3}$), is a function of air pressure and temperature,
$$\rho_a = \frac{p_*}{r T_a} , \qquad (9)$$
where $r = 287.05$ J kg$^{-1}$ K$^{-1}$ is the gas constant of air and the air pressure used is the spatially
varying air pressure calculated in Sect. 2.8.
The stomatal and aerodynamic resistances are strongly dependent on land cover due to
differences in roughness length and physiological constraints on transpiration of different
vegetation types. In addition, the albedo and emissivity are also dependent on the land cover.
In order to investigate the effect of meteorology on AED, as distinct from land use effects, the
PET was calculated for a single land cover type over the whole of the domain. If necessary, this
can be adjusted to give an estimate of PET specific to the local land cover, for example using
regression relationships (Crooks and Naden, 2007). As a standard, the Food and Agriculture
Organization of the United Nations (FAO) calculate reference crop evaporation for a
hypothetical reference crop, which corresponds to a well-watered grass (Allen et al., 1998).
Following this, the PET in the current study was calculated for a reference crop of 0.12 m
height, with constant stomatal resistance, $r_s = 70.0$ s m$^{-1}$, an albedo of 0.23 and emissivity of
0.92 over the whole of Great Britain. This study therefore neglects the effect of land-use on
evaporation, which could be investigated in future by calculating PET for different land surface
types, with different coverage for each year of the dataset.
In general, aerodynamic resistance is a function of wind speed and canopy height. Following
Allen et al. (1998), the aerodynamic resistance, $r_a$ (s m$^{-1}$), of a reference crop of 0.12 m height
is a function of the 10 m wind speed
$$r_a = \frac{278}{u_{10}}.$$      (10)
Note that, since the wind speed is likely to be biased high at sites with tall vegetation (Sect.
2.10), this implies that the aerodynamic resistance is likely to be biased low, leading to an
overestimate of PET. However, the estimate of PET here is for a reference crop over the whole
of the dataset, and does not consider the effect of tall vegetation, so the wind speed is
appropriate.
Thus the PET is a function of six of the meteorological variables: air temperature, specific
humidity, downward LW and SW radiation, wind speed and surface air pressure.
To explore the role of the different meteorological variables in the AED, it is helpful to split
the radiative component (the first part of the numerator in Eq. 4) from the wind component (the
second part). Formally, this is defined as follows (Doorenbos, 1977):
The radiative component, $E_{PR}$,
$$E_{PR} = \frac{t_d}{\lambda} \frac{\Delta A}{\Delta + \gamma\left(1 + \frac{r_s}{r_a}\right)},$$      (11)
and the aerodynamic component, $E_{PA}$,
$$E_{PA} = \frac{t_d}{\lambda} \frac{\frac{c_p \rho_a}{r_a}(q_s - q_a)}{\Delta + \gamma\left(1 + \frac{r_s}{r_a}\right)},$$      (12)
such that $E_P = E_{PR} + E_{PA}$.
## 3.1   Potential evapotranspiration with interception (PETI)
When rain falls, water is intercepted by the canopy. The evaporation of this water is not
constrained by stomatal resistance but is subject to the same aerodynamic resistance as
transpiration (Shuttleworth, 2012). At the same time, transpiration is inhibited in a wet canopy.
Suppression of transpiration is well observed both by comparing eddy-covariance fluxes and
observations of sap flow (Kume et al., 2006; Moors, 2012), and by observing stomatal and
photosynthesis response to wetting (Ishibashi and Terashima, 1995). For plants which have at
least some of their stomata on the upper surface of the leaves, this can be due to water directly
blocking the stomata. However, in GB most plants have stomata only on the underside of the
leaves, so the transpiration is inhibited by other mechanisms.
Physically, the suppression may be due to the fact that energy is used in evaporating the
intercepted water, so less is available for transpiration or that the increased humidity of the air
decreases the evaporative demand (Bosveld and Bouten, 2003). It may also be due to the
presence of water on the leaf surface causing stomatal closure through physiological reactions,
which can be observed even when the stomata are on the underside of a leaf and the water is
lying on the upper side (Ishibashi and Terashima, 1995).
In the short term after a rain event, potential water losses due to evaporation may be
underestimated if only potential transpiration is calculated, and therefore overall rates
underestimated. As transpiration is inhibited over the wet fraction of the canopy (Ward and
Robinson, 2000), the PET over a grid box will be a linear combination of the potential
interception and potential transpiration, each weighted by the fraction of the canopy that is wet
or dry. This can be accounted for by introducing an interception term to the calculation of PET,
giving PETI. This is modelled as an interception store, which is (partially) filled by rainfall,
proportionally inhibiting the transpiration. As the interception store dries, the relative
contribution of interception is decreased and the transpiration increases. In this dataset, this
correction is applied on days with precipitation, while on days without precipitation the
potential is equal to the PET defined in Eq. 4. Although an unconventional definition of PET,
a similar interception correction is applied to the PET provided at 40 km resolution by
MORECS (Thompson et al., 1981) which is used widely by hydrologists.
This method implicitly assumes that the water is liquid, however snow lying on the canopy will
also inhibit transpiration, and will be depleted by melting as well as by sublimation. The rates
may be slower, and the snow may stay on the canopy for longer than one day. However, the
difference of accounting for canopy snow as distinct from canopy water will have a small effect
on large-scale averages, as the number of days with snow cover in GB is relatively low, and
they occur during winter when the PET is small.
The PETI is a weighted sum of the PET, $E_P$, (as calculated in Eq 2.) and potential interception,
$E_I$, which is calculated by substituting zero stomatal resistance, $r_s$=0 s m$^{-1}$, into Eq. 4. To
calculate the relative proportions of interception and transpiration, it is assumed that the wet
fraction of the canopy is proportional to the amount of water in the interception store. The
interception store, $S_I$ (kg m$^{-2}$), decreases through the day according to an exponential dry down
(Rutter et al., 1971), such that
$$S_I(t) = S_o e^{-\frac{E_I}{S_{tot}}t},$$ (13)
where $E_I$ is the potential interception, $S_{tot}$ is the total capacity of the interception store (kg m$^{-2}$),
$S_0$ is the precipitation that is intercepted by the canopy (kg m$^{-2}$) and $t$ is the time (in days) since
a rain event. We assume that the interception component is only significant on the day in which
rainfall occurs, and that it is negligible on subsequent days, so the calculation is only carried
out for days of non-zero rainfall. Thus $t$ is a positive fraction between zero and one.
The total capacity of the interception store is calculated following Best et al. (2011), such that
$$S_{tot} = 0.5 + 0.05\Lambda,$$ (14)
where $\Lambda$ is the leaf area index (LAI). For the FAO standard grass land cover the LAI is 2.88
(Allen et al., 1998). The fraction of precipitation intercepted by the canopy is also found
following Best et al. (2011), assuming that precipitation lasts for an average of 3 hours.
The wet fraction of the canopy, $C_{wet}$, is proportional to the store size, such that
$$C_{wet}(t) = \frac{S(t)}{S_{tot}}.$$ (15)
The total PETI is the sum of the interception from the wet canopy and the transpiration from
the dry canopy,
$$E_{PI}(t) = E_I C_{wet}(t) + E_P\left(1 - C_{wet}(t)\right).$$ (16)
This is integrated over one day (from $t=0$ to $t=1$) to find the total PETI, $E_{PI}$ (mm d$^{-1}$), to be
$$E_{PI} = S_0\left(1 - e^{-\frac{E_I}{S_{tot}}}\right) + E_P\left(1 - \frac{S_0}{E_I}\left(1 - e^{-\frac{E_I}{S_{tot}}}\right)\right).$$ (17)
This calculation is only carried out for days on which rainfall occurs. On subsequent days it is
assumed that the canopy has sufficiently dried out that the interception component is zero.
The PETI is a function of the same six meteorological variables as the PET, plus the
precipitation.

## 3.2 Spatial and seasonal patterns of PET and PETI

Both PET and PETI have a distinct gradient from low in the north-west to high in the south-east, and they are both inversely proportional to the elevation (Fig. 6), reflecting the spatial patterns of the meteorological variables. The PETI is 8 % higher than the PET overall but this difference is larger in the north and west, where precipitation rates, and therefore interception, are higher (Fig. 6). In Scotland, the higher interception and lower AED mean that this increase is a larger proportion of the total, with the mean PETI being 11 % larger than the PET (in some areas the difference is more than 25%). In the English lowlands the difference is smaller, at 6 %, but this is a more water limited region where hydrological modelling can be sensitive to even relatively small adjustments to PET (Kay et al., 2013).

The seasonal climatology of both PET and PETI follow the meteorology (Fig. 7), with high values in the summer and low in the winter. Although the relative difference peaks in winter, the absolute difference between PET and PETI is bimodal, with a peak in March and a smaller peak in October (September in Scotland) (Fig. 7), because in winter the overall AED is low, while in summer the amount of precipitation is low, so the interception correction is small. The seasonal cycle of PET is driven predominantly by the radiative component, which has a much stronger seasonality than the aerodynamic component (Fig. 8).

On a monthly or annual timescale, the ratio of PET to precipitation is an indicator of the wet- or dryness of a region (Oldekop, 1911; Andréassian et al., 2016). Low values of PET relative to precipitation indicate wet regions, where evaporation is demand-limited, while high values indicate dry, water-limited regions. In the wetter regions (Scotland, Wales) mean-monthly PET and PETI (Fig. 7) are on average lower than the mean-monthly precipitation (Fig. 3) throughout the year, while in drier regions (England, English lowlands) the mean PET and PETI are higher than the precipitation for much of the summer, highlighting the regions' susceptibility to hydrological drought (Folland et al., 2015).

## 4 Decadal trends

### 4.1 Meteorological Variables

Annual means of the meteorological variables (Fig. 9) and the PET and PETI (Fig. 10) were calculated for each region. The trends in these annual means were calculated using linear regression; the significance ($P$ value) and 95% confidence intervals (CI) of the slope are calculated specifically allowing for the non-zero lag-1 autocorrelation, to account for possible

correlations between adjacent data points (Zwiers and von Storch, 1995; von Storch and Zwiers,
1999). The annual trends can be seen in Table 2. In addition, seasonal means were calculated,
with the four seasons defined to be Winter (December-February), Spring (March-May),
Summer (June-August) and Autumn (September-November), and trends in these means were
also found.
The trends in the annual and seasonal means for all regions are plotted in Fig. 11; trends that
are statistically significant at the 5% level are plotted with solid error bars, those that are not
significant are plotted with dashed lines. The analysis was repeated for each pixel in the 1 km
resolution dataset; maps of these rates of change can be seen in Fig. B1.
There was a statistically significant trend in air temperature in the English Lowlands throughout
the year. In the other regions the trends were statistically significant in spring and autumn, and
for the annual means. The trends agree with recent trends in the Hadley Centre Central England
Temperature (HadCET) dataset (Parker and Horton, 2005) and in temperature records for
Scotland (Jenkins et al., 2008) as well as in the CRUTEM4 dataset (Jones et al., 2012). An
increase in winter precipitation in Scotland is seen in the current dataset, which leads to a
statistically significant increase in the annual mean precipitation of GB. However, all other
regions and seasons have no statistically significant trends in precipitation. Long term
observations show that there has been little trend in annual precipitation, but a change in
seasonality with wetting winters and drying summers since records began, although with little
change over the past 50 years (Jenkins et al., 2008). The statistically significant decline in wind
speed in all regions is consistent with the results of McVicar et al. (2012) and Vautard et al.
(2010), who report decreasing wind speeds in the northern hemisphere over the late 20[th]
century.

## 4.2  Potential Evapotranspiration

The trends of the meteorological variables are interesting in their own right. But for hydrology,
it is the impact that the trends have on evaporation that matters and that depends on their
combination, which can be expressed through PET.
The regional trends of annual mean PET and PETI and the radiative and aerodynamic
components of PET can be seen in Table 2, and the trends in the annual and seasonal means are
plotted in Fig. 12 for all regions. Maps of the trends can be seen in Fig. B2. The trend in the
radiative component of PET is positive over the whole of GB. However, the trend in the
aerodynamic component varies; for much of Wales, Scotland and northern England, it is not
significant, or is slightly negative, while in south-east England and north-west Scotland it is
positive. This leads to a positive trend in PET over much of GB, but no significant trend in
southern Scotland and northern England. There is a statistically significant increase in annual
PET in all regions except Wales; the GB trend ($0.021\pm0.021$ mm d$^{-1}$ decade$^{-1}$) is equivalent to
an increase of $0.11\pm0.11$ mm d$^{-1}$ ($8.3\pm8.1$ % of the long term mean) over the whole dataset.
Increases in PETI are only statistically significant in England ($0.023\pm0.023$ mm d$^{-1}$ decade$^{-1}$)
and English lowlands ($0.028\pm0.025$ mm d$^{-1}$ decade$^{-1}$), where the increases over the whole
dataset are $0.12\pm0.12$ mm d$^{-1}$ ($8.0\pm8.0$ % of the long term mean) and $0.15\pm0.13$ mm d$^{-1}$ ($9.7\pm8.8$
% of the long term mean) respectively. There is a difference in trend between different seasons.
In winter, summer and autumn there are no statistically significant trends in PET or PETI, other
than the English lowlands in autumn, but the spring is markedly different, with very significant
trends ($P<0.0005$) in all regions. The GB spring trends in PET ($0.043\pm0.019$ mm d$^{-1}$ decade$^{-1}$)
and PETI ($0.038\pm0.018$ mm d$^{-1}$ decade$^{-1}$) are equivalent to an increase of $0.22\pm0.10$ mm d$^{-1}$
($13.8\pm6.2$ % of the long-term spring mean) and $0.20\pm0.09$ mm d$^{-1}$ ($11.2\pm5.3$ % of the long-term
spring mean) over the length of the dataset respectively. The radiative component of PET has
similarly significant trends in spring, while the aerodynamic component has no significant
trends in any season, except the English Lowlands in autumn (Fig. 12).
There are few studies of long-term trends in AED in the UK.  MORECS provides an estimate
of Penman-Monteith PET with interception correction calculated directly from the 40 km
resolution meteorological data (Hough and Jones, 1997; Thompson et al., 1981), and increases
can be seen over the dataset (Rodda and Marsh, 2011). But as the PET and PETI in the current
dataset are ultimately calculated using the same meteorological data (albeit by different
methods), it is not unexpected that similar trends should be seen. Site-based studies suggest an
increase over recent decades (Burt and Shahgedanova, 1998; Crane and Hudson, 1997), but it
is difficult to separate climate-driven trends from local land-use trends. A global review paper
(McVicar et al., 2012) identified a trend of decreasing AED in the northern hemisphere, driven
by decreasing wind speeds, however they also reported significant local variations on trends in
pan evaporation, including the increasing trend observed by Stanhill and Möller (2008) at a site
in England after 1968. Matsoukas et al. (2011) identified a statistically significant increase in
PET in several regions of the globe, including southern England, between 1983 and 2008,
attributing it predominantly to an increase in the radiative component of PET, due to global
brightening. However, these results were obtained using reanalysis data, which is limited in its
ability to capture trends in wind speed. This limitation has been documented in both northern
(Pryor et al., 2009) and southern (McVicar et al., 2008) hemispheres.
Regional changes in actual evaporative losses can be estimated indirectly using regional
precipitation and runoff or river flow. Using a combination of observations and modelling,
Marsh and Dixon (2012) identified an increase in evaporative losses in Great Britain from 1961-
2011. Hannaford and Buys (2012) note seasonal and regional differences in trends in observed
river flow, suggesting that decreasing spring flows in the English lowlands are indicative of
increasing AED. However, changing evaporative losses can also be due to changing supply
through precipitation, so it is important to formally attribute the trends in PET to changing
climate, in order to understand changing evapotranspiration.

## 4.3   Attribution of trends in potential evapotranspiration

In order to attribute changes in PET to changes in climate, the rate of change of PET, $dE_p/dt$
(mm d$^{-1}$ decade$^{-1}$), can be calculated as a function of the rate of change of each input variable
(Roderick et al., 2007),

$$\frac{dE_P}{dt} = \frac{dE_P}{dT_a}\frac{dT_a}{dt} + \frac{dE_P}{dq_a}\frac{dq_a}{dt} + \frac{dE_P}{du_{10}}\frac{du_{10}}{dt} + \frac{dE_P}{dL_d}\frac{dL_d}{dt} + \frac{dE_P}{dS_d}\frac{dS_d}{dt} \ . \tag{18}$$

Note that we exclude the surface air pressure, because this dataset uses a mean-monthly
climatology as the interannual variability of air pressure is negligible. The derivative of the PET
with respect to each of the meteorological variables can be found analytically (Appendix C).
The derivatives are calculated from the daily meteorological data at 1 km resolution.
Substituting the slopes of the linear regressions of the gridded annual means (Appendix B) for
the rate of change of each variable with time, and the overall time-average of the derivatives of
PET with respect to the meteorological variables, the contribution of each variable to the rate
of change of PET is calculated at 1 km resolution. These are then averaged over the regions of
interest. The same is also applied to the radiative and aerodynamic components independently.
Note that this can also be applied to the regional means of the derivatives of PET and the
regional trends in the meteorological variables. The results are compared in Table 3 and the two
approaches are consistent. For the regional analysis, we also quote the 95% CI. However, for
the gridded values, there is such high spatial coherence that combining the 95% CI over the
region results in unreasonably constrained results. We therefore use the more conservative CI
obtained from the regional analysis. Also note that this method assumes that the rate of change
of the variables with respect to time is constant over the seasonal cycle (and thus the product of
the means is equal to the mean of the products), and indeed this is how it is often applied
(Donohue et al., 2010; Lu et al., 2016). The effect of this assumption was investigated by
repeating the analysis with seasonal trends and means, but this makes negligible difference to
the results.
Figure 13 shows the contribution of each meteorological variable to the rate of change of the
annual mean PET and to the radiative and aerodynamic components and compares the total
attributed trend to that obtained by linear regression. The percentage contribution is in Table 4,
calculated as a fraction of the fitted trend. The final column shows the total attributed trend (i.e.
the sum of the previous columns) as a percentage of the fitted trend, to demonstrate the success
of the attribution at recovering the fitted trends. For the PET trend and for the trend in the
radiative component, these values generally sum to the linear regression to within a few percent.
However, for the aerodynamic component, the fitted trends are much smaller than the statistical
uncertainty. This means that there can be a large and/or negative percentage difference between
the attributed and fitted trends, even when the absolute difference is negligible.
The largest overall contribution to the rate of change of PET comes from increasing air
temperature, which has the effect of increasing the aerodynamic component but decreasing the
radiative component. The latter effect is due to approximating the surface temperature with the
air temperature in the calculation of upwelling LW radiation. This assumption is applied as it
simplifies the surface energy balance but it may introduce artefacts into the calculation of PET.
A more thorough formulation of PET, which linearises the net radiation in the derivation of the
Penman-Monteith equation, can be calculated to allow for a non-negligible difference between
air and surface temperature (Monteith, 1981; Thompson et al., 1981), but the difference
between the more thorough formulation and the formulation used here is small, particularly for
the temperature range of GB.
Note that in this calculation we are assuming that air temperature and downward LW radiation
vary independently, while in reality (and implicit in the calculation of downward LW in Sect.
2.4), downward LW radiation is also a function of the air temperature so that increases in
downward LW may broadly cancel the increasing upwelling LW radiation. If we instead were
to use net LW radiation as the independent variable, it is likely that dependence of the rate of
change of the radiative component on air temperature would be reduced in magnitude and
compensated by the rate of change of net LW radiation.
Overall the next largest increases are caused by increasing downward SW radiation, particularly
in the English regions in the spring, as it increases the radiative component of PET. However,
in Scotland and Wales, the increasing downward LW radiation is also important. Increasing
specific humidity strongly decreases the PET by decreasing the aerodynamic component, while
the decreasing wind speed has the effect of increasing the radiative component, but more
strongly decreasing the aerodynamic component, so overall it tends to cause a decrease in PET.
Since the increasing air temperature and downward LW and SW radiation have the effect of
increasing PET, but the increasing specific humidity and decreasing wind speed tend to
decrease it, then the overall trend is positive, but smaller than the trend due to air temperature
alone.

## 4.4   Relative humidity

The increase in PET due to increasing air temperature is largely cancelled by the decrease due
to increasing specific humidity so that the overall trend is smaller than the contribution to the
increase from air temperature alone. However, although we have assumed that specific
humidity and air temperature are independent variables, they are in fact coevolving in a
warming atmosphere. As air temperature increases, the saturated specific humidity increases
according to the Clausius-Clapeyron relation (Schneider et al., 2010). However, since
evaporation also increases with rising temperature, the increased water flux into the atmosphere
ensures that specific humidity also increases and it can be shown that there is likely to be little
change in global relative humidity even with significant change in global temperature (Held
and Soden, 2006; Schneider et al., 2010), although this may vary regionally over land (Dai,
2006). Although it is not completely independent of air temperature, an alternative way of
assessing the drivers of AED is to consider relative humidity, $R_h$, as the independent humidity
variable. In this case, the PET can be recast in terms of relative humidity, such that
$$E_P = \frac{t_d}{\lambda} \frac{\Delta A + \frac{c_p \rho_a}{r_a} q_s (1 - R_h)}{\Delta + \gamma \left(1 + \frac{r_s}{r_a}\right)} \ . \qquad (19)$$
Relative humidity is calculated from the specific humidity using
$$R_h = \frac{q_a}{q_s} \ . \qquad (20)$$
Although in this case relative humidity is a function of air temperature, through the saturated
specific humidity, in reality they are often found to behave as independent variables. It has been
shown that there is little cancellation of the air temperature and relative humidity terms when
studying both historical data  and future climate projections (Scheff and Frierson, 2014).
The relative humidity annual means, mean-monthly climatology and seasonal trends can be
seen in Fig. 14. We find a statistically significant negative trend in relative humidity in the
spring and autumn (except Wales in the autumn) but no overall negative trend in winter or
summer and no significant trend in the annual means. Maps of the overall mean relative
humidity and the trend in the annual mean are in Fig B3. There are only small regions in the
west of Scotland and the east and south west of England where there are significant trends in
the annual mean.
We calculate an alternative attribution using relative humidity as a variable, rather than specific
humidity, such that
$$\frac{dE_P}{dt} = \frac{dE_P}{dT_a}\frac{dT_a}{dt} + \frac{dE_P}{dR_h}\frac{dR_h}{dt} + \frac{dE_P}{du_{10}}\frac{du_{10}}{dt} + \frac{dE_P}{dL_d}\frac{dL_d}{dt} + \frac{dE_P}{dS_d}\frac{dS_d}{dt} \quad . \tag{21}$$
We then calculate the derivative of the PET with respect to relative humidity and the derivatives
with respect to air temperature and pressure are now taken at constant $R_h$ rather than constant
$q_a$, so these are also recalculated. See Appendix C for details.
Figure 15 shows the contribution of the different variables to the rate of change of PET with
this alternative formulation. The total attributed change is nearly the same as that in Fig. 13,
although there are small differences due to statistical uncertainty in the fits. The contributions
of downward SW and LW radiation and of wind speed to the rate of change of PET are
unchanged. Although it is not statistically significant, the negative trend in relative humidity
leads to an increase in the aerodynamic component, which is larger than the increase due to
increasing downward SW radiation.  The contribution of air temperature to the rate of change
is significantly reduced compared to the specific humidity formulation.  The air temperature-
driven decrease in the radiative component now largely cancels the temperature-driven increase
in the aerodynamic component, which is much smaller than in Sect. 4.3 as it now implicitly
includes the rising specific humidity. However, the effect of air temperature on the radiative
component comes through the effect of air temperature on the upwelling LW radiation in the
calculation of net radiation and this is dependent on the simplifying assumption that the surface
temperature is equal to the air temperature when solving the energy balance. If the fully
linearised version of the Penman-Monteith equation were used (Monteith, 1981), then the
dependence on air temperature would be more complicated as it would account for a non-
negligible difference between air and surface temperature. This may result in a different
contribution of air temperature to the changing PET, although this difference is likely to be
small.

## 5   Discussion

These high resolution datasets provide insight into the effect of the changing climate of Great
Britain on AED over the past five decades. There have been significant climatic trends in the
UK since 1961; in particular rising air temperature and specific humidity, decreasing wind
speed and decreasing cloudiness. Although some are positive and some negative, these
meteorological trends combine to give statistically significant trends in PET.
Wind speeds have decreased more significantly in the west than the east, and show a consistent
decrease across seasons. Contrary to Donohue et al. (2010) and McVicar et al. (2012), this study
finds that the change in wind speed of the late 20$^{th}$ and early 21$^{st}$ centuries has not had a
dominant influence on PET over the period of study, although it has mitigated the increasing
trend in PET. However, the previous studies were concerned with open-water Penman
evaporation, which has a simpler (proportional) dependence on wind speed than the Penman-
Monteith PET considered here (Schymanski and Or, 2015).
The air temperature trend in this study of $0.21\pm0.15$ K decade$^{-1}$ in GB is consistent with
observed global and regional trends (Hartmann et al., 2013; Jenkins et al., 2008). The
temperature trend is responsible for a large contribution to the trend in PET, although the large
negative contribution from the specific humidity (as well as a small negative contribution from
wind speed) means that the overall trend is smaller than the temperature trend alone.
When the attribution is recast in terms of relative humidity, the effect of air temperature is much
smaller, supporting the hypothesis that the temperature and specific humidity components
cancel because their changes are part of the same thermodynamic warming processes. Much of
the increase in the aerodynamic component due to air temperature is cancelled by the decrease
of the radiative component, which is due to the effect of air temperature on the calculated
upwelling LW radiation. However this is because of the assumption that surface temperature
can be approximated with air temperature, thus the real physical contribution of air temperature
in the relative humidity formulation is likely to be roughly equal to the increase in the
aerodynamic component. Although the relative humidity does not have a statistically significant
trend overall (although there are significant trends in spring and for some regions in autumn),
it is large enough that the negative trend in relative humidity is the largest contribution to the
increasing PET, followed by the downward SW radiation. This corresponds well to recent
findings in Spain (Vicente-Serrano et al., 2016).
The trend in relative humidity is consistent with that seen in historical regional (Jenkins et al.,
2008) and global (Dai, 2006; Willett et al., 2014) analyses. Although not statistically significant
overall, it contributes to between 57 % and 68 % of the trends in PET (between 39 % and 46 %
or the trends in spring PET). Globally trends in relative humidity vary spatially, with mid-
latitudes showing a decrease and the tropics and high-latitudes showing an increase, despite an
overall increase in specific humidity over land, particularly in the Northern Hemisphere (Dai,
2006; Willett et al., 2014). In these global analyses, Great Britain is in a region of transition
between decreasing relative humidity in Western Europe and increasing relative humidity in
Scandinavia, so that small decreasing trends are found, but they are not significant; this is
consistent with our findings. We found the relative humidity to be decreasing significantly in
spring, which is also when the downward SW is increasing. This is consistent with reduced
precipitation and cloud cover due to changing weather patterns (Sutton and Dong, 2012).
Increasing solar radiation has been shown to increase spring and annual AED, contributing to
between 18 % and 50 % of the fitted trend in annual PET, and to between 43 % and 53 % of
the fitted trend in spring PET. Two main mechanisms can be responsible for changing solar
radiation: changing cloud cover and changing aerosol concentrations. Changing aerosol
emissions have been shown to have had a significant effect on solar radiation in the $20^{th}$ century.
In Europe, global dimming due to increased aerosol concentrations peaked around 1980,
followed by global brightening as aerosol concentrations decreased (Wild, 2009). Observations
of changing continental runoff and river flow in Europe over the $20^{th}$ century have been
attributed to changing aerosol concentrations, via their effect on solar radiation, and thus AED
(Gedney et al., 2014).
In this study we use the duration of bright sunshine to calculate the solar radiation, using
empirical coefficients which do not vary with year, so aerosol effects are not explicitly included
and the trend in downward SW is driven by the increase in sunshine hours in the MORECS
dataset ($0.088 \pm 0.055$ h $d^{-1}$ decade$^{-1}$ over GB). The coefficients used in this study to convert
sunshine hours to radiation fluxes were empirically derived in 1978; the derivation used data
from the decade 1966-75, as this period was identified to be before reductions in aerosol
emissions had begun to significantly alter observed solar radiation (Cowley, 1978). Despite
this, the trend in SW radiation in the current dataset from 1979 onwards ($1.4 \pm 1.4$ W m$^{-2}$ decade$^{-}$
[1]) is consistent, within uncertainties, with that seen over GB in the WFDEI data ($0.9 \pm 1.1$ W m$^{-}$
$^{2}$ decade$^{-1}$), which is bias-corrected to observations and includes explicit aerosol effects
(Weedon et al., 2014).
It has been suggested that aerosol effects also implicitly affect sunshine duration since in
polluted areas there will be fewer hours above the official 'sunshine hours' threshold of 120
Wm$^{-2}$ (Helmes and Jaenicke, 1986). Several regional studies have shown trends in sunshine
hours that are consistent with the periods of dimming and brightening across the globe (eg
Liley, 2009; Sanchez-Lorenzo et al., 2009; Sanchez-Lorenzo et al., 2008; Stanhill and Cohen,
2005), and several have attempted to quantify the relative contribution of trends in cloud cover
and aerosol loading (e.g. Sanchez-Lorenzo and Wild (2012) in Switzerland, see Sanchez-
Romero et al. (2014) for a review). Therefore, it may be that some of the brightening trend seen
in the current dataset is due to the implicit signal of aerosol trends in the MORECS sunshine
duration, although this is likely to be small compared to the effects of changing cloud cover.
The trends in the MORECS sunshine duration used in this study are consistent with changing
weather patterns which may be attributed to the Atlantic Multidecadal Oscillation (AMO). The
AMO has been shown to cause a decrease in spring precipitation (and therefore cloud cover) in
northern Europe over recent decades (Sutton and Dong, 2012), and the trend in MORECS
sunshine hours is dominated by an increase in the spring mean. This has also been seen in
Europe-wide sunshine hours data (Sanchez-Lorenzo et al., 2008) and is also consistent with the
falling spring relative humidity found in the current study. On the other hand, the effect of
changing aerosols on sunshine hours is expected to be largest in the winter (Sanchez-Lorenzo
et al., 2008). However, it would not be possible to directly identify either of these effects on the
sunshine duration without access to longer data records.
The inclusion of explicit aerosol effects in the coefficients of the Ångström-Prescott equation
would be expected to reduce the positive trend in AED in the first two decades of the dataset,
and increase it after 1980. Gedney et al. (2014) attribute a decrease in European solar radiation
of 10 W m$^{-2}$ between the periods 1901-10 and 1974-80, and an increase of 4 W m$^{-2}$ from 1974-
84 to1990-99 to changing aerosol contributions. Applying these trends to the current dataset,
with a turning point at 1980, would double the overall increase in solar radiation in Great
Britain, which would lead to a 40 % increase in the overall trend in PET. So, if this effect were
to be included, it would confirm the results found in this paper.
Although the contribution is generally smaller (except in Scotland), the trends in LW radiation
in these datasets contribute to between 15% and 27% of the trends in PET and between 27%
and 46% of the trends in the radiative component. In Scotland the downward LW radiation is
the dominant driver of changing PET in the relative humidity formulation. Note, however, that
this is largely cancelled by the increasing upwelling LW, which is captured in this study in the
effect of air temperature on the radiative component, and which may be different if the
approximation that the difference between air temperature and surface temperature is negligible
were relaxed. Observations of LW radiation are often uncertain, but the trend in this dataset,
although small, is consistent with observed trends (Wang and Liang, 2009), as well as with
trends in the WFDEI bias-corrected reanalysis product (Weedon et al., 2014).
Trends in temperature and cloud cover in the UK are expected to continue into the coming
decades, with precipitation expected to increase in the winter but decrease in the summer
(Murphy et al., 2009). Therefore it is likely that AED will increase, increasing water stress in
the summer when precipitation is lower and potentially affecting water resources, agriculture
and biodiversity. This has been demonstrated for southern England and Wales by Rudd and
Kay (2015), who calculated present and future PET using high-resolution RCM output and
included the effects of $CO_2$ on stomatal opening.
The current study is concerned only with the effects of changing climate on AED and has
assumed a constant bulk canopy resistance throughout. However, plants are expected to react
to increased $CO_2$ in the atmosphere by closing stomata and limiting the exchange of gases,
including water (Kruijt et al., 2008), and observed changes in runoff have been attributed to this
effect (Gedney et al., 2006; Gedney et al., 2014). It is possible that the resulting change of
canopy resistance could partially offset the increased atmospheric demand (Rudd and Kay,
2015) and may impact runoff (Gedney et al., 2006; Prudhomme et al., 2014), but further studies
would be required to quantify this.
**6 Conclusion**
This paper has presented a unique, high-resolution, observation-based dataset of meteorological
variables and AED in Great Britain since 1961. Key trends in the meteorological variables are
(i) increasing air temperature and specific humidity, consistent with global temperature trends;
(ii) increasing solar radiation, particularly in the spring, consistent with changes in aerosol
emissions and weather patterns in recent decades; (iii) decreasing wind speed, consistent with
observations of global stilling; (iv) increasing precipitation, driven by increasing winter
precipitation in Scotland; and (v) no significant trend in relative humidity overall, but
decreasing relative humidity in the spring. The meteorological variables were used to evaluate
AED in Great Britain via calculation of PET and PETI. It has been demonstrated that including
the interception component in the calculation of PETI gives a mean estimate that is overall 8%
larger than PET alone, with strong seasonality and spatial variation of the difference. PET was
found to be increasing by $0.021 \pm 0.021$ mm $d^{-1}$ decade$^{-1}$ in GB over the study period. With the
interception component included, the trend in PETI is weaker ($0.019 \pm 0.020$ mm $d^{-1}$), and over
GB is not significant at the 5% level. The trend in PET was analytically attributed to the trends
in the meteorological variables, and it was found that the dominant effect was that increasing
air temperature was driving increasing PET, with smaller increases from increased downward
SW and LW radiation. However, the effect of temperature is largely compensated by the
associated increase in specific humidity, while decreasing wind speed tended to decrease the
PET. When the attribution was recast in terms of relative humidity, temperature was found to
have a small effect on the trend in PET due to cancellation between the increase in the
aerodynamic component and decrease in the radiative component, while the decreasing relative
humidity caused PET to increase, at a similar rate to the downward SW radiation (and
downward LW radiation in Scotland). The increase in PET due to these variables is mitigated
by the observed northern hemisphere wind stilling, which causes a decrease in PET, however,
the overall trend in PET is positive over the period of study.
In addition to providing meteorological data and estimates of AED for analysis, the
meteorological variables provided are sufficient to run LSMs and hydrological models. The
high spatial (1 km) and temporal (daily) resolution will allow this dataset to be used to study
the effects of climate on physical and biological systems at a range of scales, from local to
national.
**Data Access**
The data can be downloaded from the Environmental Information Platform at the Centre for
Ecology & Hydrology. The meteorological variables (CHESS-met) can be found at
https://catalogue.ceh.ac.uk/documents/80887755-1426-4dab-a4a6-250919d5020c,
while the PET and PETI (CHESS-PE) can be accessed at
https://catalogue.ceh.ac.uk/documents/d329f4d6-95ba-4134-b77a-a377e0755653.
**Author contribution**
EB, JF and DBC designed the study. JF, ACR, DBC and ELR developed code to create
meteorological data. ELR created the PET and PETI. ELR and EB analysed trends. ELR, EB,
ACR and DBC wrote the manuscript.
**Acknowledgements**
The meteorological variables presented are based largely on GB meteorological data under
licence from the Met Office, and those organisations contributing to this national dataset
(including the Met Office, Environment Agency, Scottish Environment Protection Agency
(SEPA) and Natural Resources Wales) are gratefully acknowledged. The CRU TS 3.21 daily
temperature range data were created by the University of East Anglia Climatic Research Unit,
and the WFD air pressure data were created as part of the EU FP6 project WATCH (Contract
036946). Collection of flux data was funded by EU FP4 EuroFlux (Griffin Forest); EU FP5
CarboEuroFlux (Griffin Forest); EU FP5 GreenGrass (Easter Bush); EU FP6 CarboEuropeIP
(Alice Holt , Griffin Forest, Auchencorth Moss, Easter Bush); EU FP6 IMECC (Griffin
Forest); the Forestry Commission (Alice Holt); the Natural Environment Research Council,
UK (Auchencorth Moss, Easter Bush).
Fig. 1, panels a) and b) of Fig. 6 and panel a) of Fig. B3 were produced with the python
implementation of the cubehelix colour scheme (Green, 2011).
Thanks to Nicola Gedney and Graham Weedon for useful discussions.
Thanks to three anonymous reviewers, who provided insightful and helpful comments.
This work was partially funded by the Natural Environment Research Council in the
Changing Water Cycle programme: NERC Reference: NE/I006087/1.

**Appendix A: Data validation**

Meteorological data were downloaded from the European Fluxes Database Cluster (http://gaia.agraria.unitus.it) for four sites positioned around Great Britain. Two were woodland sites (Alice Holt (Wilkinson et al., 2012; Heinemeyer et al., 2012) and Griffin Forest (Clement, 2003)), while two had grass and crop cover (Auchencorth Moss (Billett et al., 2004) and Easter Bush (Gilmanov et al., 2007; Soussana et al., 2007)). Table A1 gives details of the data used. The data are provided as half-hourly measurements, which were used to create daily means, where full daily data coverage was available. The daily means of the observed data were compared to the daily data from the grid square containing the site and the Pearson correlation ($r^2$), mean bias and root mean square error (RMSE) were calculated. For each site, monthly means were calculated where the full month had available data, then a climatology calculated from available months. The same values were calculated from the relevant grid squares, using only time periods for which observed data were available.

Fig. A1 shows the comparison of the data set downward SW radiation against daily mean air temperature observed at the four sites. Fig. A2 shows the mean-monthly climatology of the daily values. The observed values of the mixing ratio of water vapour in air were compared with values calculated from the meteorological dataset, using the equation

$$r_w = q_a \left( \frac{m_a}{m_w} \right) \tag{A1}$$

where $m_a$ is the molecular mass of dry air and $m_w$ is the molecular mass of water. The comparisons are shown in Figs. A3 and A4.

Table A2 shows the $r^2$, mean bias and RMSE for each of the variables included in the validation exercise. The correlations indicate a good relationship between the dataset variables and the independent observations at the sites, while the mean-monthly climatologies demonstrate that the data represent the seasonal cycle well. The data set downward SW in Auchencorth Moss is biased high compared to the observations, while the wind speed is biased high at two sites.

**Appendix B: Trend maps**

Fig. B1 shows the rate of change of each of the meteorological variables at the 1 km resolution, while Fig. B2 shows the rate of change of the PET, PETI, and the two components of PET at the same resolution. This shows that the regional trends are consistent with spatial variation and are not dominated by individual extreme points.

## Appendix C: Derivatives of PET

The wind speed affects the PET through the aerodynamic resistance. The derivative with respect to wind speed is

$$\frac{\partial E_P}{\partial u_{10}} = \frac{(\Delta+\gamma)E_{PA}-\gamma\frac{r_s}{r_a}E_{PR}}{u_{10}\left(\Delta+\gamma\left(1+\frac{r_s}{r_a}\right)\right)} \ . \tag{C1}$$

The downward LW and SW radiation affect PET through the net radiation, and the derivatives are

$$\frac{\partial E_P}{\partial L_d} = E_{PR}\frac{\epsilon}{R_n} \tag{C2}$$

$$\frac{\partial E_P}{\partial S_d} = E_{PR}\frac{(1-\alpha)}{R_n} \ . \tag{C3}$$

The derivative of PET with respect to specific humidity is

$$\frac{\partial E_P}{\partial q_a} = \frac{E_{PA}}{q_a-q_s} \ . \tag{C4}$$

The air temperature affects PET through the saturated specific humidity and its derivative, the net radiation and the air density, so that the derivative of PET with respect to air temperature is

$$\frac{\partial E_P}{\partial T_a} = E_{PR}\left[\left(1-\frac{\Delta}{\Delta+\gamma\left(1+\frac{r_s}{r_a}\right)}\right)\left(\frac{T_{sp}}{T_a^2}\frac{\sum_{i=1}^4 i(i-1)a_iT_r^{i-2}}{\sum_{i=1}^4 ia_iT_r^{i-1}}+\Delta\frac{p_*+(1-\varepsilon)e_s}{p_*q_s}-\frac{2}{T_a}\right)-\frac{4\epsilon\sigma T_a^3}{R_n}\right]+$$

$$E_{PA}\left[\frac{\Delta}{q_s-q_a}-\frac{1}{T_a}-\frac{\Delta}{\Delta+\gamma\left(1+\frac{r_s}{r_a}\right)}\left(\frac{T_{sp}}{T_a^2}\frac{\sum_{i=1}^4 i(i-1)a_iT_r^{i-2}}{\sum_{i=1}^4 ia_iT_r^{i-1}}+\Delta\frac{p_*+(1-\varepsilon)e_s}{p_*q_s}-\frac{2}{T_a}\right)\right] \ . \tag{C5}$$

When calculating the attribution with relative humidity as the dependent variable, the derivative of PET with respect to relative humidity is

$$\frac{\partial E_P}{\partial R_h} = \frac{E_{PA}}{R_h-1} \ , \tag{C6}$$

and the derivative of PET with respect to air temperature is

$$\frac{\partial E_P}{\partial T_a} = E_{PR}\left[\left(1-\frac{\Delta}{\Delta+\gamma\left(1+\frac{r_s}{r_a}\right)}\right)\left(\frac{T_{sp}}{T_a^2}\frac{\sum_{i=1}^4 i(i-1)a_iT_r^{i-2}}{\sum_{i=1}^4 ia_iT_r^{i-1}}+\Delta\frac{p_*+(1-\varepsilon)e_s}{p_*q_s}-\frac{2}{T_a}\right)-\frac{4\epsilon\sigma T_a^3}{R_n}\right]+$$

$$E_{PA}\left[\frac{\Delta}{q_s}-\frac{1}{T_a}-\frac{\Delta}{\Delta+\gamma\left(1+\frac{r_s}{r_a}\right)}\left(\frac{T_{sp}}{T_a^2}\frac{\sum_{i=1}^4 i(i-1)a_iT_r^{i-2}}{\sum_{i=1}^4 ia_iT_r^{i-1}}+\Delta\frac{p_*+(1-\varepsilon)e_s}{p_*q_s}-\frac{2}{T_a}\right)\right] \ . \tag{C7}$$

The difference between Eq. C7 and Eq. C5 is the factor of $\Delta/q_s$ instead of $\Delta/(q_s-q_a)$ in the second bracket.

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

Table 1. Description of input meteorological variables

| Variable (units) | Source data | Ancillary files | Assumptions | Height |
|---|---|---|---|---|
| Air temperature (K) | MORECS air temperature | IHDTM elevation | Lapsed to IHDTM elevation | 1.2 m |
| Specific humidity (kg kg$^{-1}$) | MORECS vapour pressure | IHDTM elevation | Lapsed to IHDTM elevation<br><br>Constant air pressure = 100 kPa | 1.2 m |
| Downward LW radiation (W m$^{-2}$) | MORECS air temperature, vapour pressure, sunshine hours | IHDTM elevation | Constant cloud base height | 1.2 m |
| Downward SW radiation (W m$^{-2}$) | MORECS sunshine hours | IHDTM elevation<br><br>Spatially-varying aerosol correction | No time-varying aerosol correction | 1.2 m |
| Wind speed (m s$^{-1}$) | MORECS wind speed | ETSU average wind speeds | Wind speed correction is constant | 10 m |
| Precipitation (kg m$^{-2}$ s$^{-1}$) | CEH-GEAR precipitation | - | No transformations performed | n/a |
| Daily temperature range (K) | CRU TS 3.21 daily temperature range | - | No spatial interpolation from 0.5° resolution.<br><br>No temporal interpolation | 1.2 m |

| | | | (constant values for each month) | |
|---|---|---|---|---|
| Surface air pressure (Pa) | WFD air pressure | IHDTM elevation | Mean-monthly values from WFD used (each year has same values). Lapsed to IHDTM elevation. No temporal interpolation (constant values for each month). | n/a |


Table 2: Rate of change of annual means of meteorological and potential evapotranspiration
variables in Great Britain. Bold indicates trends that are significant at the 5% level. The
ranges are given by the 95% CI.

| Variable | Rate of change ± 95% CI | | | | |
| | Great Britain | England | Scotland | Wales | English lowlands |
|---|---|---|---|---|---|
| Air temperature (K dec$^{-1}$) | **0.21 ± 0.15** | **0.23 ± 0.14** | **0.17 ± 0.12** | **0.21 ± 0.15** | **0.25 ± 0.17** |
| Specific humidity (g kg$^{-1}$ dec$^{-1}$) | **0.049 ± 0.037** | **0.054 ± 0.04** | **0.040 ± 0.036** | **0.055 ± 0.037** | **0.053 ± 0.044** |
| Downward SW radiation (W m$^{-2}$ dec$^{-1}$) | **1.0 ± 0.8** | **1.3 ± 1.0** | 0.5 ± 0.6 | **1.1 ± 0.9** | **1.5 ± 1.0** |
| Downward LW radiation (W m$^{-2}$ dec$^{-1}$) | **0.50 ± 0.48** | 0.45 ± 0.48 | **0.58 ± 0.48** | 0.50 ± 0.55 | 0.42 ± 0.48 |
| Wind speed (m s$^{-1}$ dec$^{-1}$) | **-0.18 ± 0.09** | **-0.16 ± 0.09** | **-0.20 ± 0.10** | **-0.25 ± 0.16** | **-0.13 ± 0.07** |
| Precipitation (mm d$^{-1}$ dec$^{-1}$) | **0.08 ± 0.06** | 0.04 ± 0.06 | **0.14 ± 0.09** | 0.08 ± 0.09 | 0.03 ± 0.05 |
| Daily temperature range (K dec$^{-1}$) | -0.06 ± 0.06 | -0.03 ± 0.06 | **-0.13 ± 0.08** | 0.00 ± 0.06 | -0.04 ± 0.07 |
| Relative humidity (% dec$^{-1}$) | -0.39 ± 0.44 | -0.43 ± 0.46 | -0.33 ± 0.33 | -0.36 ± 0.4 | -0.50 ± 0.53 |
| PET (mm d$^{-1}$ dec$^{-1}$) | **0.021 ± 0.021** | **0.025 ± 0.024** | **0.015 ± 0.015** | 0.017 ± 0.021 | **0.03 ± 0.026** |
| Radiative component of PET (mm d$^{-1}$ dec$^{-1}$) | **0.016 ± 0.010** | **0.018 ± 0.011** | **0.013 ± 0.008** | **0.020 ± 0.013** | **0.018 ± 0.011** |
| Aerodynamic component of PET (mm d$^{-1}$ dec$^{-1}$) | 0.007 ± 0.011 | 0.009 ± 0.013 | 0.004 ± 0.009 | 0.001 ± 0.013 | 0.015 ± 0.015 |
| PETI (mm d$^{-1}$ dec$^{-1}$) | 0.019 ± 0.020 | **0.023 ± 0.023** | 0.014 ± 0.014 | 0.016 ± 0.020 | **0.028 ± 0.025** |


Table 3. Contributions to the rate of change of PET and its radiative and aerodynamic components. For each variable, the first column shows the contribution calculated using regional averages, along with the associated 95% CI. The second column shows the contribution calculated at 1 km resolution, then averaged over each region. The uncertainty on this value is difficult to calculate as the pixels are highly spatially correlated, so the uncertainty range from the regional analysis is used in Fig. 13.

a) Contribution to rate of change of PET (mm d$^{-1}$ decade$^{-1}$)

| | Air temperature | | Specific humidity | | Wind speed | | Downward LW | | Downward SW | | Total | |
|---|---|---|---|---|---|---|---|---|---|---|---|---|
| | Regional | Pixel | Regional | Pixel | Regional | Pixel | Regional | Pixel | Regional | Pixel | Regional | Pixel |
| England | **0.041 ± 0.025** | 0.039 | **-0.025 ± 0.019** | -0.024 | **-0.010 ± 0.005** | -0.007 | 0.005 ± 0.006 | 0.005 | **0.013 ± 0.009** | 0.012 | 0.025 ± 0.034 | 0.024 |
| Scotland | **0.029 ± 0.021** | 0.023 | **-0.020 ± 0.018** | -0.017 | **-0.010 ± 0.005** | -0.007 | **0.006 ± 0.005** | 0.006 | 0.005 ± 0.005 | 0.004 | 0.010 ± 0.029 | 0.008 |
| Wales | **0.039 ± 0.028** | 0.036 | **-0.026 ± 0.018** | -0.025 | **-0.011 ± 0.007** | -0.009 | 0.006 ± 0.006 | 0.006 | **0.010 ± 0.009** | 0.009 | 0.017 ± 0.036 | 0.017 |
| English lowlands | **0.043 ± 0.029** | 0.042 | **-0.024 ± 0.020** | -0.023 | **-0.008 ± 0.004** | -0.008 | 0.005 ± 0.006 | 0.005 | **0.015 ± 0.010** | 0.015 | 0.031 ± 0.038 | 0.030 |
| Great Britain | **0.037 ± 0.026** | 0.031 | **-0.023 ± 0.018** | -0.022 | **-0.010 ± 0.005** | -0.007 | **0.006 ± 0.005** | 0.005 | **0.010 ± 0.007** | 0.007 | 0.019 ± 0.033 | 0.014 |

b) Contribution to rate of change of radiative component of (mm d$^{-1}$ decade$^{-1}$)

| | Air temperature | | Specific humidity | | Wind speed | | Downward LW | | Downward SW | | Total | |
|---|---|---|---|---|---|---|---|---|---|---|---|---|
| | Regional | Pixel | Regional | Pixel | Regional | Pixel | Regional | Pixel | Regional | Pixel | Regional | Pixel |
| England | **-0.009 ± 0.006** | -0.009 | n/a | n/a | **0.009 ± 0.005** | 0.007 | 0.005 ± 0.006 | 0.005 | **0.014 ± 0.010** | 0.013 | **0.018 ± 0.013** | 0.016 |
| Scotland | **-0.006 ± 0.005** | -0.006 | n/a | n/a | **0.009 ± 0.004** | 0.007 | **0.006 ± 0.005** | 0.006 | 0.005 ± 0.005 | 0.004 | **0.014 ± 0.010** | 0.012 |
| Wales | **-0.007 ± 0.005** | -0.007 | n/a | n/a | **0.014 ± 0.009** | 0.013 | 0.006 ± 0.006 | 0.006 | **0.010 ± 0.009** | 0.010 | **0.023 ± 0.015** | 0.022 |
| English lowlands | **-0.010 ± 0.007** | -0.010 | n/a | n/a | **0.007 ± 0.004** | 0.006 | 0.005 ± 0.006 | 0.005 | **0.016 ± 0.011** | 0.015 | **0.017 ± 0.014** | 0.017 |
| Great Britain | **-0.008 ± 0.006** | -0.007 | n/a | n/a | **0.009 ± 0.005** | 0.007 | **0.006 ± 0.006** | 0.006 | **0.010 ± 0.008** | 0.008 | **0.017 ± 0.012** | 0.013 |

c) Contribution to rate of change of aerodynamic component of PET (mm d$^{-1}$ decade$^{-1}$)

| | Air temperature | | Specific humidity | | Wind speed | | Downward LW | | Downward SW | | Total | |
|---|---|---|---|---|---|---|---|---|---|---|---|---|
| | Regional | Pixel | Regional | Pixel | Regional | Pixel | Regional | Pixel | Regional | Pixel | Regional | Pixel |

| | | | | | | | | | | | | |
|---|---|---|---|---|---|---|---|---|---|---|---|---|
| England | **0.052 ± 0.032** | 0.050 | **-0.026 ± 0.020** | -0.026 | **-0.018 ± 0.010** | -0.015 | n/a | n/a | n/a | n/a | 0.007 ± 0.039 | 0.009 |
| Scotland | **0.037 ± 0.027** | 0.033 | **-0.021 ± 0.019** | -0.019 | **-0.019 ± 0.010** | -0.015 | n/a | n/a | n/a | n/a | -0.003 ± 0.034 | -0.001 |
| Wales | **0.048 ± 0.035** | 0.046 | **-0.028 ± 0.019** | -0.027 | **-0.026 ± 0.016** | -0.023 | n/a | n/a | n/a | n/a | -0.005 ± 0.042 | -0.003 |
| English lowlands | **0.056 ± 0.037** | 0.055 | **-0.026 ± 0.021** | -0.025 | **-0.015 ± 0.008** | -0.014 | n/a | n/a | n/a | n/a | 0.015 ± 0.044 | 0.015 |
| Great Britain | **0.046 ± 0.033** | 0.041 | **-0.025 ± 0.019** | -0.023 | **-0.020 ± 0.010** | -0.015 | n/a | n/a | n/a | n/a | 0.002 ± 0.039 | 0.003 |


Table 4. Contribution of the trend in each variable to the trends in annual mean PET and its
radiative and aerodynamic components as a percentage of the fitted trend in PET and its
components.

| a) Potential evapotranspiration (PET) | | | | | | |
|---|---|---|---|---|---|---|
| | Air temperature | Specific humidity | Wind speed | Downward LW | Downward SW | Total |
| England | 154 % | -88 % | -22 % | 17 % | 47 % | 108 % |
| Scotland | 150 % | -74 % | -23 % | 26 % | 18 % | 97 % |
| Wales | 200 % | -130 % | -38 % | 28 % | 50 % | 109 % |
| English lowlands | 142 % | -77 % | -20 % | 15 % | 45 % | 105 % |
| Great Britain | 155 % | -87 % | -23 % | 19 % | 31 % | 96 % |
| b) Radiative component of PET | | | | | | |
| | Air temperature | Specific humidity | Wind speed | Downward LW | Downward SW | Total |
| England | -47 % | n/a | 40 % | 28 % | 71 % | 92 % |
| Scotland | -42 % | n/a | 62 % | 46 % | 36 % | 102 % |
| Wales | -34 % | n/a | 69 % | 29 % | 52 % | 116 % |
| English lowlands | -53 % | n/a | 35 % | 27 % | 86 % | 95 % |
| Great Britain | -44 % | n/a | 46 % | 31 % | 53 % | 87 % |
| c) Aerodynamic component of PET | | | | | | |
| | Air temperature | Specific humidity | Wind speed | Downward LW | Downward SW | Total |
| England | 245 % | -115 % | -48 % | n/a | n/a | 82 % |
| Scotland | 68 % | -14 % | -33 % | n/a | n/a | 21 % |
| Wales | -135 % | 72 % | -42 % | n/a | n/a | -105 % |
| English lowlands | 282 % | -126 % | -47 % | n/a | n/a | 109 % |
| Great Britain | 168 % | -76 % | -44 % | n/a | n/a | 48 % |


Table 5. Contributions to the rate of change of PET and its radiative and aerodynamic
components when relative humidity is used. For each variable, the first column shows the
contribution calculated using regional averages, along with the associated 95% CI. The
second column shows the contribution calculated at 1 km resolution, then averaged over each
region. The uncertainty on this value is difficult to calculate as the pixels are highly spatially
correlated, so the uncertainty range from the regional analysis is used in Fig. 13.

a) Contribution to rate of change of PET (mm d$^{-1}$ decade$^{-1}$)

| | Air temperature | | Relative humidity | | Wind speed | | Downward LW | | Downward SW | | Total | |
|---|---|---|---|---|---|---|---|---|---|---|---|---|
| | Regional | Pixel | Regional | Pixel | Regional | Pixel | Regional | Pixel | Regional | Pixel | Regional | Pixel |
| England | **-0.002 ± 0.001** | -0.000 | 0.015 ± 0.016 | 0.013 | **-0.010 ± 0.005** | -0.007 | 0.005 ± 0.006 | 0.005 | **0.013 ± 0.009** | 0.012 | **0.021 ± 0.020** | 0.023 |
| Scotland | **-0.001 ± 0.001** | 0.000 | 0.011 ± 0.011 | 0.008 | **-0.010 ± 0.005** | -0.007 | **0.006 ± 0.005** | 0.006 | 0.005 ± 0.005 | 0.004 | 0.010 ± 0.014 | 0.011 |
| Wales | **-0.002 ± 0.001** | -0.000 | 0.013 ± 0.014 | 0.012 | **-0.011 ± 0.007** | -0.009 | 0.006 ± 0.006 | 0.006 | **0.010 ± 0.009** | 0.009 | 0.015 ± 0.019 | 0.018 |
| English lowlands | **-0.003 ± 0.002** | -0.000 | 0.017 ± 0.018 | 0.017 | **-0.008 ± 0.004** | -0.008 | 0.005 ± 0.006 | 0.005 | **0.015 ± 0.010** | 0.015 | **0.026 ± 0.022** | 0.028 |
| Great Britain | **-0.002 ± 0.001** | 0.000 | 0.013 ± 0.015 | 0.011 | **-0.010 ± 0.005** | -0.007 | **0.006 ± 0.005** | 0.005 | **0.010 ± 0.007** | 0.007 | 0.016 ± 0.018 | 0.016 |

b) Contribution to rate of change of radiative component of (mm d$^{-1}$ decade$^{-1}$)

| | Air temperature | | Relative humidity | | Wind speed | | Downward LW | | Downward SW | | Total | |
|---|---|---|---|---|---|---|---|---|---|---|---|---|
| | Regional | Pixel | Regional | Pixel | Regional | Pixel | Regional | Pixel | Regional | Pixel | Regional | Pixel |
| England | **-0.009 ± 0.006** | -0.009 | n/a | n/a | **0.009 ± 0.005** | 0.007 | 0.005 ± 0.006 | 0.005 | **0.014 ± 0.010** | 0.013 | **0.018 ± 0.013** | 0.016 |
| Scotland | **-0.006 ± 0.005** | -0.006 | n/a | n/a | **0.009 ± 0.004** | 0.007 | **0.006 ± 0.005** | 0.006 | 0.005 ± 0.005 | 0.004 | **0.014 ± 0.010** | 0.012 |
| Wales | **-0.007 ± 0.005** | -0.007 | n/a | n/a | **0.014 ± 0.009** | 0.013 | 0.006 ± 0.006 | 0.006 | **0.010 ± 0.009** | 0.010 | **0.023 ± 0.015** | 0.022 |
| English lowlands | **-0.010 ± 0.007** | -0.010 | n/a | n/a | **0.007 ± 0.004** | 0.006 | 0.005 ± 0.006 | 0.005 | **0.016 ± 0.011** | 0.015 | **0.017 ± 0.014** | 0.017 |
| Great Britain | **-0.008 ± 0.006** | -0.007 | n/a | n/a | **0.009 ± 0.005** | 0.007 | **0.006 ± 0.006** | 0.006 | **0.010 ± 0.008** | 0.008 | **0.017 ± 0.012** | 0.013 |

c) Contribution to rate of change of aerodynamic component of PET (mm d$^{-1}$ decade$^{-1}$)

| | Air temperature | | Relative humidity | | Wind speed | | Downward LW | | Downward SW | | Total | |
|---|---|---|---|---|---|---|---|---|---|---|---|---|
| | Regional | Pixel | Regional | Pixel | Regional | Pixel | Regional | Pixel | Regional | Pixel | Regional | Pixel |

| | | | | | | | | | | | | |
|---|---|---|---|---|---|---|---|---|---|---|---|---|
| England | **0.006 ± 0.004** | 0.006 | 0.015 ± 0.017 | 0.014 | **-0.018 ± 0.010** | -0.015 | n/a | n/a | n/a | n/a | 0.003 ± 0.020 | 0.004 |
| Scotland | **0.004 ± 0.003** | 0.004 | 0.011 ± 0.011 | 0.009 | **-0.019 ± 0.010** | -0.015 | n/a | n/a | n/a | n/a | -0.004 ± 0.015 | -0.002 |
| Wales | **0.005 ± 0.004** | 0.005 | 0.013 ± 0.015 | 0.012 | **-0.026 ± 0.016** | -0.023 | n/a | n/a | n/a | n/a | -0.007 ± 0.022 | -0.006 |
| English lowlands | **0.007 ± 0.004** | 0.006 | 0.018 ± 0.019 | 0.017 | **-0.015 ± 0.008** | -0.014 | n/a | n/a | n/a | n/a | 0.009 ± 0.021 | 0.010 |
| Great Britain | **0.005 ± 0.004** | 0.005 | 0.014 ± 0.015 | 0.011 | **-0.020 ± 0.010** | -0.015 | n/a | n/a | n/a | n/a | -0.001 ± 0.019 | 0.000 |


Table 6. Contribution of the trend in each variable to the trends in annual mean PET and its
radiative and aerodynamic components as a percentage of the fitted trend in PET and its
components when relative humidity is used.

| a) Potential evapotranspiration (PET) | | | | | |
| --- | --- | --- | --- | --- | --- |
| | Air temperature | Relative humidity | Wind speed | Downward LW | Downward SW | Total |
| England | -0% | 57% | -22% | 17% | 47% | 99% |
| Scotland | 0% | 65% | -23% | 26% | 18% | 85% |
| Wales | -0% | 68% | -38% | 27% | 50% | 107% |
| English lowlands | -0% | 57% | -20% | 15% | 45% | 97% |
| Great Britain | 0% | 60% | -23% | 19% | 31% | 87% |
| b) Radiative component of PET | | | | | |
| | Air temperature | Relative humidity | Wind speed | Downward LW | Downward SW | Total |
| England | -47% | n/a | 40% | 28% | 71% | 92% |
| Scotland | -42% | n/a | 62% | 46% | 36% | 102% |
| Wales | -34% | n/a | 69% | 29% | 52% | 116% |
| English lowlands | -53% | n/a | 35% | 27% | 86% | 95% |
| Great Britain | -44% | n/a | 46% | 31% | 53% | 87% |
| c) Aerodynamic component of PET | | | | | |
| | Air temperature | Relative humidity | Wind speed | Downward LW | Downward SW | Total |
| England | 29% | 78% | -48% | n/a | n/a | 59% |
| Scotland | 8% | 14% | -33% | n/a | n/a | -11% |
| Wales | -15% | -33% | -42% | n/a | n/a | -90% |
| English lowlands | 33% | 98% | -47% | n/a | n/a | 84% |
| Great Britain | 19% | 52% | -44% | n/a | n/a | 27% |


Table A1. Details of sites used for validation of meteorological data.

| Site (ID) | Latitude | Longitude | Years | Land cover | Citation |
|---|---|---|---|---|---|
| Alice Holt (UK-Ham) | 51.15 | -0.86 | 2004-2012 | Deciduous broadleaf woodland | (Wilkinson et al., 2012; Heinemeyer et al., 2012) |
| Griffin Forest (UK-Gri) | 56.61 | -3.80 | 1997-2001, 2004-2008 | Evergreen needleleaf woodland | (Clement, 2003) |
| Auchencorth Moss (UK-AMo) | 55.79 | -3.24 | 2002-2006 | Grass and crop | (Billett et al., 2004) |
| Easter Bush (UK-EBu) | 55.87 | -3.21 | 2004-2008 | Grass | (Gilmanov et al., 2007; Soussana et al., 2007) |


Table A2. Correlation statistics for meteorological variables with data from four sites.

a) Air temperature

| Site | $r^2$ | Mean bias | RMSE |
|------|-------|-----------|------|
| Alice Holt | 0.95 | 0.10 K | 1.17 K |
| Griffin Forest | 0.94 | 0.21 K | 1.17 K |
| Auchencorth Moss | 0.98 | -0.02 K | 0.78 K |
| Easter Bush | 0.97 | -0.46 K | 0.96 K |

b) Downward SW radiation

| Site | $r^2$ | Mean bias | RMSE |
|------|-------|-----------|------|
| Alice Holt | 0.94 | -3.01 W m$^{-2}$ | 22.92 W m$^{-2}$ |
| Griffin Forest | 0.85 | -4.90 W m$^{-2}$ | 31.29 W m$^{-2}$ |
| Auchencorth Moss | 0.91 | 14.27 W m$^{-2}$ | 27.96 W m$^{-2}$ |
| Easter Bush | 0.88 | 5.73 W m$^{-2}$ | 27.15 W m$^{-2}$ |

c) Mixing ratio

| Site | $r^2$ | Mean bias | RMSE |
|------|-------|-----------|------|
| Alice Holt | 0.90 | -0.02 mmol mol$^{-1}$ | 1.09 mmol mol$^{-1}$ |
| Griffin Forest | 0.76 | 0.08 mmol mol$^{-1}$ | 1.56 mmol mol$^{-1}$ |

d) Wind speed

| Site | $r^2$ | mean bias | RMSE |
|------|-------|-----------|------|
| Alice Holt | 0.88 | 1.24 m s$^{-1}$ | 1.45 m s$^{-1}$ |
| Griffin Forest | 0.59 | 1.36 m s$^{-1}$ | 1.81 m s$^{-1}$ |
| Auchencorth Moss | 0.63 | -0.38 m s$^{-1}$ | 1.37 m s$^{-1}$ |
| Easter Bush | 0.82 | 0.44 m s$^{-1}$ | 1.03 m s$^{-1}$ |

e) Surface air pressure

| Site | $r^2$ | Mean bias | RMSE |
|------|-------|-----------|------|
| Griffin Forest | 0.05 | -0.42 hPa | 1.38 hPa |
| Auchencorth Moss | 0.01 | -1.06 hPa | 1.57 hPa |
| Easter Bush | 0.03 | 0.01 hPa | 1.33 hPa |


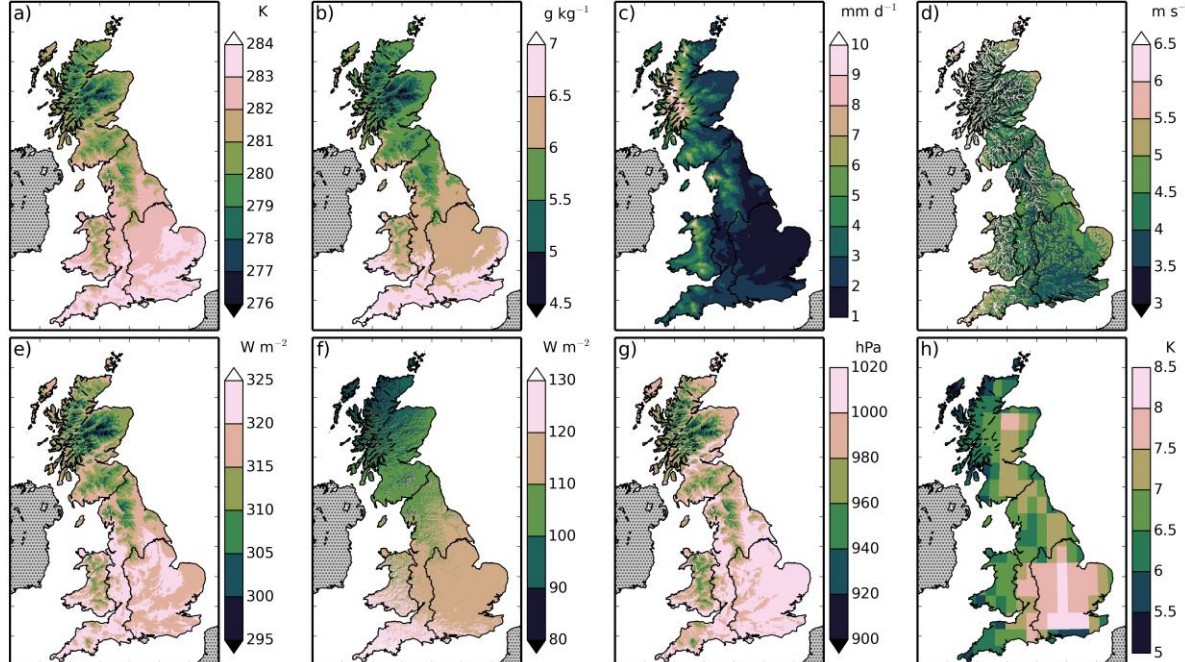


Figure 1. Means of the meteorological variables over the years 1961-2012. The variables are
a) 1.2 m air temperature, b) 1.2 m specific humidity, c) precipitation, d) 10 m wind speed, e)
downward LW radiation, f) downward SW radiation, g) surface air pressure, h) daily air
temperature range.
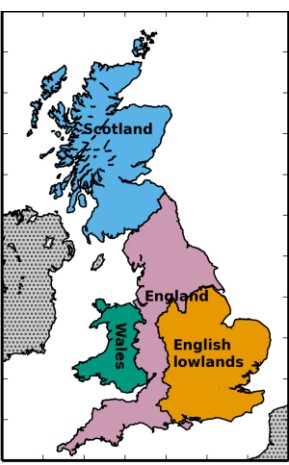
Figure 2. The regions used to calculate the area means. The English lowlands are a sub-region
of England. England, Scotland and Wales together form the fifth region, Great Britain.

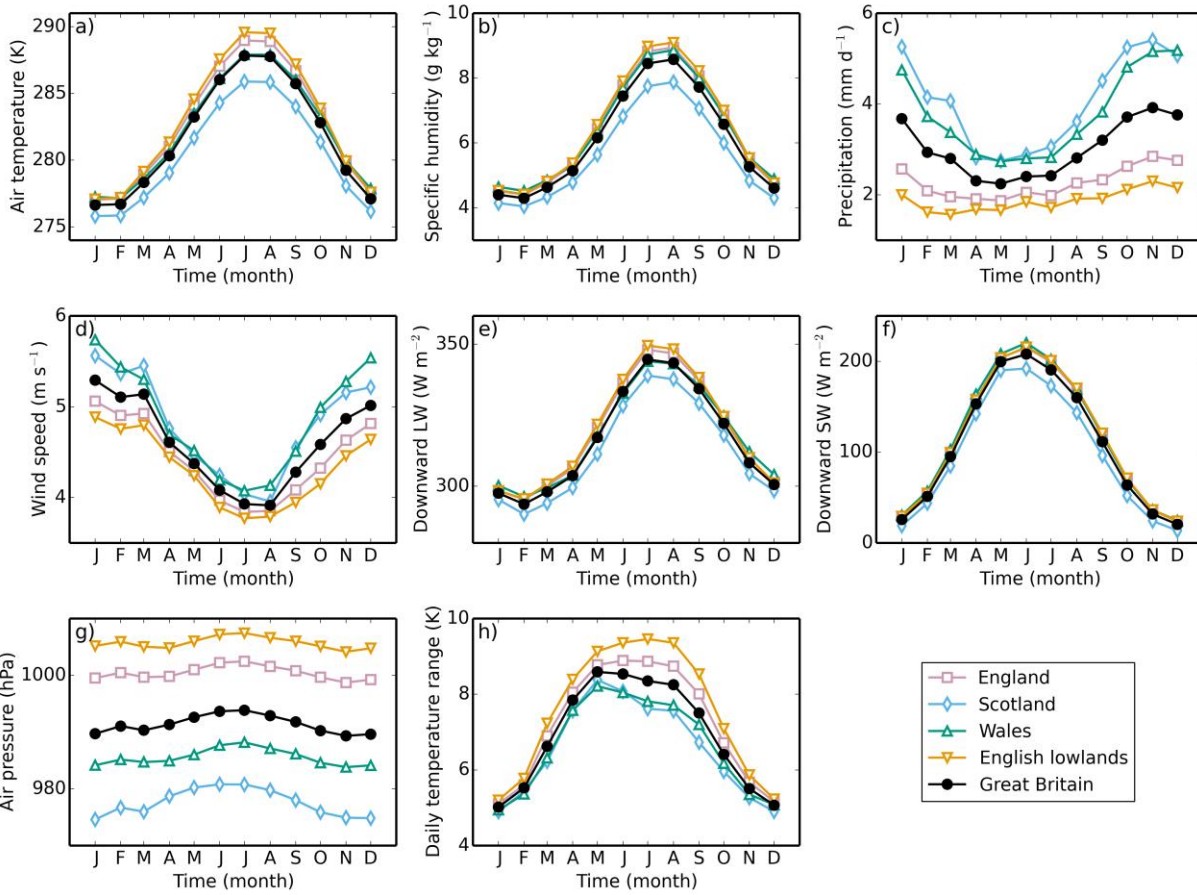


Figure 3. Mean monthly climatology of meteorological variables, a) 1.2 m air temperature, b)
1.2 m specific humidity, c) precipitation, d) 10 m wind speed, e) downward LW radiation, f)
downward SW radiation, g) surface air pressure, h) daily air temperature range, for five
different regions of Great Britain, calculated over the years 1961-2012.

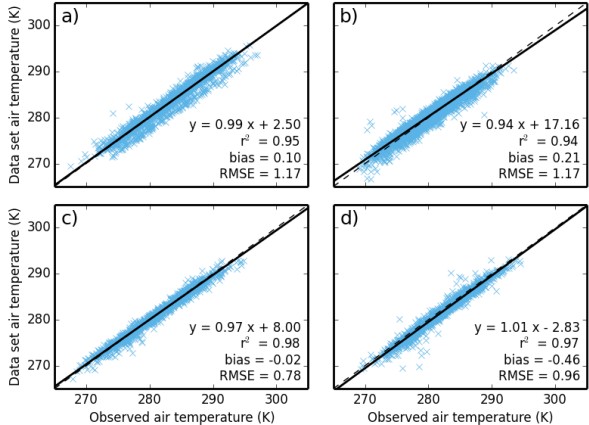


Figure 4. Plot of data set air temperature against daily mean observed air temperature at four
sites. The dashed line shows the one to one line, while the solid line shows the linear regression,
the equation of which is shown in the lower right of each plot, along with the $r^2$ value, the mean
bias and the RMSE. The sites are a) Alice Holt; b) Griffin Forest; c) Auchencorth Moss; d)
Easter Bush.

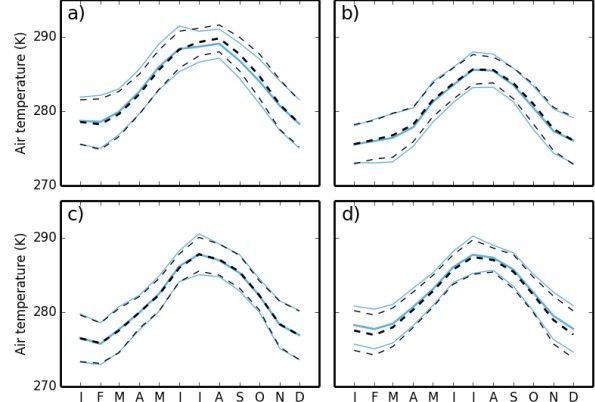

Figure 5. Mean monthly climatology of the dataset (black, dashed lines) and observed (blue,
solid lines) air temperatures, calculated for the period of observations. The thicker lines show
the means, while the thinner lines show the standard errors on each measurement. Sites as in
Fig. 4.

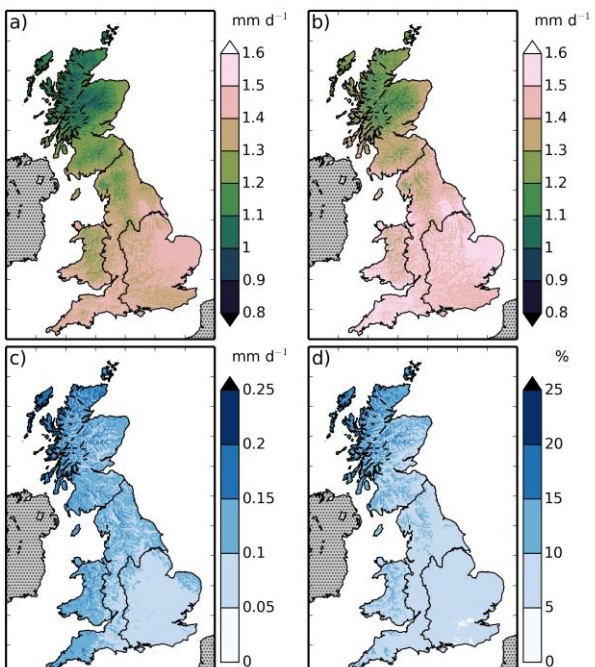


Figure 6. Mean a) PET, b) PETI, c) absolute difference between PETI and PET and d) relative
difference calculated over the years 1961-2012.

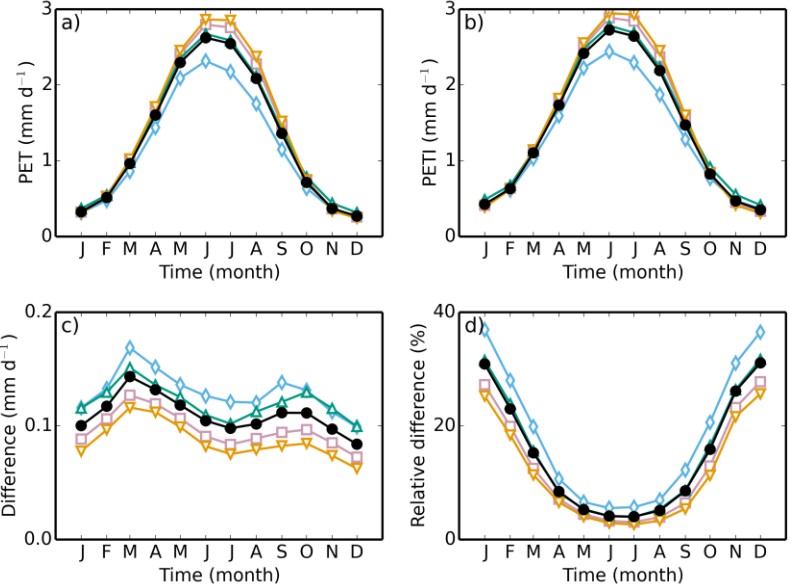


Figure 7. Mean monthly climatology of a) PET, b) PETI, c) absolute difference between PETI
and PET, d) relative difference, for five different regions of Great Britain, calculated over the
years 1961-2012. Symbols as in Fig. 3.

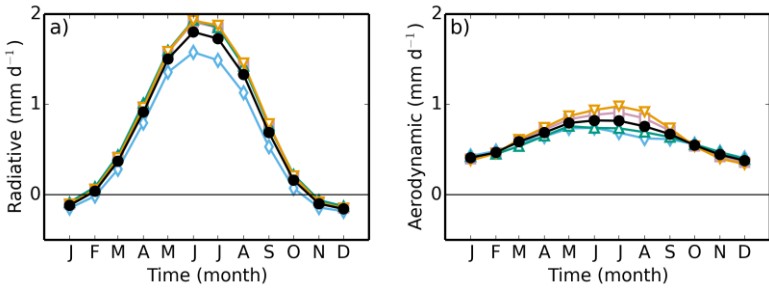


Figure 8. Mean-monthly climatology of the a) radiative and b) aerodynamic components of the
PET for five different regions of Great Britain, calculated over the years 1961-2012. Symbols
as in Fig. 3.

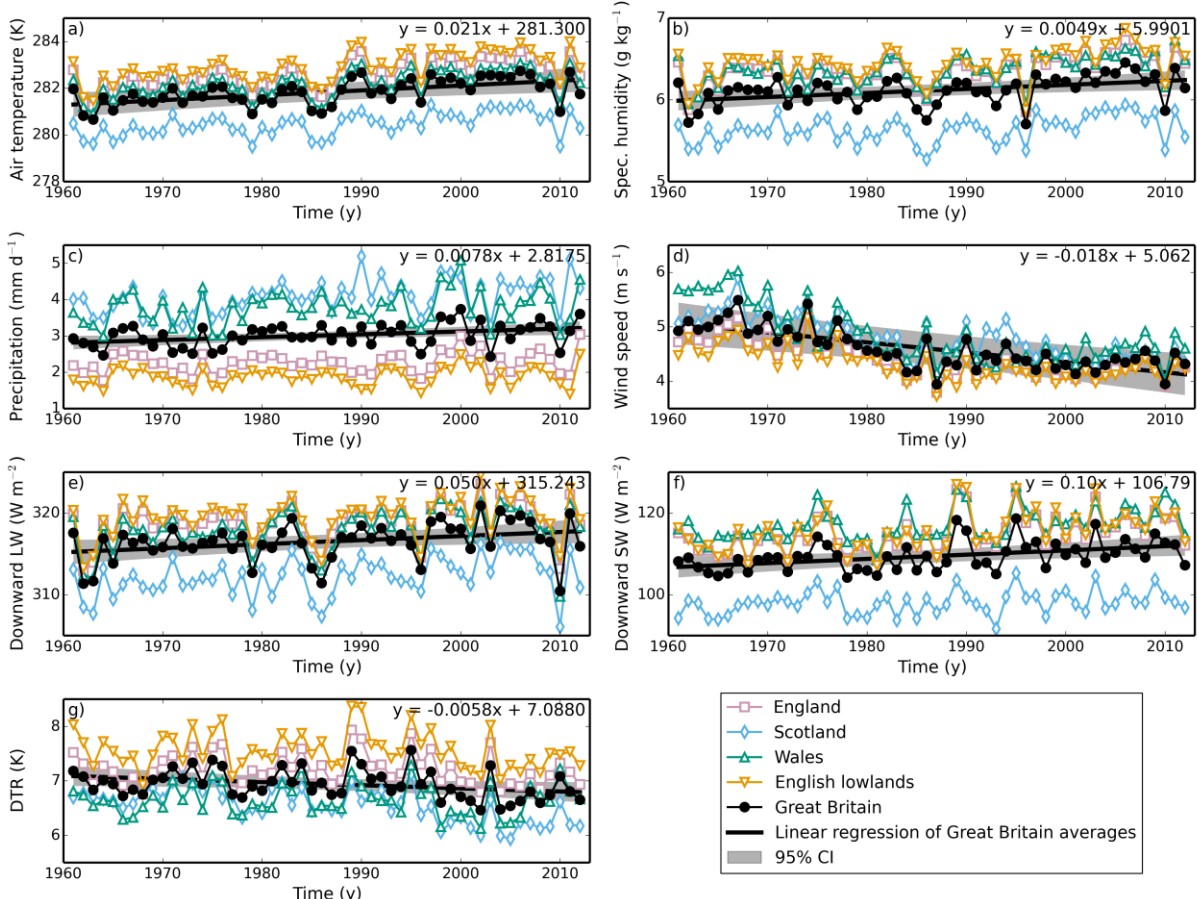


Figure 9. Annual means of the meteorological variables, a) 1.2 m air temperature, b) 1.2 m specific humidity, c) precipitation, d) 10 m wind speed, e) downward LW radiation, f) downward SW radiation, g) daily air temperature range, over five regions of Great Britain. The solid black lines show the linear regression fit to the Great Britain annual means, while the grey strip shows the 95% CI of the same fit, assuming a non-zero lag-1 correlation coefficient. The equation of this fit is shown in the top right-hand corner of each plot.

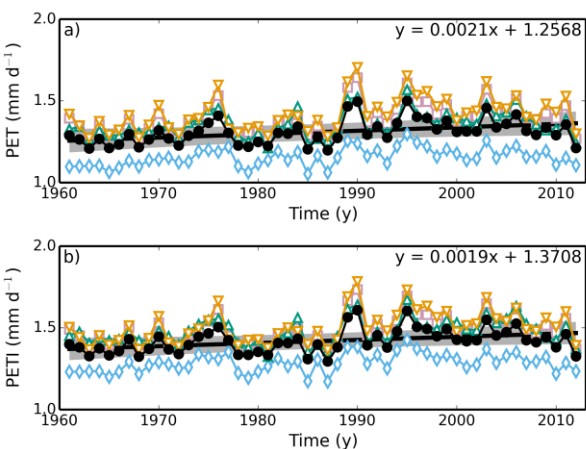


Figure 10. Annual means of a) PET and b) PETI for five regions of Great Britain. Symbols as
in Fig. 9.

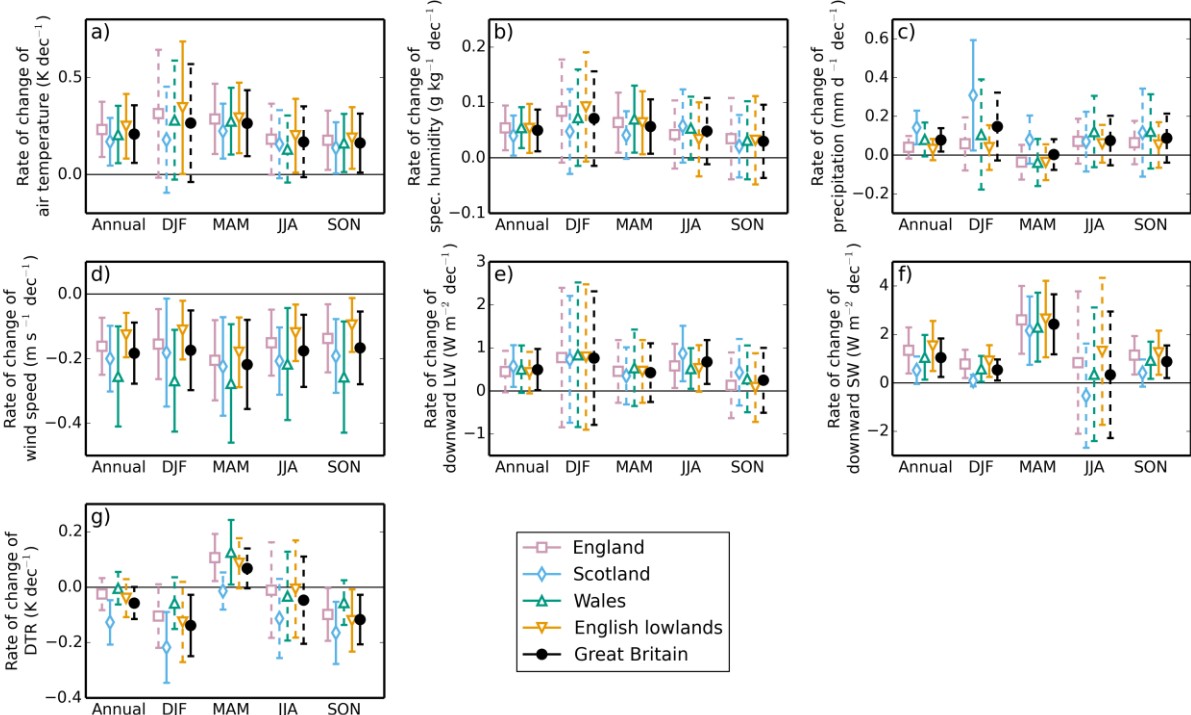


Figure 11. Rate of change of annual and seasonal means of meteorological variables, a) 1.2 m
air temperature, b) 1.2 m specific humidity, c) precipitation, d) 10 m wind speed, e) downward
LW radiation, f) downward SW radiation, g) daily air temperature range, for five regions of
Great Britain for the years 1961-2012. Error bars are the 95% CI calculated assuming a non-
zero lag-1 correlation coefficient. Solid error bars indicate slopes that are statistically significant
at the 5% level, dashed error bars indicate slopes that are not significant at the 5% level.

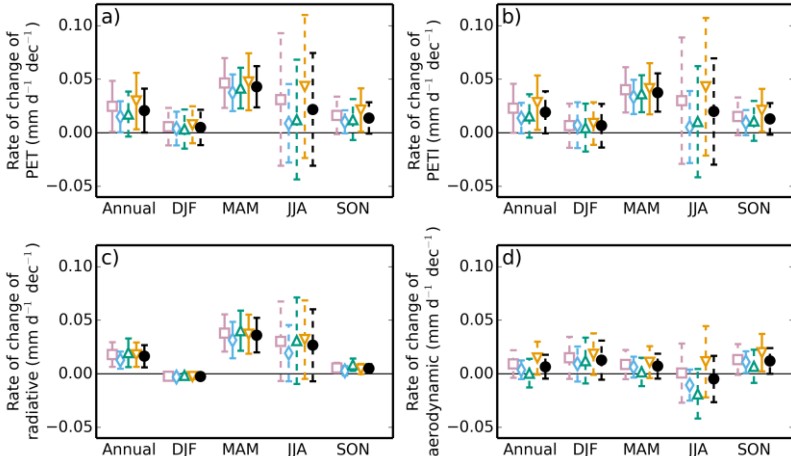


Figure 12. Rate of change of annual and seasonal means of a) PET, b) PETI, c) the radiative
component of PET and d) the aerodynamic component of PET for five regions of Great Britain
for the years 1961-2012. Symbols as in Fig. 11.

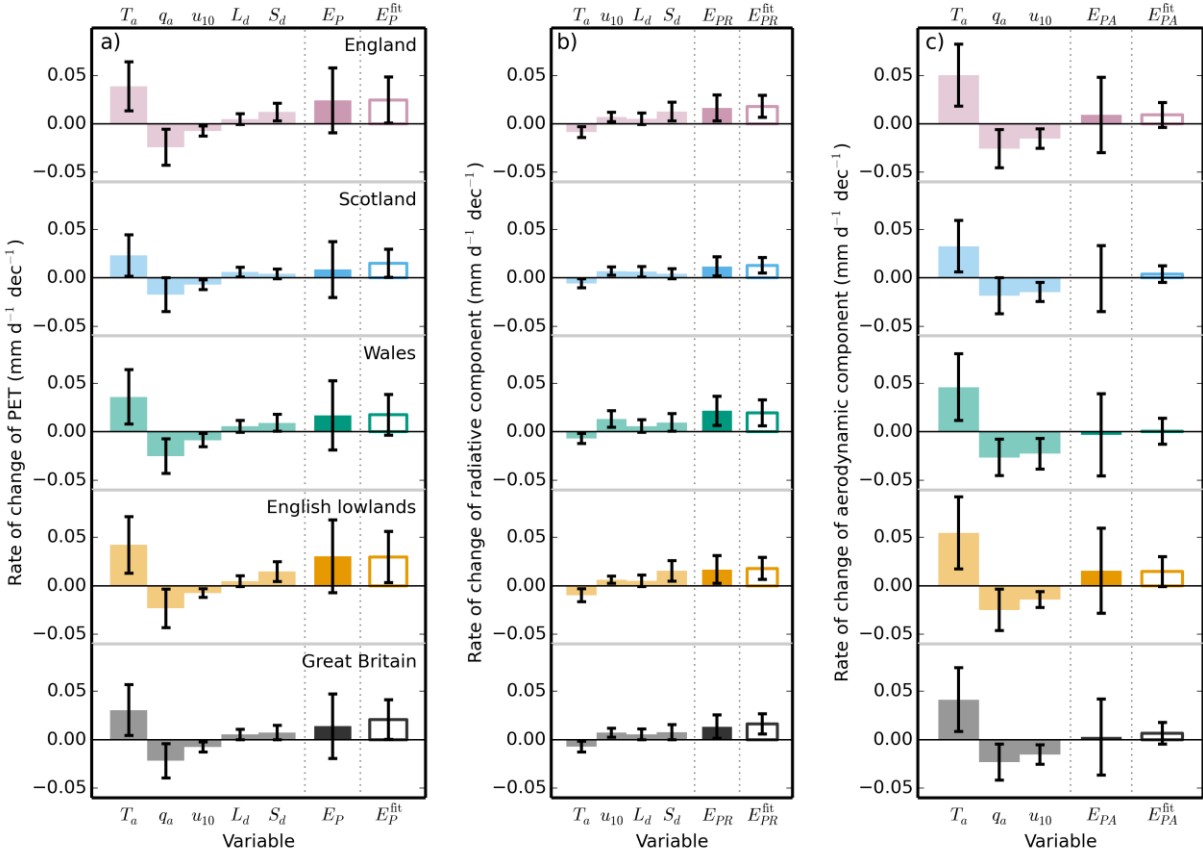


Figure 13. The contribution of the rate of change of each meteorological variable to the rate of change of a) PET, b) the radiative component and c) the aerodynamic component. The first five (four; three) bars are the contribution to the rate of change of annual mean PET from the rate of change of each of the variables, calculated per pixel, than averaged over each region. Each bar has an error bar showing the 95% CI on each value. Since the pixels are highly spatially correlated, we use the more conservative CI calculated by applying this analysis to the regional means. The next bar is the sum of the other bars and shows the attributed rate of change of annual mean PET. The final bar shows the slope and its associated CI obtained from the linear regression of the mean annual PET for each region.


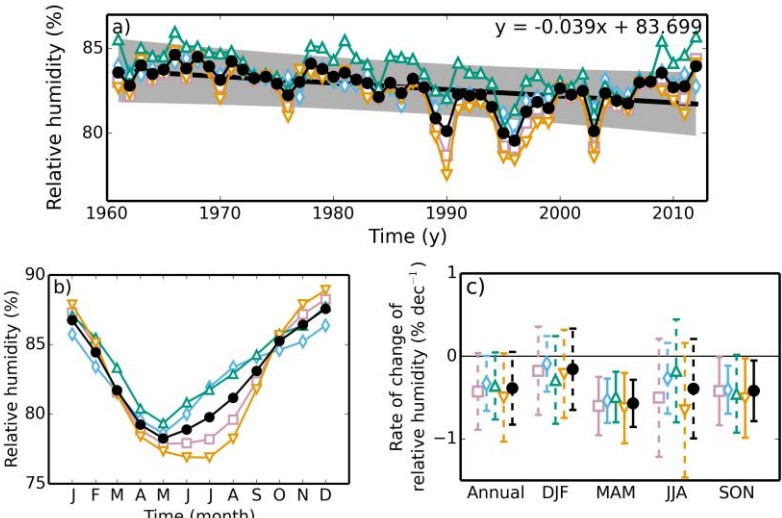


Figure 14. Regional annual means (a), regional mean-monthly climatology (b) and regional
rates of change of relative humidity for the years 1961-2012.

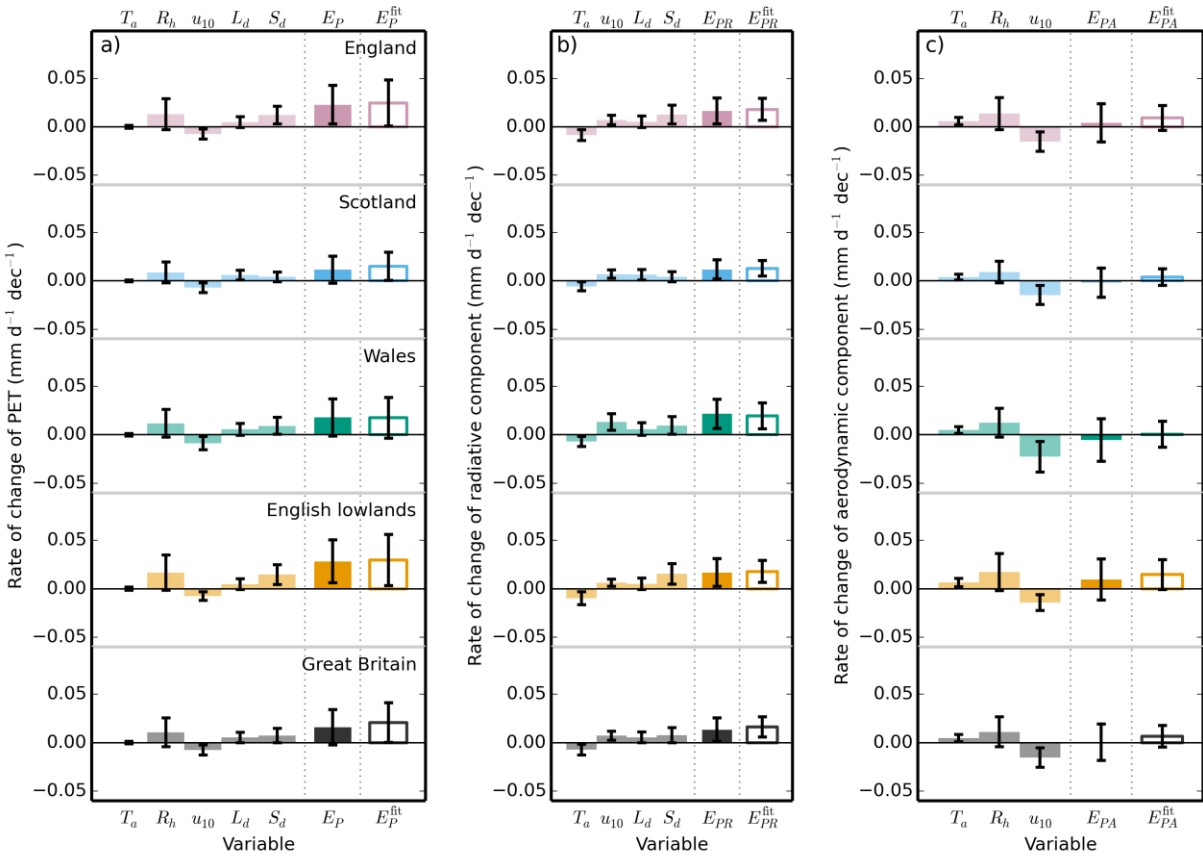


Figure 15. The contribution of the rate of change of each meteorological variable to the rate of
change of a) PET, b) the radiative component and c) the aerodynamic component, with relative
humidity instead of specific humidity. The first five (four; three) bars are the contribution to
the rate of change of annual mean PET from the rate of change of each of the variables,
calculated per pixel, than averaged over each region. Each bar has an error bar showing the
95% CI on each value. Since the pixels are highly spatially correlated, we use the more
conservative CI calculated by applying this analysis to the regional means. The next bar is the
sum of the other bars and shows the attributed rate of change of annual mean PET. The final
bar shows the slope and its associated CI obtained from the linear regression of the mean annual
PET for each region.


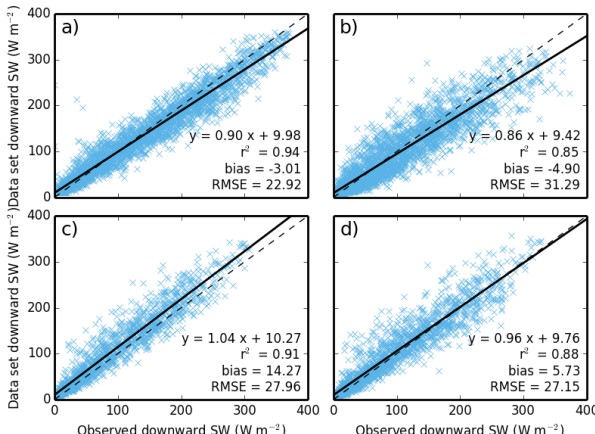


Figure A1. Plot of data set downward SW radiation against daily mean observed downward
SW radiation at four flux sites. Symbols and sites as in Fig. 4.


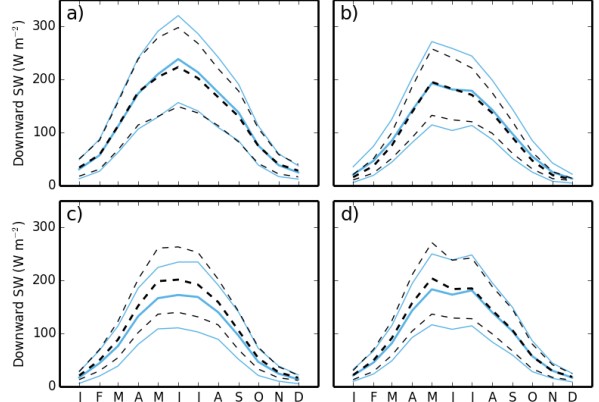


Figure A2. Mean monthly climatology of the dataset (black, dashed lines) and observed (blue,
solid lines) downward SW radiation, calculated for the period of observations. Symbols as in
Fig. 5, sites as in Fig. 4.

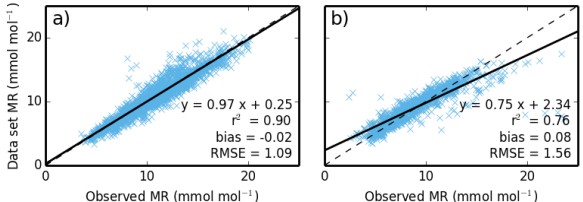


Figure A3. Plot of mixing ratio calculated using dataset meteorology against daily mean
observed mixing ratio at four sites. Symbols as in Fig. 4. The sites are a) Alice Holt and b)
Griffin Forest.

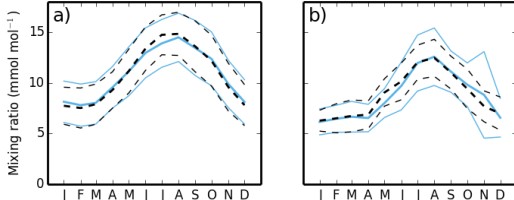

Figure A4. Mean monthly climatology of the dataset (black, dashed lines) and observed (blue,
solid lines) mixing ratio, calculated for the period of observations. Symbols as in Fig. 5. Sites
as in Fig. A3.

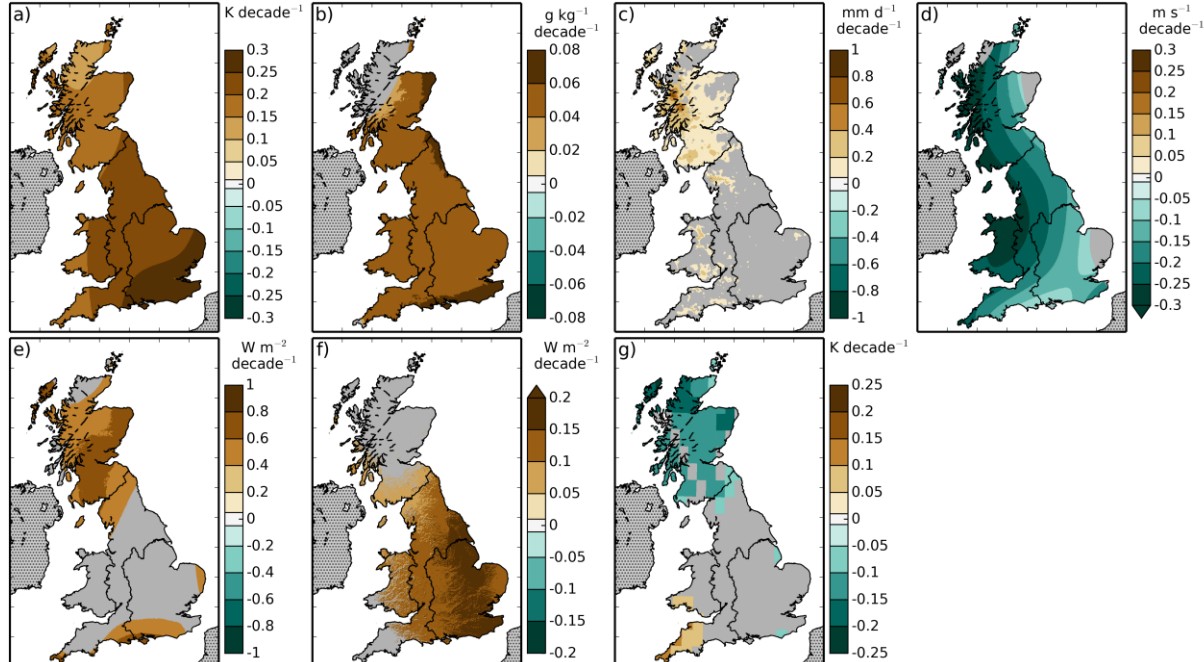


Figure B1. Rate of change of the annual means of the meteorological variables, a) 1.2 m air
temperature, b) 1.2 m specific humidity, c) precipitation, d) 10 m wind speed, e) downward LW
radiation, f) downward SW radiation, g) daily air temperature range over the period 1961-2012.
Areas for which the trend was not significant are shown in grey.

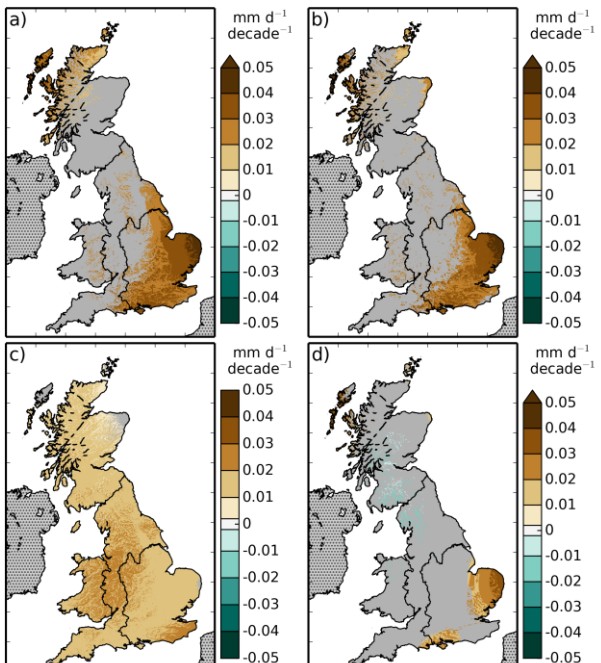


Figure B2. Rate of change the annual means of a) PET, b) PETI, c) the radiative component of
PET, d) the aerodynamic component of PET over the period 1961-2012. Areas for which the
trend was not significant are shown in grey.

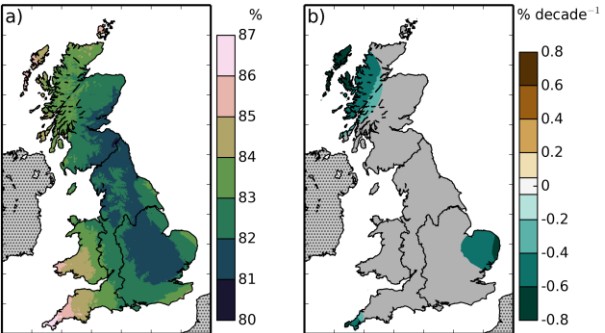


Figure B3. Maps of a) mean and b) rate of change of annual mean of the relative humidity over
the years 1961-2012. Areas for which the trend was not significant are shown in grey.