# Peer review of "Trends in atmospheric evaporative demand in Great Britain"

_Hydrology and Earth System Sciences, 2015_

## Short Comment (SC1) · 3 Feb 2016

Robinson et al have presented a new 1 x 1 km resolution dataset of meteorological variables for Great Britain (i.e. most of the UK). The paper is clearly structured and focusses on trends in evaporative demand and their attribution with the addition of allowance for interception by the canopy in calculations of potential evapotranspiration. A few minor changes suggested here could tighten the presentation a little more:

1) Page 10, line 21: It is correctly stated that six meteorological variables contribute to the standard Penman-Monteith ET calculation, but unfortunately the authors only listed five variables (they missed out wind speed).

2) Page 11 The discussion of interception impacts on ET calculation is welcome, but no mention has been made of the effect of snow as distinct from rain water on the canopy.

[Figure]

Presumably as there are relatively few snow days in most of Britain it is not necessary to make allowances for intercepted snow rather than intercepted rain, but snow should be mentioned at least.

3) Page 13 line 8: For clarity I think it would be clearer if in place of: "the 95% confidence intervals of the slope are calculated assuming a non-zero lag-1 autocorrelation,..." this was written as: "the 95% confidence intervals of the slope were calculated specifically allowing for the non-zero lag-1 autocorrelation...".

4) Page 13 line 31, p14 lines 1 and 2: add the 95% CIs for the trends.

5) Page 17 line 16: specify the trends (and 95% CIs) for SWdown for CHESS and for the WFDEI.

---

## Referee Comment (RC1) · Anonymous Referee #1 · 11 Feb 2016

General Comments:

This manuscript presents a new spatial, daily meteorological dataset for Great Britain for the years 1961-2012, whereby a special focus is given on trend estimation of potential evapotranspiration with and without considering interception storage and the effect of interception rain on stomatal opening. The paper is well structured and written, methods are clearly listed in the text or summarized in tables, using a good balance between explicit formulation and reference to former literature.

As a potential user of this data set and also in order to evaluate the derived trends, I would be interest on the overall quality of the data set itself and on the derived ETp estimates (incl. trends). Have there been evaluation (cross validation) of the spatial fields of meteorological variables such as radiation components, T, rH, wind speed etc.

[Figure]

Possible these are analysed or mentioned in the original literature, but at least the reader should explicitly get an idea of a possible range of uncertainties in that product.

The authors put some effort in addressing the effect of intercepted water on the "operation/functioning" of the stomata system. While my area of research is not in plant ecology, my general understanding is, that for most of the plant the stomata are located at the lower epidermis – how can snow and rain strongly influence stomata control by closing the stomata openings?

I would like to see those two points at least addressed in a revised version of the manuscript. As a result of my evaluation I would suggest moderate revisions of the manuscript before a possible publication in HESS.

———————————————

---

## Referee Comment (RC2) · Anonymous Referee #2 · 23 Feb 2016

This study derives a new 1km x 1km, daily, gridded dataset of Penman-Monteith potential evapotranspiration (PET), and its input variables, over all of Great Britain for 1961-2012 using existing station-based observational products. It then looks at the trends in this PET, and the reasons for those trends. This is a very useful and insightful study.

However, the procedure for the annual-mean PET trends due to individual variables contains a major conceptual error (and possibly also some sort of numerical or units error), there are no maps of trends (a main purpose for such a dataset), and the procedure for obtaining 1km x 1km specific humidity from the coarser-resolution parent dataset has several potential problems. Therefore, I recommend major revisions. I also have a number of minor and technical comments, listed after these major issues.

Major comments, beginning with the most important:

***p15 li10-11: It's not correct to multiply the annual mean of a derivative of PET with respect to a variable, by the annual-mean linear trend in that variable. That's because the derivative of PET may be much larger in, say, summer than in winter, causing the summertime trend in a variable to matter much more than its wintertime trend. The underlying mathematical issue is that annualmean(a*b) does not equal annualmean(a)*annualmean(b), because multiplication is not linear. To give a simple numerical example, suppose the derivative of PET w.r.t. X is 1 in winter and 3 in summer (annual mean of 2). And suppose the trend in X is +8 in winter and -6 in summer (annual mean of +1). In your method, the attributed PET trend would be 2*(+1) = +2, reflecting the sign in the annual-mean trend of X. But actually the attributed PET trend is 1*(+8) = +8 in winter but 3*(-6) = -18 in summer, so it is -5 in the annual mean, dominated by the summer trend in X. So both the sign and magnitude end up very incorrect here.

So you need to somehow do the multiplications in eq. 16 *before* you take annual means. Most straightforward would be to take just seasonal means of the PET derivatives rather than annual means (and use your linear regressions of the *seasonal* means of the variables for the rates of change), compute each product in eq. 16 for each season, and then average each product over the four seasons. This is not perfect, since the derivatives will still vary even within a season - but the variation would be much less than the variation over the whole annual cycle and you will have quashed the main potential source of error. You may get very different results after doing this.

All of the above also applies to aerodynamic PET (EPA) and radiative PET (EPR), of course. In particular, I bet this is why you're getting such huge differences between the actual EPA trends (leftmost symbols) and the sum-of-terms estimates (second from left) in the right-hand panel of Fig. 11 - you can particularly see this for England and its lowlands. [And same for PET trends, since EPA+EPR=PET.] The actual EPA trends show that the key statements at p15 li19 and p15 li28 (and thus perhaps

p16 li9,10,13,18,etc) aren't so solid - e.g. for the English lowlands the actual EPA and EPR trends look practically equal! This is also clear in Fig 10. So you need to change these sentences.

[In fact, all of this nonlinearity stuff also applies *spatially*, not just temporally, especially if there are large spatial contrasts in the PET derivatives like the large seasonal contrasts (you should check whether there are.) So, strictly, it's best to compute the products in eq. 16 separately at each 1x1km point, rather than plugging in regional means. This would require finding linear trends of the seasonal-mean input variables at each 1x1 point, but is probably worth it, and will probably improve the attributions even further.]

Because of this issue, I also think you should give percentages of the *actual* trend in Table 3, not percentages of the total attributed trend. This will both more honestly reflect any shortcomings of the attribution (e.g. when the percentages don't add to 100%, it will mean the attribution wasn't completely successful) and will shrink the oddly huge numbers in Table 3c, which stem from the very small magnitude of the total attributed EPA trend. (Of course, both of these problems may greatly diminish anyhow once you do the annual and spatial averaging in eq. 16 correctly, as described above - so this may not end up mattering so much.)

If in fact the total attributed EPA trend is *still* an order of magnitude less than the actual calculated EPA trend in England, English lowlands and Britain in Fig 11 even after fixing the averaging procedure as described above, then I would guess there is some error in the implementation of the EPA parts of eqs A1-A5, or even a units issue somewhere. It doesn't make sense for the magnitude discrepancy to be this large.

***General: It would be very good to leverage the full resolution of this dataset to make *maps* of the PET trends, not just trends in the geographic mean PET over very large regions as in Fig 8, 10, 11. Perhaps you could include a 5-panel figure showing maps of the trends in annual-mean PET, spring PET, summer PET, fall PET, and winter PET.
And/or a 4-panel figure showing maps of the trend in annual-mean PET, annual-mean PETI, annual-mean radiative PET, and annual-mean aerodynamic PET. (These are just examples; the point is to include trend maps of some sort.)

***Section 2.2: A couple of things seem wrong with this method as stated - first, p* should not be 100,000 Pa, but should be the local surface pressure discussed in section 2.8, which is much lower in high-elevation regions as you can see in Figure 1. (This also applies throughout section 3 where p* is used in the calculation of Penman-Monteith PET.) Second, the units on the vapor pressure lapse rate should be hPa/100m = Pa/m as stated in Hough and Jones (1997), not %/m as you write. I'm not sure whether this is a typo. (Or perhaps Hough and Jones are incorrect, and you are correcting them here?) In any case you should clarify your intent - it's not clear whether you lapsed by 0.025 %/m or 0.025 Pa/m.

More generally, is a vapor pressure lapse rate assumption the best way to extrapolate near-surface humidity in elevation in Britain? Since most parts are well within the boundary layer, and are not interacting with the free troposphere much, I would think qa (plus condensate) in Britain would be fairly well-mixed vertically at a location. So the most reasonable procedure would be to first turn each MORECS vapor pressure into a qa using eq.(1) with the local surface pressure, then horizontally interpolate the qa to 1km resolution using your bicubic spline method, and leave it at that (i.e. no elevation correction.) You would then have to apply a switch to eq.(2) to specify that "qs-qa" should be treated as zero if/when it goes nominally negative (this corresponds to higher elevation, low-qs locations that are above the LCL, or within the cloud layer. In those cases, most of the excess of "qa" above qs would go into cloud condensate.)

[In contrast I think the vapor pressure lapse rate idea comes from thinking about the free troposphere, where qa can more easily have vertical contrasts, and where RH (not qa) is the more slowly varying, vertically-conserved quantity.]

But if you have a good reason to use a vapor pressure lapse rate rather than a wellmixed qa framework (e.g. some earlier literature demonstrating that the former works well for surface conditions in Britain), you should say so. I am not an expert on this issue.

Minor comments:

page 5 lines 10-11: You write "The WFD and CRU TS 3.21 datasets were used where variables could not be calculated solely from MORECS", but the reader is left wondering where/when specifically was this true? What parts of the study used WFD and CRU derived data instead of MORECS, and for what variables? As written, the reader has no idea what to take away from this phrase. So you should be more specific, e.g. "The WFD and CRU TS 3.21 datasets were used for surface air pressure and DTR, since those two variables were not in MORECS" or similar.

p7 li17-19: Does this ever result in negative wind speed, if the topographic correction is negative and is added on a day when the wind speed is much weaker than the time-mean wind speed? If so, this is unphysical - you may instead want to make the topographic correction a ratio rather than a difference (i.e. compute ETSU(1km)/ETSU(40km) instead of ETSU(1km)-ETSU(40km)). Then the interpolated daily wind speed would be multiplied by this ratio. That way, negative values are not possible.

Section 2.7: Any reason why interpolation is not done for DTR, unlike all the other variables? If there is a reason, you should state it in the text.

p8 li9-10: Can you provide a bit more detail on what "lapse the air pressure from the WFD elevation to the 1 km resolution elevation using the temperature lapse rate" means? I don't ordinarily think of the temperature lapse rate as applying to pressure - instead, pressure behaves in the vertical according to the hydrostatic or hypsometric equation, $dp = -density*g*dz$ or equivalently $dlnp = -g/(rT)*dz$. Is the Shuttleworth (2012) method some sort of fancy integral of this differential equation which takes into account the variation in T with height?

Fig 1: Why does the wind speed scale extend all the way to 10 m/s when there seem to be few or no points stronger than ∼5 m/s (or weaker than ∼3 m/s)? It's hard to see any spatial wind speed contrasts with the current scale, it all just looks green. In particular I can't see the positive correlation between elevation and wind speed (p8 li23) at all with this scale.

Fig 4: Since you discuss the difference in terms of % in the text rather than absolute millimeters per day, it may be better to plot the % difference in the last panel instead of the absolute difference.

Section 3-4 in general: Where is the DTR used?? Why was DTR derived and explained in section 2, if it has no role in the calculation or in the rest of the text at all? Yes, it is included in the figures and tables, but it's never discussed or mentioned in the text after section 2.

Table 2 and Section 4 vs Figs 9-11: The table and body text report trends per decade, but the figures report trends per year. This should be made consistent - pick one or the other.

Or, alternatively, you could just report per-51yr trends over the whole time period (like you helpfully provide at the end of p13 and beginning of p14) in all cases in the table, figures and body text, and skip the per year and per decade numbers completely! This would be perhaps easiest to interpret. The units would then be given as "mm d-1 (51 yr)-1" throughout the tables and figures.

p13 li21-23: I assume "long term" here means going back much further than 1961, since I certainly don't see "drying summers" in Fig 9's precipitation plot. It may be good to replace "long term" with a more specific (but still could be rough) time period, and to explicitly point out that summers don't have declining precip post 1961, so the reader is not confused. (Unless "drying" here refers to declines in something like P/PET or P-ET, not P.)

[Figure]

Figure 11 caption: You need to more clearly state that the leftmost symbols are the regular computed trends from Fig 10, while the second-to-leftmost symbols are the sums of all the pieces attributed to the different variables. I understood this but it should be explicit. See the major comment above.

Technical corrections and/or typos:

Eqs 2, 9, 10: Should be 1 + rs/ra in the denominator, not 1 - rs/ra. (Is this merely a typo, or is the implementation wrong in your product as well? I would assume not, but just wanted to check.)

p9 li9: Should be qa, not q (to be consistent with eq 2 and rest of paper.)

p9 li20 (eq 4): Last factor should be raised to the power i-1, not i. (Again, is this just a typo or also a code error?)

p10 li21-22: And also wind speed, of course. (That makes six.)

p14 end of li23: Should be PET, not P.

---

## Referee Comment (RC3) · Anonymous Referee #3 · 1 Mar 2016

The authors develop a high resolution data set of required meteorological grids for Great Britain (GB) and then use these to drive potential evapotranspiration (ETp) formulation, specifically Penman-Monteith (ETp_PM) to understand the drivers of changes of ETp_PM. While this manuscript (MS) has some potential to make a very solid contribution to the international understanding of drivers of large-area trends of atmospheric evaporative demand (AED – note I would use this phrase in the title), currently it does not reach that potential. The topic of the MS is very well suited for publication in HESS, yet the current MS needs much improving, via a total major overhauling revision, to ensure that the resultant MS meets the standards required for publication in HESS. I would like to see the following issues resolved / considered thus allowing me to recommend acceptance to HESS; I hope that the authors rise to the challenge. Ultimately this has the potential to be a very good and very influential HESS paper.

[Figure]

1) Major: while using ETp_PM to parameterise some of the resistance terms the authors have incorrectly equated Allen's FAO-56 crop reference evapotranspiration (ETo) as a form of ETp. They are different concepts, and cannot be equated. Nowhere in (Allen et al., 1998) does it suggest that ETo replaces estimates of ETp. After downloading the FAO-56 report from http://www.fao.org/docrep/X0490E/x0490e00.htm please searched for the term 'potential evaporation' and it is only found twice in the body text and in these two instances the authors are not equating crop reference evaporation with potential evaporation. Additionally in Chapter 1 of (Allen et al., 1998) they state (on page 30 of the PDF file) "The use of other denominations such as potential ET is strongly discouraged due to ambiguities in their definitions." This can be found by searching for the word 'potential' in the FAO-56 report. This illustrates basic conceptual misunderstandings.

To achieve the ultimate goal of understanding GB AED trends, there are a number of ways to improve this: (a) parameterise the ETp_PM resistances as function moisture availability; (b) use the Penman formulation of ETp (denoted ETp_P herein).

2) Major: the current use of these FAO-56 derived prescribed land surface conditions implicitly implies that the land-cover for all of GB from 1961-2012 was covered by 'A hypothetical reference crop with an assumed crop height of 0.12 m, a fixed surface resistance of 70 s m–1 and an albedo of 0.23. The reference surface closely resembles an extensive surface of green grass of uniform height, actively growing, completely shading the ground and with adequate water. The requirements that the grass surface should be extensive and uniform result from the assumption that all fluxes are one dimensional upwards' (quoted from Allen et al. FAO-56 report, 1998, p 23). When looking at time series remote sensing of albedo (e.g., AVHRR GIMMS 3g) and Lidar imagery of vegetation height (Simard et al 2011) and any GB land-cover map (e.g., http://www.ceh.ac.uk/services/land-cover-map-2007) it is obvious these assumptions are not scientifically warranted. If the authors continue to use ETp_PM then I would suggest that they parameterise the resistances in more dynamic and appropriate fashion, or else they could use the ETp_P formulation.

Simard M, Pinto N, Fisher JB and A. B (2011) Mapping forest canopy height globally with spaceborne lidar. Journal of Geophysical Research 116(G04021), doi:10.1029/2011JG001708.

2a) Why not a 2014 or 2015 end date in the time series?

3) Major: what is the justification for ignoring changes in albedo? The land-cover induced changes in albedo are likely important for dynamics of AED in GB and currently these are ignored. I suggest you use AVHRR estimates of albedo which are available from 1981 to ensure that you provide realistic estimates to the calculation of net radiation. This may mean your study covers from 1981 to 2015.

4) Major: in the form of cross-plots please compare observed annual trends of the main meteorological parameters from a series of at least 50 stations across GB with the annual trends exacted from the derived grids. This would be a new figure and would provide quantitative confidence to the reader that the input grids are replicating / capturing trends observed at meteorological stations. Without such cross-plots the reader does not know how accurate your input grids are.

5) Major: to assess the accuracy of the grids you could calculate the PenPan model (Rotstayn et al 2006; Roderick et al 2007) and assess compare the observed trends with modelled PenPan trends. This would allow you to independently validate the trends of AED (which is estimated by ETp, ETo or measured by pan evaporation (Epan) rates) and would be a very strong contribution to your MS.

Roderick, M.L., Rotstayn, L.D., Farquhar, G.D., Hobbins, M.T., 2007. On the attribution of changing pan evaporation. Geophys. Res. Lett. 34, L17403. doi:10.1029/2007GL031166.

Rotstayn, L.D., Roderick, M.L., Farquhar, G.D., 2006. A simple pan-evaporation model for analysis of climate simulations: evaluation over Australia. Geophys. Res. Lett. 33,

L17715. doi:10.1029/2006GL027114.

6) Page (P) 3/Line (L) 2-9, why the strong UK / ecology focus, HESS is an international hydrology and earth system science journal not a regional hydrology and ecology journal. Improve the opening motivating comments by making them more attractive to HESS readers.

7) P3/L10, gridded meteorological datasets are not the only physical drivers of this, for example there have been changes in land-cover due to increased urbanisation in the UK since 1961 and this is also a physical driver.

8) P4/L22, what are the specific objectives of your study? That is in the last paragraph of your Introduction can you please explicitly state what your 'aim(s)' or 'objective(s)' or 'hypothesis (hypotheses)' is (are)? That is, specifically use one of these words. While you "present the method" (P4/L22), this is a little broader than having specific aims or objectives. Consider using a bulleted sentence structure to list these. Note the word 'question' is used in the following to generically mean aim / objective / hypothesis. Note the grammar of such a sentence follows (please pay careful attention to the use of colons, semi-colons and capital letters): (i) question 1 is interesting; (ii) question 2 is really interesting; and (iii) my Mum thought I should write something about question 3.

9) Improved structure: once you've explicitly used one the following words to state what your 'aim(s)' or 'objective(s)' or 'hypothesis (hypotheses)' is (are), then, assuming you have objectives, use these objectives to provide structure to your revised MS. For example, let's assume you have three objectives, then use them to structure your Methods section, Results section and Discussion sections, as follows. 1 Introduction 2 Study Site and Materials (have as many sub-headings as needed to introduce all the datasets used, their pre-processing – or maybe this needs to be 2 main headings, noting you might also need a "2 Theoretical Background" section too, in which case this would heading #3, and all others would increment by 1) 3 Methods 3.1 Objective 1 (4-8 words to summarise objective 1) 3.2 Objective 2 (4-8 words to summarise objective

2) 3.3 Objective 3 (and so on) 4 Results 4.1 Objective 1 (same words as 3.1) 4.2 Objective 2 (same words as 3.2 and so on) 4.3 Objective 3 5 Discussion 5.1 Objective 1 5.2 Objective 2 5.3 Objective 3 6 Conclusion

10) P4/L26-28. Some of these variables are likely daily extremes (such are air temperature to provide Tmax and Tmin), some could be daily averages (e.g., specific humidity) whereas others (Rs_in and Rl_in) are likely daily integrals. It would be best is you explicitly mentioned what the daily variable represents.

11) P5/L4, how do you define hours of bright sunshine? Can this be defined as Rs_in > XX W/^2? Plus also use 'precipitation' as opposed to 'rainfall' as the former is all encompassing.

12) P5/L18, what is 'too far'; please quantify this subjective phrase.

13) P5/L21, air temperature environmental lapse rates vary throughout the year (see McVicar et al, 2007, JoH, doi:10.1016/j.jhydrol.2007.02.018 and the references therein), and it would be best yours did too.

14) P6/L2, atmospheric pressure will vary with elevation and will also vary on a daily basis in GB and also on a longer-term basis in GB as a function of atmospheric stability / latitude (as you clearly show in Fig 1). This needs to be improved.

15) P6/L15, what where the empirical coefficients used?

16) P8/L27, why not calculate the daily extremes of Tair, which would be topographically corrected, and then calculate DTR = Tmax − Tmin?

17) Major: P7/L5-6, both references are to model Rl_in for clear-sky conditions. My understanding is that there are parts of GB for certain times of the year that are cloudy. This needs improvement.

18) P7/L16, you should be aware that the required topographic correction is temporally changing (McVicar et al 2010, GRL, doi:10.1029/2009GL042255).

[Figure]

19) P7/L23, what does 'natural neighbour interpolation' mean? This is first time I've ever seen these term, so you need to provide a citation to the algorithm or better explain what it means.

20) P9/L24, in a climate that is can be clear one day and cloudy the next day, or clear for a few days then cloudy for a few days following that, the ground (or soil) heat flux would be important. I'd prefer if you had a dynamic model for this, or at the very minimum, you need supporting citations that this is small component on cloudy days directly following several clear days in GB.

21) Major: P11/L26, yes but all of GB all year will not adhere the prescribed crop reference land-surface conditions used in FAO-56. They are very different concepts under the AED umbrella. I strongly suggest that you use AVHRR GIMMS 3g based estimates of LAI in the modelling (Zhu et al 2013).

Zhu, Z. C. et al. Global Data Sets of Vegetation Leaf Area Index (LAI)3g and Fraction of Photosynthetically Active Radiation (FPAR)3g Derived from Global Inventory Modeling and Mapping Studies (GIMMS) Normalized Difference Vegetation Index (NDVI3g) for the Period 1981 to 2011. Remote Sensing 5, 927–948, doi: 10.3390/rs5020927 (2013).

22) P11/L27, some recent papers have made head-way into the interception issue, and you may find them of value.

F.L. Pereira, F. Valente, J.S. David, N. Jackson, F. Minunno and J.H. Gash (2016) Rainfall interception modelling: Is the wet bulb approach adequate to estimate mean evaporation rate from wet/saturated canopies in all forest types? Journal of Hydrology 534 (2016) 606–615, http://dx.doi.org/10.1016/j.jhydrol.2016.01.035

van Dijk, A.I.J.M., Gash, J.H., van Gorsel, E., Blanken, P.D., Cescatti, A., Emmel, C., Gielen, B., Harman, I.N., Kiely, G., Merbold, L., Montagnani, L., Moors, E., Sottocornola, M., Varlagin, A., Williams, C.A., Wohlfahrt, G., 2015. Rainfall interception and the coupled surface water and energy balance. Agric. For. Meteorol. 214–215,

402–415. http://dx.doi.org/10.1016/j.agrformet.2015.09.006.

Zhang, Y., Peña-Arancibia, J.L., McVicar, T.R., Chiew, F.H.S., Vaze, J., Liu, C., Lu, X., Zheng, H., Wang, Y., Liu, Y.Y., Miralles, D.G., Pan, M., 2016. Multi-decadal trends in global terrestrial evapotranspiration and its components. Sci. Rep. 6, 19124. http://dx.doi.org/10.1038/srep19124.

23) P12/L26, the aridity index has been around for a long time, I suggest more historic references could be used here, see Oldekop (1911) whose scientific contribution is the basis for the following.

Vazken Andréassian, Ülo Mander, Taavi Pae (2016) The Budyko hypothesis before Budyko: The hydrological legacy of Evald Oldekop. Journal of Hydrology 535 (2016) 386–391, http://dx.doi.org/10.1016/j.jhydrol.2016.02.002

24) P13/L3, these areas are considered 'equitant', those that straddle the water-limited and energy-limited regimes, see McVicar et al 2012, Ecohydrology, doi:10.1002/eco.1298).

25) P23/L23-25, you should mention that these findings are using reanalysis output as input, and that these outputs have been shown to have limited capacity to capture trends in a key aerodynamic variable, wind speed, and that this limitation has been documented in the both northern (Pryor et al., 2009, JGR-Atm, doi:10.1029/2008JD011416) and southern (McVicar et al , 2008, GRL, doi:10.1029/2008GL035627) hemispheres.

26) P16/L12-15, these is certainly very interesting ecological insights, yet given that HESS is an international hydrology and earth system science journal not a regional hydrology and ecology journal I question the value to the HESS readership.

27) I would like a very short Conclusion section (note singular like your Introduction section, with both containing multiple ideas) added.

28) Table 1, for the radiation components while in essence they are assumed to be

representative of being at the surface (i.e., 0 m height) they are usually observed at 1.2 m (or thereabouts) above the land-surface.

29) Figure 1, place parts (a) to (h) on sub-plots and then use these in the caption and in the text. For wind it's all uniform, can you reduce the range shown to have more colours on the sub-plot. In Scotland there is an area with low specific humidity, yet some of this area has very high daily precipitation; is this correct? What is happening here?

30) Some of the maps could be made larger to fill both columns of the HESS published page.

31) Figure 7, it would be useful to provide the equation for the GB linear regression. You wish to consider providing the other regional linear regression statistics in the Supplementary Material; these are hard number that other may cite. Also for Figure 8

32) Figure 11, what to the left-most bars in each of the sub-plot indicate? I've read the caption four times and don't get it, I've asked a colleague and they read it 3 times and did not get it. I mean what is the non-coloured bar that is the left-most on each sub-plot mean? In the top-left sub-plot, you have two bars for ETp_PM for England, the left-most is only black and the one directly next to it has pink background for part of it; what is the difference between these two.

---

## Author Comment (AC1) · 15 Apr 2016

We thank Dr Weedon for his comments, and intend to implement changes to address all of them.

1) We will add the missing variable (wind speed).

2) Yes, the effect of lying snow will be different from intercepted water. However, as Dr Weedon states, for much of Great Britain snow days are rare. For regions where there is lying snow, this occurs during times when the evaporative demand is low, as the atmosphere is cold and wet. Therefore the difference of accounting for canopy snow as distinct from canopy water would be small. However, we do agree that it would be useful to mention this in the manuscript, and will do so.

[Figure]

3) We agree with the different wording and will change it accordingly.

4) These have been calculated, and will be added to the text.

5) These will also be added to the text.

———————————————

---

## Author Comment (AC2) · 15 Apr 2016

We thank the reviewer for their comments and will address them as follows.

1) Uncertainties of the CEH-GEAR precipitation (Keller et al., 2015), the CRU TS 3.21 DTR (Harris et al., 2014) and the WFD surface air pressure (Weedon et al., 2011) are discussed in the original papers. For the other variables, we are aware of other projects in progress to more thoroughly evaluate spatial fields generated from station data, but there are no results yet available. Our own work has involved some assessment of the data against flux sites. We will consider including examples of this to demonstrate uncertainties.

2) Note that there are plants which do have stomata on the upper sides of leaves as well as below (Camargo and Marenco, 2011). However, it is true that in the temperate

climate of the UK plants are more likely to have stomata only on the underside of the leaves. This does not, however, preclude the inhibition of transpiration by intercepted water.

Suppression of transpiration is well observed both by comparing eddy-covariance fluxes and observations of sap flow (Kume et al., 2006; Moors, 2012), and by observing stomatal and photosynthesis response to wetting (Ishibashi and Terashima, 1995). The suppression may simply be due to the fact that energy is used in evaporating the intercepted water, so less is available for transpiration (Bosveld and Bouten, 2003). It may also be due to water directly blocking the stomata, even if they are open (most likely to affect leaves with stomata on the upper side), or due to the presence of water causing stomatal closure. This latter mechanism can be observed even when the stomata are on the underside of a leaf and the water is lying on the upper side (Ishibashi and Terashima, 1995). It is also possible that the increased humidity of the air, due to evaporation of intercepted water, causes the stomata to close.

We will add some more detail about this to the manuscript, to make it clear that it is not simply that the intercepted water is directly blocking the stomata.

References: Bosveld, F. C., and Bouten, W.: Evaluating a Model of Evaporation and Transpiration with Observations in a Partially Wet Douglas-Fir Forest, Boundary-Layer Meteorology, 108, 365-396, 10.1023/a:1024148707239, 2003. Camargo, M. A. B., and Marenco, R. A.: Density, size and distribution of stomata in 35 rainforest tree species in Central Amazonia, Acta Amazonica, 41, 205-212, 2011. Harris, I., Jones, P. D., Osborn, T. J., and Lister, D. H.: Updated high-resolution grids of monthly climatic observations - the CRU TS3.10 Dataset, International Journal of Climatology, 34, 623-642, doi:10.1002/Joc.3711, 2014. Ishibashi, M., and Terashima, I.: Effects of continuous leaf wetness on photosynthesis: adverse aspects of rainfall, Plant, Cell & Environment, 18, 431-438, 10.1111/j.1365-3040.1995.tb00377.x, 1995. Keller, V. D. J., Tanguy, M., Prosdocimi, I., Terry, J. A., Hitt, O., Cole, S. J., Fry, M., Morris, D. G., and Dixon, H.: CEH-GEAR: 1 km resolution daily and monthly areal rainfall estimates

for the UK for hydrological and other applications, Earth Syst. Sci. Data, 7, 143-155, doi:10.5194/essd-7-143-2015, 2015. Kume, T., Kuraji, K., Yoshifuji, N., Morooka, T., Sawano, S., Chong, L., and Suzuki, M.: Estimation of canopy drying time after rainfall using sap flow measurements in an emergent tree in a lowland mixed-dipterocarp forest in Sarawak, Malaysia, Hydrological Processes, 20, 565-578, 10.1002/hyp.5924, 2006. Moors, E.: Water Use of Forests in the Netherlands, PhD, Vrije Universiteit, Amsterdam, the Netherlands, 2012. Weedon, G. P., Gomes, S., Viterbo, P., Shuttleworth, W. J., Blyth, E., Osterle, H., Adam, J. C., Bellouin, N., Boucher, O., and Best, M.: Creation of the WATCH Forcing Data and Its Use to Assess Global and Regional Reference Crop Evaporation over Land during the Twentieth Century, Journal of Hydrometeorology, 12, 823-848, doi:10.1175/2011jhm1369.1, 2011.

---

## Author Comment (AC3) · 15 Apr 2016

We thank the reviewer for their detailed reading of the paper and for their comments and suggestions. We will address them as follows.

Major comments

- P15 L10-11. This is a well-established method (see for example (Donohue et al., 2010; McVicar et al., 2012)). However, we appreciate the reviewer's concerns, and so have investigated their suggestions. This gives the same results to within a few percent, so we suggest we continue with the original method.

We will re-check the code for errors, but the difference between the fitted trends and the trend calculated as a sum of the contributions of the other variables are consistent

to within the quoted uncertainties.

- Table 3: The suggestion of giving percentages of the actual trend is good, and we will do this.

- Trend maps: Rather than present maps of trends, we have carried out the analysis aggregated over regions, as this is commensurate with the level of detail and uncertainties in the data. We prefer to keep the analysis of regions, as this provides a summary of the same information that would be seen in the maps.

- For historical reasons, the code used to create the specific humidity uses a constant air pressure of 100kPa rather than air pressure from Sect 2.8. The difference it makes to the air pressure has been checked for a subset of the data and was found to be small (of the order of a few percent), particularly in lowland areas where the air pressure is close to 100kPa. For any future updates of the data we will revise the procedure to use the varying air pressure.

The vapour pressure lapse rate used was indeed %/m, which is quoted in the earlier MORECS reference (Thompson et al., 1981). After some discussion with the MetOffice, we have determined that the units in Hough and Jones (1997) should actually be hPa/hPa/100m (ie, a fraction, rather than a percentage). We will cite the earlier document, to remove confusion.

- The vapour pressure lapse rate is implemented in order to keep relative humidity constant with altitude, rather than assuming well-mixed specific humidity. We are not certain as to the best method with which to adjust humidity as the alternatives all involve assumptions that we do not have data to test, nor have we found much literature on the subject. Given this uncertainty we have preferred to follow the method used by the Met Office in calculating the MORECS data. We will continue to investigate this, and the impact on evaporative demand and other model calculations, and are open to adopting another procedure in any future versions of the product.

Minor comments

- P5 L10-11. We will re-write this to clarify

- P7 L17-19. We have accounted for the possibility that this produces negative wind speeds and will add something to this effect in the manuscript.

- Section 2.7. The DTR data (from CRU TS) are only available at coarser temporal and spatial resolution than the MORECS variables and there is large uncertainty in how DTR varies with altitude across the landscape. In view of these uncertainties we decided not to interpolate DTR and will explain this in the text.

- P8 L9-10. This is indeed a method which integrates the hydrostatic equation, taking into account the variation of T with altitude. This will be more clearly explained in the text

- Fig 1. We will improve the choice of limits on these maps

- Fig 4. We think that the absolute difference will be more interesting (and relevant) to people who may use the PET(I) products, therefore we would prefer to keep the figure as it is, and add absolute differences to the text.

- Section 3-4. DTR is required for running a sub-daily land surface model with daily inputs, but is not required for the PET(I). We will clarify this.

- Trends per year vs. trends per decade: We will make these consistent through the text

- P13 L21-23: Yes, the evidence for drying summers is over a much longer time period. The report (Jenkins et al., 2008) on which we base this statement states that summers have been drying (decreasing precipitation) over the last 250 years, but it is not observable over the past 50 years (ie the duration of our dataset). We will clarify this in the text.

- Figure 11 caption: Yes, we will alter the caption to better describe the plot.

Technical corrections

- Eqs 2, 9, 10: This is just a typo in the manuscript, not in the code, and will be corrected

- P9 L9: Yes, this will be changed to qa

- P9 L20: Again, a typo in the manuscript, which will be corrected. The code is unaffected.

- P10 L21-22: Yes, wind speed will be added

- P14 L23: This will be changed from P to PET.

References

Donohue, R. J., McVicar, T. R., and Roderick, M. L.: Assessing the ability of potential evaporation formulations to capture the dynamics in evaporative demand within a changing climate, Journal of Hydrology, 386, 186-197, doi:10.1016/j.jhydrol.2010.03.020, 2010.

Hough, M. N., and Jones, R. J. A.: The United Kingdom Meteorological Office rainfall and evaporation calculation system: MORECS version 2.0-an overview, Hydrology and Earth System Sciences, 1, 227-239, doi:10.5194/hess-1-227-1997, 1997.

Jenkins, G. J., Perry, M. C., and Prior, M. J.: The climate of the United Kingdom and recent trends, Met Office Hadley Centre, Exeter, UK, 2008.

McVicar, T. R., Roderick, M. L., Donohue, R. J., Li, L. T., Van Niel, T. G., Thomas, A., Grieser, J., Jhajharia, D., Himri, Y., Mahowald, N. M., Mescherskaya, A. V., Kruger, A. C., Rehman, S., and Dinpashoh, Y.: Global review and synthesis of trends in observed terrestrial near-surface wind speeds: Implications for evaporation, Journal of Hydrology, 416, 182-205, doi:10.1016/j.jhydrol.2011.10.024, 2012.

Thompson, N., Barrie, I. A., and Ayles, M.: The Meteorological Office rainfall and evaporation calculation system: MORECS, Meteorological Office, Bracknell, 1981.

---

## Author Comment (AC4) · 15 Apr 2016

The review starts by suggesting we change the title of the paper from 'evaporative demand' to 'atmospheric evaporative demand' and suggests the paper needs a 'total major overhauling revision'. This is not justified by the scientific criticisms of the paper and we therefore intend to keep the structure broadly as it is. While we will implement some changes as suggested in this review, many of the comments are merely a different definition or presentation of the material and we believe some are unnecessary. We will address the comments and suggestions individually as follows.

1) Point 1 argues that there is a difference between Reference Crop Evaporation and Potential Evaporation – of course there is if you define it to be so. Potential Evaporation is a concept, not a real thing, and depends on the definition. We have been very careful

to define which Potential Evaporation we mean; calculated using Penman-Monteith for a reference crop surface. This is not a scientific issue, just an issue of definition.

It is also suggested we need to recalculate the potential evaporation accommodating soil moisture status – however, this would no longer be an estimate of evaporative demand, rather it would be an estimate of actual evaporation, which is not in the scope of this study.

2) The aim of this study is to quantify the change in evaporative demand, i.e. a function of the meteorology, not land use change or vegetation response. Therefore we use a constant standard reference surface and investigate only the effect of changing meteorology.

The use of the Penman equation (point 2) would be inappropriate as an estimate of evaporative demand as it does not include the wind-humidity deficit demand.

In response to point 2a, the input data were only available to 2012, therefore we were unable to extend this dataset to more recent years. We intend to extend the dataset to 2015 when all of the inputs are available.

3) Again, we are investigating the effect of changing meteorology, not land use/albedo change. This would be an interesting study in itself, but is outside of the scope of this paper.

4) The data are ultimately derived from station data, as is made clear in the paper. So it is not very helpful to compare the data with the station data used as input – they are similar by design.

5) It is not clear how calculating PenPan evaporation would help to 'assess the accuracy of the grids'. Again, PenPan is a different estimate of evaporative demand, and is not necessary as a validation of the PM potential evaporation that we are using.

6) Ecology is part of the Earth System, and this is a regional study (many of which have appeared in HESS), so we see no problem with these references which already

cover both hydrology and ecology (i.e. not just ecology). Readers will come with some knowledge of the larger context (that the UK is not alone in experiencing these changes) and we do not feel that more international references will help to introduce this regional study.

7) We will change 'physical drivers' to 'climate drivers', as this study does not include the effects of land use change etc.

8) Points 8 and 9 suggest a restructuring of the paper based on 'Objectives'. We believe that we have structured the paper in a relevant and readable way (note that referee #1 and Dr Weedon have both commented that the paper is well structured). We have made some changes, but do not think that it is necessary to structure the paper in the way that is suggested here.

9) See point 8.

10) Yes, we will rewrite this to make it clearer.

11) Hours of bright sunshine definition will be included. We will change rainfall to precipitation throughout.

12) We will clarify this.

13) This is a standard lapse rate (specifically used by the MetOffice in calculating MORECS), which we have applied here. For future studies it may be a useful extension to consider temporally varying lapse rates, but it was not deemed necessary here.

14) For historical reasons, the code used to create the specific humidity uses a constant air pressure of 100kPa rather than air pressure from Sect 2.8. The difference it makes to the air pressure has been checked for a subset of the data and was found to be small (of the order of a few percent), particularly in lowland areas where the air pressure is close to 100kPa. For any future updates of the data we will revise the procedure to use the varying air pressure.

15) These coefficients vary spatially and by month. They are available from the Met Office, and are reproduced in the originating paper (Cowley, 1978).

16) We would like to have been able to use the daily extremes but MORECS only provides daily mean air temperature, not the minimum and maximum, so we are unable to do this.

17) Indeed, the two models mentioned (Dilley and O'Brien, 1998; Prata, 1996) were used for the clear sky component, and we have calculated the longwave radiation component from cloud cover following Kimball et al. (1982). The subsequent sentence (P7 L6-8) states this.

18) This is an interesting point, but since we do not have any information about the spatial and temporal variation of lapse rates in GB, we use the standard one.

19) Reference will be added.

20) The soil heat flux is generally negligible over time periods of a few days, as stated in Allen et al. (1998):

"As the magnitude of the day or ten-day soil heat flux beneath the grass reference surface is relatively small, it may be ignored and thus:

$G_{day} \approx 0$ (42)"

21) Again, we are interested in this particular definition of potential evapotranspiration, and are not investigating land use or land use change.

22) We are not sure what the intention of this comment is, but thank you for the references.

23) We will update this with a more appropriate reference.

24) This is again interesting, but we are not sure what this comment is asking for.

25) Yes, we will mention this.

26) As noted previously, this is clearly a regional study (within the scope of HESS) and these references are appropriate. Readers will understand that similar effects (e.g. on biodiversity) could occur in other regions too. (See also point 4 above.)

27) We suggest that the final paragraph of the current "Discussion" section functions as a conclusion, so we will add the "Conclusion" title here.

28) True. We will add this to the text (although the negligible divergence of radiative fluxes between the surface and 1.2m means there is little impact of the information).

29) We will use a) to h) and refer to these in the caption/text. We will also alter the scales appropriately.

Areas with high precipitation but low specific humidity - yes, this is correct. These are higher, colder regions, with low saturated specific humidity. Even with high relative humidity the specific humidity remains low.

30) We will consider this.

31) Yes, we will provide these numbers.

32) The left-hand error bars with symbols are the slopes obtained from the linear regression. We have not explained this sufficiently well, so will rewrite the caption.

References

Allen, R. G., Pereira, L. S., Raes, D., and Smith, M.: Crop evapotranspiration - Guidelines for computing crop water requirements, Food and Agriculture Organization of the United Nations, Rome, Italy, FAO Irrigation and Drainage Paper, 1998.

Cowley, J. P.: The distribution over Great Britain of global solar irradiation on a horizontal surface, Meteorological Magazine, 107, 357-372, 1978.

Dilley, A. C., and O'Brien, D. M.: Estimating downward clear sky long-wave irradiance at the surface from screen temperature and precipitable water, Quarterly Journal of the

[Figure]

Royal Meteorological Society, 124, 1391-1401, doi:10.1256/Smsqj.54902, 1998.

Kimball, B. A., Idso, S. B., and Aase, J. K.: A Model of Thermal-Radiation from Partly Cloudy and Overcast Skies, Water Resources Research, 18, 931-936, doi:10.1029/Wr018i004p00931, 1982.

Prata, A. J.: A new long-wave formula for estimating downward clear-sky radiation at the surface, Quarterly Journal of the Royal Meteorological Society, 122, 1127-1151, doi:10.1002/qj.49712253306, 1996.

---

## Author Response (AR1)

Dear Dr Schymanski and anonymous reviewers,

Thank you for the opportunity to revise our manuscript, and for your helpful and insightful comments. We have submitted a revised version, which has taken into account all of the comments and we hope presents a more robust piece of work.

As has already been communicated to the editor, in the course of addressing Reviewer 2's points about whether it was reasonable to carry out the attribution analysis at the regional and annual scale and why the fitted and attributed trends in E_PA were so different, we discovered that there was indeed a bug in the code that calculated the attribution. We thank you for your patience while we addressed the revised results. The bug had the effect of artificially suppressing the contribution of the temperature, humidity and wind speed to the rate of change of PET. This was responsible for the marked difference between the attributed and the fitted trend, particularly for the aerodynamic component. Now that the bug has been fixed, the trends are consistent between the linear regression and the sum-of-components. We have edited the table of percentage contributions as suggested by reviewer 2, to be percentages of the fitted trend to illustrate this.

The dominant contribution to the positive PET trend now comes from the air temperature, but this is largely cancelled out by the trend in humidity and windspeed. The radiation components are still relatively large, and the downward shortwave still has a large effect on the spring PET, so we have retained the discussion of aerosol and circulation effects.

We have submitted a revised manuscript, and a version with changes highlighted, as requested. We implemented the changes as described in our responses to the reviewers, and as requested by the editor. We address specific points from the editor's comments below. All other changes are as defined in our previous responses to the reviewers. Finally, this document also contains the marked-up version of the paper showing the changes. Note that line numbers below refer to the new submitted paper, not the marked-up version.

Regards,

Emma Robinson, on behalf of the authors.

**Response to editor**

1. We have added objectives at the end of the introduction, and return to them in the conclusion. We have also included discussion of other regional studies of PET and its trends and attribution in the introduction.

2. In Section 2.10 we have included some discussion of data validation that we have carried out with meteorological data from UK flux sites, which are independent of the synoptic stations from which the MORECS data are derived. Unfortunately these observations are not long enough to calculate robust trends (the longest is 10 years), but we have looked at the comparison of daily and monthly means with the appropriate squares from the gridded data and provided statistics to show the good agreement between the gridded data and the observations. We have also pointed the reader to discussions of uncertainty in the original data papers where appropriate.

   Note that none of the flux sites have pan evaporation data sets available, so this suggestion could not be implemented.

3.  We have retained the Penman-Monteith formulation, and retained the 'interception corrected' PETI as well. PM potential evaporation with an interception component is known and used by hydrologists, so we do still wish to discuss what difference it makes. We have improved our description of the interception component in Section 3.1 and hope that it makes it clear that the PETI assumes that a wet canopy has a potential evaporation with no stomatal/canopy resistance, a dry canopy has potential evaporation with stomatal resistance = 70 sm-1, and an intermediate canopy has potential evaporation which is a linear combination of the two, dependent on how wet the canopy is.

    We have further discussed the choice of using Penman-Monteith PET defined for a reference crop in Section 3, and we continue to use this rather than the Penman formulation because of its inclusion of the effect of vegetation.

    While we do agree that investigating the PM equation with different resistances would be interesting, it is outside of the scope of this paper.

    We have taken the advice to use the phrase "atmospheric evaporative demand" (or AED) throughout the text.

4.  Due to the suggestion from Reviewer 2 we have recalculated the attribution by calculating the results for each pixel, then calculated a weighted mean over each region. We have also continued with the analysis on the regional means, as this allows us to calculate more conservative confidence intervals (otherwise it is difficult to account for spatial correlation in the trend maps). We also discuss the product-of-means vs mean-of-products issue, and note that our seasonal and annual analyses ultimately give the same results.

5.  We have included both a table of absolute values of the contributions, and a table of the contributions as a percentage of the linear regression to PET (and the radiative and aerodynamic components).

6.  We agree that the description of Fig 11 (now Fig 13) was not adequate, but we also decided that the figure itself could be improved. We have rearranged the figure, and rewritten the legend, and hope that this serves to clarify the results.

    As mentioned above, in investigating the results, we realised that there was indeed a problem with the numerics (the code was artificially supressing the contributions of air temperature, humidity and wind speed relative to the contributions of the LW and SW radiation). We have fixed this and the results are more consistent with other regional and global studies.

7.  We have presented trend maps in the appendix as requested.

8.  The original test we carried out of constant vs varying air pressure in the specific humidity calculation was for the whole of the dataset, not just the uplands, and it makes only a few percent difference in all regions. Apologies that this was not clear before. We have included some discussion of this in the text.

9.  We have more clearly identified where CRU, WFD, GEAR and MORECS data were used.

10. We have added some of these points to the text.

11. The DTR is included in the meteorological data set as it is required for some models (particularly the JULES LSM), so we have clarified why we have included it, despite it not being used for the PET calculations.

    Because MORECS only provides daily mean air temperature, the DTR was obtained from the CRU TS 3.21 data. We have clarified this in the text (see point 9).

12. We have added relative differences to the PET maps (Fig. 6) and climatology plots (Fig. 7).

**Other changes**

1.  We have added a section about validation of the meteorological data (Section 2.10). This includes references for some variables, and description of validation carried out against meteorological observations from four UK flux sites.

2.  We have added some more detail about interception inhibiting transpiration to the manuscript (lines 362-376), to make it clear that it is not simply that the intercepted water is directly blocking the stomata.

3.  We have investigated the reviewer's suggestions about product-of-mean vs mean-of-products. This gives the same results to within a few percent, but have mentioned this in the manuscript (lines 538-548)

    As mentioned above, we found a bug in the code, and now recover the fitted trends more successfully.

4.  Table 3: We now give the contributions as percentages of the actual trend (although it is now Table 4, not table 3).

5.  Trend maps: We have included trend maps in Appendix B

6.  We have kept the constant 100kPa air pressure in the calculation of specific humidity, but have mentioned that this makes only a small difference (lines 182-184)

    We have included some discussion of the vapour pressure lapse rate and have changed the citation (lines 167-168)

7.  We have discussed the choice of vapour pressure lapse rate for adjusting the humidity for height (lines 170-175)

8.  Line 149-152. We have clarified which variables come from data sources other than MORECS

9.  Line 225-226. We have added a note that if the method does not allow negative wind speed

10. Line 237-239. We have explained why we have not interpolated DTR

11. Line 246-248: We have more clearly explained the adjustment of air pressure with elevation.

12. Fig 1. We have improved the choice of limits on the colour maps

13. Figs 6, 7. We have added relative difference to the PET maps and climatology plots.

14. Line 239-240. We have clarified the role of DTR in this dataset.

15. Trends per year vs. trends per decade: We have made these consistent through the text

16. Line 466-471: The evidence for drying summers is over a much longer time period than this dataset, we have clarified this in the text.

17. Figure 13 caption: We have altered the plot and the caption.

18. Eqs 2, 9, 10: We have corrected typos

19. q has been changed to $q_a$ where necessary throughout

20. A typo in Equation 4 has been fixed.

21. Line 352: Wind speed was added to the list

22. P has been changed to PET where necessary

23. Line 297-305: We have added more discussion of the particular choice of PET in this paper.

24. Line 334-347: We have added to the discussion of a constant standard reference surface.

25. Introduction: We have expanded the discussion of regional studies within a global context, and of previous studies looking at trends in PET and reference crop evaporation

26. Line 47: We changed 'physical drivers' to 'climate drivers', as this study does not include the effects of land use change etc.

27. We have added objectives to the introduction and added a conclusion section

28. Line 138-139: We have added more detail about the variables.

29. Line 186-187: Hours of bright sunshine definition has been included.

30. We will change rainfall to precipitation throughout.

31. Line 156-157: We have quantified where islands have been excluded.

32. Line 230: Reference to natural neighbour interpolation will be added.

33. Line 439: We have added references to Oldekop and Andréassian.

34. Line: 513-515: Added a discussion of Matsoukas results being based on reanalysis output

35. Table 1: Added reference height for radiation

36. Added letter labels to plots where necessary

37. Line 390-295: Added a discussion of snow

38. Line 450-452: Changed the sentence about allowing for non-zero lag-1 autocorrelation

39. Added 95% CIs on trends throughout

40. Line 614-615: Added trends for SW down

[revised manuscript text omitted]
⁻¹) | **0.20 (21 ± 0.07, 0.31) K decade⁻¹ 15** | **0.23 ± 0.14** | **0.17 ± 0.12** | **0.21 ± 0.15** | **0.25 ± 0.17** |
| Specific humidity (g kg⁻¹ dec⁻¹) | **0.046 (0.010, 0.082) g kg⁻¹ decade⁻¹ 0.049 ± 0.037** | **0.054 ± 0.04** | **0.040 ± 0.036** | **0.055 ± 0.037** | **0.053 ± 0.044** |
| Downward shortwaveSW radiation (W m⁻² dec⁻¹) | **1.0 ± 0.8** | **1.3 ± 1 (.0.3, 1.8) W m⁻² decade⁻¹** | 0.5 ± 0.6 | **1.1 ± 0.9** | **1.5 ± 1.0** |
| Downward longwaveLW radiation (W m⁻² dec⁻¹) | **0.50 ± 0.48** | 0.45 (± 0.01,0.91) W m⁻² decade⁻¹ 48 | **0.58 ± 0.48** | 0.50 ± 0.55 | 0.42 ± 0.48 |
| Wind speed -0.17 (-0.27, -0.08) Wind speed (m s⁻¹ decadedec⁻¹) | | **-0.18 ± 0.09** | **-0.16 ± 0.09** | **-0.20 ± 0.10** | **-0.25 ± 0.16** | **-0.13 ± 0.07** |
| Precipitation (mm d⁻¹ dec⁻¹) | **0.08 (± 0.02, 0.14) mm day⁻¹ decade⁻¹ 06** | 0.04 ± 0.06 | **0.14 ± 0.09** | 0.08 ± 0.09 | 0.03 ± 0.05 |

[revised manuscript text omitted]

---

## Referee Report (RR1)

Clarification of review suggestion for hess-2015-520, Trends in atmospheric evaporative demand in Great Britain using high-resolution meteorological data by E. L. Robinson et al.

Near the top of my review I again suggested changing the attribution procedure to use net-LW as the LW variable rather than downward longwave, so as to put all of the canceling temperature dependencies in LW radiation together and simplify the story as to which terms are important. After submitting the review, I realized that it may be much easier for you all to make this change **only** in section 4.4, and not in section 4.3.

The idea is exactly analogous to how you already implemented the RH attribution: keep section 4.3 as-is using your "official" variables from your derived product, and then let section 4.4 be the "alternative" analysis where you use RH instead of q and, **simultaneously**, Ln instead of Ld. That way you will have just two analyses in total… one original/naive and one using more fundamental LW **and** humidity variables together to simplify the story and get rid of the cancelling terms. That is, the number of figures and tables will not change: Fig 15, Table 5 and Table 6 will now just employ Ln instead of Ld (in addition to employing Rh instead of qa.) Fig 13, Table 3 and Table 4 will be left alone.

Though, I suppose if you wanted to bother making equivalents of Figs. 14 and B3 for Ln, then you would have extra figures after all. I don't know the space limitations for this journal, hopefully that would not be an issue.

In any case, the appendix C equations would then be changed as follows: there would be an extra (short) equation for dEp/dLn between C6 and C7, with the exact same right-hand side as C2. And, C7 would have the -4*eps*sigma*T^3/Rn term removed. C1-C5 would be unchanged, contrary to my original suggestion.

Again, this all should be thought of as optional but potentially very useful – it is a great study regardless!

---

## Author Response (AR2)

Dear Dr Schymanski and reviewers,

Thank you for taking the time to re-review our manuscript. Once again the comments were insightful and we have taken the time to fully respond. In particular, we have taken reviewer 2's suggestion of recasting the attribution in terms of relative humidity and have performed the analysis. We have added it to the manuscript, rather than replacing the specific humidity analysis, as we believe that including both helps to tell the whole story. We think this has greatly added to our analysis, and we would very much like to know the name of reviewer 2 (if they are agreeable) so that we can properly acknowledge them.

Please find below our responses to the editor and reviewers, followed by a marked up version of the manuscript showing the changes made. Note that, since we updated the references, Word has marked all of the citations as changes, even though most are the same as before.

Regards,

Emma Robinson on behalf of the authors
* * *
**Response to comments by editor**

**L18:** Yes, we have changed this.

**L22:** Yes, changed.

**L298-299:** We have added references for the Pen-Pan model and the Penman equation.

**L301:** We have changed this.

**L348:** It is specifically for the 0.12m reference crop, we have clarified this.

**L372-374:** Yes, this was the wrong way round – the increased humidity decreases the evaporative demand. We have amended this.

**L394:** Changed 'the number of lying snow days' to 'the number of days with snow cover'.

**Response to Reviewer 1 (Report 2)**

We thank the reviewer for their comment and respond as follows.

"Concerning the issue of reduced transpiration due to interception I have still one comment:

In line 487 – 490 authors argue, that due to evaporation of snow/rain at the upper side of the leaf, the transpiration is reduced by closed stomata due to high relative humidity. I would not agree with that statement. Given a high relative humidity in the surrounding air, the plant will open the stomata as it is able to exchange carbon for "free" – so the reason for no transpiration is the missing gradient in vapour partial pressure – I am sure that can be easily shown in FLUXNET data when looking at carbon fluxes after rain events!"

Yes, we agree that this sentence was wrong – we have changed it to mention the physical mechanism by which increased humidity decreases the evaporative demand.

**Response to Reviewer 2 (Report 1)**

We thank the reviewer for their detailed and insightful comments, in particular the suggestion to use relative humidity in the attribution. We have implemented the following changes in response.

**Major comment:**

This is an excellent suggestion, which we believe supplements the analysis with specific humidity to tell the story of the response of PET to changing climate. We find, as the reviewer expected, that the response to air temperature is reduced significantly. There is a negative trend in the relative humidity across our dataset, although this is only statistically significant in spring. This trend in humidity drives an increase in the PET of similar magnitude as the SW down (this is the same as the difference between the increase in PET due to air temperature and the PET decrease due to specific humidity in the first formulation). We have therefore added this to the manuscript in Section 4.4.

**Minor:**

**General:** Thompson et al (1981) is a Met Office technical report, which is, unfortunately, difficult to get hold of outside of the Met Office. However, this is the only place that the technical details of MORECS are available (since it is a commercial product). The Field (1983) paper does provide a good summary of the technical details however, so we have also added a reference to it.

**General (but optional):** We have kept the units of dec-1, largely for intercomparison with other studies.

**21-22:** We have edited the abstract to be more consistent.

**54:** We have added that the Princeton forcings are available at 0.25 deg.

**85-91 vs 91-96:** Yes, we've added some text to make the fact that PET is a representation of AED clearer.

**99:** We've added "1km resolution" to this sentence to make it clearer.

**125:** Deleted the "annual means" reference.

**168:** For the lapse rates, we did indeed use the approximate method, by which we calculate

$e\_seaM = (1 - (elevM * 0.025))*e\_M$

where e_M is the MORECS vapour pressure (VP), elevM is the elevation of the MORECS square and e_seaM is the MORECS VP lapsed to sea level.

To lapse the interpolated VP back from sea-level to the grid square elevation we used the fraction of the sea-level value

$e\_1km = ( 1 + (elev1km * 0.025))*e\_sea1km$

where e_sea1km is the interpolated sea level VP, elev1km is the elevation and e_1km is the VP used to calculate our specific humidity.

We have added these equations to the text to clarify.

**183-184:** We have investigated the effect of the 100000Pa assumption on the PET on the same first year of data. We calculated what the VP would have been if we had used the dataset air pressure (p*) instead of the constant 100000Pa air pressure (pc).

You are right that there is a larger relative effect on the PET and PETI where the humidity deficit is small. However, this is generally in the winter and the high ground, where the PET(I) itself is small. So even if the relative difference is high (>~10%), the absolute difference is small, so this is still reasonable.

We haven't quantified the effect this approximation would have on the trends or the attribution, although it is clear from the equations in Appendix C that only the temperature and windspeed contributions would be affected.

**236-238:** This first part of the paper is describing the calculation of a dataset which has already been published. While we agree that for future releases interpolating these data would be an improvement, the existing data set has not been interpolated. We have looked at the output of runs of the JULES model driven by these data and we do not see significant artefacts associated with the CRU grid cell boundaries.

**261:** You are right that the problem is that many different values are mixed together in the highlands. The wind speed is quite variable over small ranges – there are values of ~3 m s-1 and values of ~6 m s-1 very close to each other – so it is very hard to see the different colours. We have tried different limits on the colour bar and there's not much improvement, so we don't really have a solution.

**289-292:** We have added a discussion of this wind speed bias in the description of aerodynamic resistance in the PET calculation. The wind speed bias would lead to a high bias in the PET, but we are calculating a 'reference crop' PET and ignoring the land cover, so this is reasonable.

**313-314, 321, 332:** We have added statements to identify that the spatially varying p* is used everywhere apart from the VP calculation.

**373-374:** Yes, this should have read that the high humidity causes a low humidity deficit, and therefore low evaporative demand. We have fixed this.

**402-405:** Although the equation allows for intercepted water to take more than one day to evaporate, we assume that after the first day the interception is negligible, so that any day without rainfall does not include the interception correction. We have added this to the text.

**End of 455:** We have moved the mention of annual trends in Table 2 to the appropriate point in the paragraph.

**460:** We've amended the discussion of seasonal temperature trends to say that the trend is only significant in spring and autumn.

**476:** We've mentioned the aerodynamic and radiative components in Table 2 in the text. The extra line in Table 2 was actually part of the previous line, but the formatting was such that it was not obvious where one row ended and the next began. We've added some separating lines for clarity.

**479-483:** We've explicitly mentioned Fig B2 where applicable.

**496-497:** We've added the autumn English Lowlands exception

**545:** We have changed "whole dataset" to "seasonal cycle"

**562-563:** We haven't changed the variables from downward LW/SW radiation to net LW/SW radiation, but we have added this caveat to the text.

**563-564:** Yes, the stilling is still significant, so we have changed the text to reflect this.

**568:** We've changed "decreasing" to "more strongly decreasing"

**582:** We've changed "negligible" to say that the wind speed has not had a dominant effect.

**674:** This discussion has now changed somewhat, so we no longer have this sentence.

**729-730:** We have mentioned the overestimate of SWdown in the text (note that we had the legend text wrong – the black lines are the new dataset, the blue lines are the observations – so that the dataset SWdown is an overestimate compared to the observations.)

**748-749 vs rest of paper:** We've changed to Tsp (and also psp) throughout.

**Beginning of 749:** Yes, just a typo, which has been fixed.

**Table 1, Specific Humidity line:** Yes, also a typo which has been fixed.

**Fig A2 caption, lines 1287-1288:** Yes, the caption has been fixed.

**Fig B1 (and probably B2 as well):** We have added annual mean to the caption.

**Also Fig B1:** Yes, we've amended the caption.

**Typos:**

**Beginning of 593:** "or" should be "of": fixed

**1292:** "Ration" should be "ratio": fixed

[revised manuscript text omitted]

---

## Author Response (AR3)

Dear Dr Schymanski,

Thank you again for your comments. We have addressed them and those of the reviewer, particularly with reference to the interdependence of air temperature and LW radiation and the effect this will have on the calculation of PET. We have tabulated the response to your comments below.

We decided against reformulating the second attribution using net LW (and net SW) and instead have expanded the discussion of the validity of the use of air temperature to approximate surface temperature. This is largely because calculation of net radiation requires information about the surface (albedo and emissivity), which are parameters that are chosen when defining the surface for which the PET is calculated. We are rather interested in how the particular chosen PET is dependent on purely meteorological variables. We also think that the discussion of the use of air temperature to approximate surface temperature reads better when considering the upwelling LW as a function of temperature, rather than as a driving variable.

Regards,

Emma Robinson on behalf of the authors

| Editor comment | Author response |
|---|---|
| I am not sure I would agree with the referee's strong opinion about relative humidity being constant as the climate warms, as this may only apply over the oceans (see the wording in Schneider et al., 2010). However, I do agree that it is difficult to separate the effects of air temperature and humidity and therefore I wonder if it might be beneficial for the reader to see attribution based on assuming constant absolute and relative humidity as well as constant vapour pressure deficit, accompanied by a short discussion of the interdependency between air temperature and humidity. | After reflection we decided not to add any extra attributions to the paper, but we have expanded the discussion of the interdependency of air temperature and humidity. As part of this we have added discussion of expected trends in Rh (that it's expected not to change much overall, but that this may vary regionally over land) and references as requested by the reviewer. (Section 4.4) |
| I would like to re-emphasise the reviewer's point about inconsistent consideration of air temperature effects in the radiative component. On L347-350, you explain that the available energy (A) is computed as absorbed shortwave plus absorbed downwelling longwave minus emitted longwave, assuming that surface temperature is equal to air temperature. Obviously, replacement of surface temperature by air temperature is a gross simplification and it needs to be pointed out clearly that this may lead to artefacts. For example, on L591, you explain that increasing air temperature "decreases the radiative component (due to | The assumption that the upwelling LW can be calculated with air temperature as an approximation of surface temperature is indeed a strong simplification and may lead to artefacts. When the air temperature is greater than (less than) the surface temperature, the outgoing LW is over (under) estimated, therefore the net radiation is under (over) estimated, and so the radiative component of the PET is under (over) estimated.

In the absence of information about the surface temperature, a more thorough treatment is to linearise the net radiation as well as the latent and sensible heat fluxes, resulting in an adjusted |

| Editor comment | Author response |
|---|---|
| increasing outgoing LW radiation)", whereas a few lines later you clarify that "downward LW radiation is also proportional to the air temperature so that increases in downward LW broadly cancel the increasing outgoing LW radiation". My impression is that the net effect of air temperature on the radiative component in your analysis is likely an artefact of the simplifications inherent in your estimation of incoming and outgoing longwave. Although it may be justifiable, in the context of previous studies, to present the results as you did, I would urge you to mention this problem prominently in the description of the results (e.g. L633) and in the discussion. See also the referee's comment about L562-563, and his proposed way of considering the problem in hess-520-referee-report-1.pdf, attached to his review. I quite like the idea of first presenting results based on common methods and then present a new set of results based on an improved method, accompanied by a critical discussion of the approach. I leave the choice up to you, but the paper definitely needs an adequate discussion of the deficiencies in the longwave calculations due to lacking knowledge about surface temperature. | Penman-Monteith equation which implicitly solves the surface energy balance by including the net radiation (Monteith, 1981; Thompson et al., 1981).This effectively has an extra resistance, $r_R$, which represents the resistance to LW radiative transfer between the surface and the measurement height, and the net radiation is still calculated using air temperature to find an "isothermal flux density" – the net radiation which would be received if surface temperature were equal to air temperature(Monteith, 1981). As we are using the air temperature to calculate upwelling LW in this study, ideally this adjusted PM equation should have been used and for future studies we will look into implementing it fully. However, for the present study, the difference between the two is relatively small. For typical UK conditions we find that it makes a difference of a few percent to the calculated PET, see the figure below for an example, which shows the PM equation as we have calculated it in the paper (black) and the adjusted PM equation (orange) for typical UK conditions and for daily mean air temperature between 0C and 20C. We find a slightly stronger dependence of the adjusted PET on air temperature, so we may be underestimating the effect of temperature change on PET, although this is likely to be small.
[Figure]
 We have added a brief discussion of this to the text (Section 3 after Eq 8, Section 4.3). In addition, we have added further discussion of the effect of treating the LWdown as an independent variable and LWup as a function of temperature, as recommended by the reviewer. (Section 4.3) |

| Editor comment | Author response |
| --- | --- |
| I am also a bit puzzled about the derivatives in Appendix C, as mentioned below. Could you please check that they are correct (especially wind speed) and explain in the paper how they were obtained? If they are not verifiably correct, this could shed doubt on the whole attribution section. | I have written a summary which gives more details of the steps in the calculation of these derivatives. This can be found after the response to the reviewer below. As part of our study we also verified them by comparing with numerical approximations of the derivatives and are satisfied that they are correct. |

| Additional editor comments | Author response |
| --- | --- |
| L323: double word: "that that" | This has been fixed |
| L 589-592: I would firstly remove the comment that increased air temperature increases the aerodynamic component of PET "as it makes the air more able to hold water", and secondly clarify that the decrease of the radiative component is due to the assumption that surface temperature equals air temperature, which is a gross simplification. The effect of air temperature on PET (or ET) is through the surface energy balance, by its effect on sensible heat flux, as explained in Schymanski & Or (2016, PCE 39(7): 1448-1459). Given the simplifications about the surface energy balance in this study, I would be careful not to over-interpret the results. It is probably too late now to suggest removing the distinction between radiative and aerodynamical components, but I really think that the reader would benefit from a critical discussion of this distinction given the simplifications used here and elsewhere in the literature. | Yes, we have removed the comment that increased air temperature increases the aerodynamic component of PET "as it makes the air more able to hold water".

 As mentioned above, ideally when using air temperature as an input to calculating net radiation, the adjusted Penman-Monteith PET (Monteith, 1981) would be used. In this adjusted formulation, the net radiation is included as part of solving the surface energy balance. Within this framework, the surface temperature is implicitly proportional to air temperature, plus a correction which is a function of air temperature plus other atmospheric variables.

 The current paper assumes that the difference between surface and air temperature is negligible, which is, as you point out, a gross assumption. Relaxing this would incur a different dependence of PET on air temperature, through the extra resistance term $r_R$. The adjusted PM equation is implicitly assuming a relationship between air temperature and surface temperature, so the radiative component is still a function of air temperature.

 For typical conditions in the UK, the difference between the two variants of PET is small. However, we do note that the assumption that surface and air temperature are equal is very strong and may lead to artefacts in this analysis, so we have added text to point this out. (Section 4.3) |

| Additional editor comments | Author response |
|---|---|
| L471: You mentioned that you would adopt the wording proposed by G.P. Weedon here, but have not done so. Was this forgotten? | The discussion at L471 in the current draft ("…while in drier regions (England, English lowlands) the mean PET and PETI are higher than the precipitation for much of the summer…") was not mentioned by GP Weedon. However, assuming that this comment is referring to the discussion of how the confidence intervals were calculated (at L477-481) then G P Weedon suggested that the text be changed to "the 95% confidence intervals of the slope were calculated specifically allowing for the non-zero lag-1 autocorrelation…", which has already been implemented in the manuscript submitted in response to the initial reviews (8th July 2016). (Section 4.1) Please let us know if there is something else we should change. |
| L614: "... to consider relative humidity (R_h) as the independent variable." I propose to add "as" so that the sentence does not imply that RH is indeed an independent variable. | Yes, we have changed this. (Section 4.4) |
| L628: The reference to Jenkins and Dai is probably misplaced here, as the reviewer pointed out. | We had put in a comparison with the relative humidity trends in Jenkins and Dai here, but moved it to the discussion without deleting the references. These have now been deleted. |
| L700: Echoing comment by G.P. Weedon: Please mention the magnitude (and confidence interval) of the trends seen in this data set. | I have added the trend in the MORECS sunshine hours to the previous paragraph. (Section 5) |
| L720: As the referee pointed out, something went wrong here with the references. Please fix this. | Yes, we have deleted these |
| L833: Eq. C1 appears wrong to me, as r_a is a function of the square root of wind speed, so the derivative is not complete. Could you also describe the steps that led to the derivatives in C2-C7? I wasn't able to follow the derivations. | In Eq 10 we define aerodynamic resistance following Allen et al. (1998), so that it is a function of 1/wind speed. This is applicable for ET from vegetated surfaces in neutral conditions, and we use the specific parameters for the reference crop. The derivative is correct for this aerodynamic resistance formulation. As mentioned above, I have attached a summary of the calculations to illustrate this. |

**References**

Allen, R. G., Pereira, L. S., Raes, D., and Smith, M.: Crop evapotranspiration - Guidelines for computing crop water requirements, Food and Agriculture Organization of the United Nations, Rome, Italy, FAO Irrigation and Drainage Paper, 1998.
Monteith, J. L.: Evaporation and surface temperature, Quarterly Journal of the Royal Meteorological Society, 107, 1-27, 10.1002/qj.49710745102, 1981.

Thompson, N., Barrie, I. A., and Ayles, M.: The Meteorological Office rainfall and evaporation calculation system: MORECS, Meteorological Office, Bracknell, 1981.

Dear reviewer #2

Thank you once again for your invaluable input and suggestions for the paper. Although an interesting suggestion, we have decided not to re-implement the attribution with net LW but rather made the suggested changes to the text. We have tabulated the details of our response below.

Regards,

Emma Robinson on behalf of the authors

| Reviewer comment | Author response |
|---|---|
| This revision soundly implements my major RH vs. qa suggestion from last round and largely fixes the minor issues as well. The results are quite interesting in the new framework, and correspond rather well to the recent Spain results of Vicente-Serrano (this should perhaps be noted.) I recommend acceptance pending the following minor revisions. Almost all of these are just writing suggestions or typos stemming from the new revisions, and should be quick to address. | Thank you for the Rh suggestion, we believe that it has been a valuable addition to the paper. Ultimately we decided against using LWnet (and SWnet) in the second attribution section, and focus just on the relative humidity. However, we have amended the text in places to more fully discuss this interdependence of LW radiation and air temperature. We also have implemented your suggestions for text revisions and typo fixes. |
| | We have added a reference to the V-S paper. (Section 5) |
| Since several of the writing suggestions toward the end come from the decision to keep the temperature dependence of longwave radiation artificially split between the temperature and Ld contributions, I also remind the authors that they could still use net longwave if they choose... I still think this would simplify their story a lot. Again this would only be for attribution purposes, not for the main part of the paper. Eq C2 would be exactly the same, but it would now get multiplied by the trend in net longwave rather than the trend in Ld (so I guess it would be named dEp/dLn not dEp/Ld.) And Eqs C5,C7 would just be missing the term involving -4sigmaT3. Other than that, exact same procedure. This would take a bit longer than just changing the writing, but could still be considered.
Thanks again for this huge effort and excellent study! | We have decided to keep the downwelling LW as the variable in the second attribution. Therefore we have extended the discussion of the split between the LWup and LWdown components – details are in the responses below. |

| Minor comments | Author response |
| --- | --- |
| 243-244: Should change the text here to clarify that it makes a negligible difference in the calculated *PET and PETI*, not just the specific humidity. You explained this to me in your written response, but did not change the text to reassure the reader. | We have edited the text to include this. (End of Section 2.2) |
| (Why are there no lines numbered 339 through 369 - is a page missing?) | We have checked and there is no content missing. This must have been an error in the line numbering introduced when producing the marked up version in Word. |
| 558-559 vs. 576-577: This should still be re-arranged for the reader's sake. Right now the reader sees "t is the time in days since a rain event" at 558-559 and is very confused because they just read that this whole thing is only done *on* rain days. So you should say the equivalent of 576-577 right away afterward, and further explain that t is always a fractional value less than 1 (i.e. number of hours since rain divided by 24) rather than a proper number of days (e.g. 3.7 or 6 or 10.2), which is how it reads right now. That has to be made clear for the procedure to make sense. | Yes, we have amended the text to clarify this. (Section 3.1) |
| 765-768 or 758-759 (optional): It may be good here to stick in a reference to the Held and Soden (2006) or Schneider et al (2010) papers I cited in the previous comments, which both argue on theoretical grounds that RH shouldn't be affected as much as qa by climate change. Schneider et al have a particularly nice and transparent derivation, though I suppose it's only valid over ocean. | We have added these references (Section 4.4) |
| 773: Fig B3 is not actually taken from Jenkins or Dai, so what sentence(s) of yours are citing these two papers? Is there a missing intended sentence or two before this citation? (should the citation even be in this location?) | This is a mistake – the text which required these references has been moved to the Discussion but the references were left behind here. This has now been rectified. |
| 779: You mean "aerodynamic component to increase" - need to fix this typo. (Ta bars in Fig 15c are all positive, as they should be!) To show the contrast with the qa framework, you should also add something like ", but much less than before" after the parenthetical remark at 780. | Yes, we have edited the text as requested (End of Section 4.4) |

| Minor comments | Author response |
|---|---|
| Section 4.4 in general: I think you need to explain the method just a bit more… you need to explicitly say here that you computed Rh and Ta derivatives of your new eq. 19 and multiplied them by the Rh and Ta trends, analogous to the beginning of 4.3. Otherwise the reader might be a little confused, particularly as to how the Ta piece of EPA change here could be different in magnitude from the Ta piece already presented. (I totally get it of course, but just making sure a third-party reader does!) An effective way to do this might be to write another simple equation like eq 18 but with dEp/dRh instead of dEp/dqa, and also explicitly noting that dEp/dTa is now taken at constant Rh rather than constant qa. In the last (results) paragraph you could also explain that the Ta piece of EPA is now smaller because it now implicitly includes a constant-Rh rise in qa (this is connected with the previous comment about 780). You don't have to do all of these steps but even doing some of them would help. | These are all useful suggestions and we have implemented these. (Section 4.4) |

| Minor comments | Author response |
|---|---|
| Fig B3b: A bit odd that the local Rh trend is significant over almost all of GB, but the trend in the area-average back at Fig 14c (leftmost symbol) is not significant. Are these significances defined consistently? | Thank you for noticing this – the maps were in fact plotted using the wrong CIs. As part of the process of analysis we calculated CIs based on a simple linear regression, as well as the CIs allowing for the non-zero lag-1 autocorrelation. The latter are the ones that have been used throughout the analysis, but we mistakenly used the former to calculate the significant regions of the trend maps (Figures B1, B2 and B3b). The simple linear regression CIs are less conservative and more restrictive, so make it appear that more of the area of the UK has significant trends. This was the case for all the variables, although most noticeable for the relative humidity. Now that we have remade the plots using the correct CIs, they are consistent with the rest of the analysis, and show smaller regions for which the trends are significant. For the relative humidity this leaves only a few small regions with significant trends, and this is consistent with the area means having non-significant trends as well. Note that the trends shown in the maps are unchanged, just the regions that are grey (for non-significant trends) have increased. (Figures B1, B2 and B3b) |
| 810 (and perhaps 777 and/or 951 as well?) It would be good to remind the reader that the Ta piece only looks so "negligible" because the modest positive part from EPA gets canceled by that funny negative contribution from the EPR term, which as you now explain at 736-742 is kind of "unfair" anyway. And that, thus, the real physical contribution of Ta in the Rh framework is probably not so negligible, and is roughly equal to the EPA part. But agreed, even this is still way smaller than the qa-based result!
A clear way to see this using the Great Britain row of Fig 15 is to assume the Ta bar in the middle column is fictitious, and mentally add its negative to the near-zero Ta bar in the left column. The result is now as-big or bigger than the Sd bar in the left column! (The Ld bars would also be assumed to be fictitious in this estimate, so the total would still be the same.) | Yes, we have added this. (Section 5, also Section 6). |

| Minor comments | Author response |
|---|---|
| 908-914: Similarly, if LW didn't unfairly include the temperature-driven downward but not upward part, these conclusions would also be quite different (the numbers would be way smaller.) | Again, we have added text to point this out. (Section 5) |
| And similar to the 773 comment, what's up with the "ghost" references at the very beginning of the paragraph? | Again, this seems to be to do with moving chunks of text around – the references get copied to the new location, but also abandoned in the original place. They have now been deleted from this paragraph. |
| Table 1, specific humidity row: The constant air pressure is 100 kPa, not 1 kPa (right?) | Yes, this has been changed |
| Table 5 caption: needs to include a "when relative humidity is used" clause, like Table 6 caption. (Unless I misunderstand what Table 5 actually is??) | Yes, this has been changed |
| Also, Table 5 still has a stray "Specific Humidity" heading in part c… this should be changed to "Relative Humidity" unless I misunderstand. (Are the numbers in Table 5c still Rh numbers, or are they accidentally qa numbers?) | Yes, this was an oversight in the column heading and has been changed. The numbers are the Rh numbers, not qa, so have not been altered. |

**Summary of derivatives calculated for Appendix C**

**Wind speed**

The aerodynamic resistance is inversely proportional to the wind speed (not a function of the square root). If we substitute the aerodynamic resistance (Eq 10) into the PET (Eq 4), then we have

$$E_P = \frac{t_d}{\lambda} \frac{\Delta A + \frac{c_p \rho_a u_{10}}{278}(q_s - q_a)}{\Delta + \gamma \left(1 + \frac{r_s u_{10}}{278}\right)} \tag{S1}$$

We differentiate this (using the quotient rule) to get

$$\frac{\partial E_P}{\partial u_{10}} = \frac{t_d}{\lambda} \left( \frac{\frac{c_p \rho_a}{278}(q_s - q_a)}{\Delta + \gamma \left(1 + \frac{r_s}{r_a}\right)} - \frac{\frac{\gamma r_s}{278}\left(\Delta A + \frac{c_p \rho_a u_{10}}{278}(q_s - q_a)\right)}{\left(\Delta + \gamma \left(1 + \frac{r_s}{r_a}\right)\right)^2} \right) \tag{S2}$$

We can then rearrange this to get

$$\frac{\partial E_P}{\partial u_{10}} = \frac{\frac{t_d}{\lambda} \frac{\frac{c_p \rho_a}{r_a}(q_s - q_a)}{\Delta + \gamma \left(1 + \frac{r_s}{r_a}\right)}(\Delta + \gamma) - \frac{t_d}{\lambda} \frac{\Delta A}{\Delta + \gamma \left(1 + \frac{r_s}{r_a}\right)} \frac{\gamma r_s}{r_a}}{u_{10}\left(\Delta + \gamma \left(1 + \frac{r_s}{r_a}\right)\right)} \tag{S3}$$

And notice that the numerator can be simplified by replacing $E_{PA}$ and $E_{PR}$ to get

$$\frac{\partial E_P}{\partial u_{10}} = \frac{(\Delta + \gamma)E_{PA} - \gamma \frac{r_s}{r_a} E_{PR}}{u_{10}\left(\Delta + \gamma \left(1 + \frac{r_s}{r_a}\right)\right)} \tag{S4}$$

**Downward shortwave radiation**

Using the equation for available energy (Eq 8), the PET is

$$E_P = \frac{t_d}{\lambda} \frac{\Delta \left((1 - \alpha)S_d + \varepsilon(L_d - \sigma T_*^4)\right) + \frac{c_p \rho_a}{r_a}(q_s - q_a)}{\Delta + \gamma \left(1 + \frac{r_s}{r_a}\right)} \tag{S5}$$

Differentiating this with respect to downward shortwave gives

$$\frac{\partial E_P}{\partial S_d} = \frac{t_d}{\lambda} \frac{\Delta(1 - \alpha)}{\Delta + \gamma \left(1 + \frac{r_s}{r_a}\right)} \tag{S6}$$

Which is equal to

$$\frac{\partial E_P}{\partial S_d} = \frac{t_d}{\lambda} \frac{\Delta A}{\Delta + \gamma \left(1 + \frac{r_s}{r_a}\right)} \frac{(1 - \alpha)}{A} \tag{S7}$$

Again, we can simplify by replacing $E_{PR}$ to get

$$\frac{\partial E_P}{\partial S_d} = E_{PR} \frac{(1 - \alpha)}{A} \tag{S8}$$

Note that in the paper we were using $R_n$ rather than $A$, but have changed this to make the notation consistent with the rest of the paper.

In practice, when coding we would tend to use the expanded version (Eq S6), rather than this final version (Eq S8), as having available energy in the denominator is badly behaved as $A$ becomes small.

**Downward longwave radiation**

Similarly to the shortwave, we use the available energy equation and then differentiate Eq S5 with respect to the downward longwave to get

$$\frac{\partial E_P}{\partial L_d} = \frac{t_d}{\lambda} \frac{\Delta \varepsilon}{\Delta + \gamma \left(1 + \frac{r_s}{r_a}\right)} \tag{S9}$$

Which rearranges to

$$\frac{\partial E_P}{\partial L_d} = \frac{t_d}{\lambda} \frac{\Delta A}{\Delta + \gamma \left(1 + \frac{r_s}{r_a}\right)} \frac{\varepsilon}{A} \tag{S10}$$

Which is equivalent to

$$\frac{\partial E_P}{\partial L_d} = E_{PR} \frac{\varepsilon}{A} \tag{S11}$$

**Specific humidity**

Differentiating the PET (Eq 4) with respect to specific humidity gives

$$\frac{\partial E_P}{\partial q_a} = \frac{t_d}{\lambda} \frac{-\frac{c_p \rho_a}{r_a}}{\Delta + \gamma \left(1 + \frac{r_s}{r_a}\right)} \tag{S12}$$

Which is equivalent to

$$\frac{\partial E_P}{\partial q_a} = \frac{t_d}{\lambda} \frac{-1}{(q_s - q_a)} \frac{\frac{c_p \rho_a}{r_a}(q_s - q_a)}{\Delta + \gamma \left(1 + \frac{r_s}{r_a}\right)} \tag{S13}$$

Which simplifies to

$$\frac{\partial E_P}{\partial q_a} = \frac{E_{PA}}{q_a - q_s} \tag{S14}$$

**Air temperature**

This is more complicated, as many of the variables in the standard PET equation are functions of air temperature, namely specific humidity and its derivative, available energy and air density. Using the quotient rule, we can calculate the derivative of the PET to be

$$\frac{\partial E_P}{\partial T_a} = \frac{t_d}{\lambda}\left(\frac{\Delta \frac{\partial A}{\partial T_a} + A \frac{\partial \Delta}{\partial T_a} + \frac{c_p}{r_a}\left(\rho_a \frac{\partial q_s}{\partial T_a} + (q_s - q_a)\frac{\partial \rho_a}{\partial T_a}\right)}{\Delta + \gamma\left(1 + \frac{r_s}{r_a}\right)}\right.$$
$$\left. - \frac{\left(\Delta A + \frac{c_p \rho_a}{r_a}(q_s - q_a)\right)\frac{\partial \Delta}{\partial T_a}}{\left(\Delta + \gamma\left(1 + \frac{r_s}{r_a}\right)\right)^2}\right)$$

(S15)

We then substitute the derivatives

$$\frac{\partial A}{\partial T_a} = -4\,\varepsilon\,\sigma T_a^3$$

(S16)

$$\frac{\partial \rho_a}{\partial T_a} = \frac{-p_*}{rT_a^2} = \frac{-\rho_a}{T_a}$$

(S17)

$$\frac{\partial q_s}{\partial T_a} = \Delta$$

(S18)

$$\frac{\partial \Delta}{\partial T_a} = \Delta \left(\frac{T_{sp}}{T_a^2}\frac{\sum_{i=1}^4 i(i-1)a_i\left(1-\frac{T_{sp}}{T_a}\right)^{i-2}}{\sum_{i=1}^4 ia_i\left(1-\frac{T_{sp}}{T_a}\right)^{i-1}} + \Delta\frac{p_* + (1-\varepsilon)e_s}{p_*q_s} - \frac{2}{T_a}\right)$$

(S19)

And rearrange, substituting $E_{PA}$ and $E_{PR}$ as necessary, to get

$$\frac{\partial E_P}{\partial T_a} = E_{PR}\left[\left(1 - \frac{\Delta}{\Delta+\gamma\left(1+\frac{r_s}{r_a}\right)}\right)\left(\frac{T_{sp}}{T_a^2}\frac{\sum_{i=1}^4 i(i-1)a_i\left(1-\frac{T_{sp}}{T_a}\right)^{i-2}}{\sum_{i=1}^4 ia_i\left(1-\frac{T_{sp}}{T_a}\right)^{i-1}} + \Delta\frac{p_*+(1-\varepsilon)e_s}{p_*q_s} - \frac{2}{T_a}\right) - \right.$$
$$\frac{4\varepsilon\sigma T_a^3}{R_n}\Bigg] + \qquad E_{PA}\left[\frac{\Delta}{q_s-q_a} - \frac{1}{T_a} - \frac{\Delta}{\Delta+\gamma\left(1+\frac{r_s}{r_a}\right)}\left(\frac{T_{sp}}{T_a^2}\frac{\sum_{i=1}^4 i(i-1)a_i\left(1-\frac{T_{sp}}{T_a}\right)^{i-2}}{\sum_{i=1}^4 ia_i\left(1-\frac{T_{sp}}{T_a}\right)^{i-1}} + \right.\right.$$
$$\left.\left.\Delta\frac{p_*+(1-\varepsilon)e_s}{p_*q_s} - \frac{2}{T_a}\right)\right]$$

(S20)

**Relative humidity**

When we look at the problem in terms of relative humidity instead of specific humidity, the PET is

$$E_P = \frac{t_d}{\lambda}\frac{\Delta A + \frac{c_p \rho_a}{r_a}q_s(1 - R_h)}{\Delta + \gamma\left(1 + \frac{r_s}{r_a}\right)}$$

(S21)

In this case, the derivative with respect to relative humidity is reasonably straightforward, giving

$$\frac{\partial E_P}{\partial R_h} = \frac{t_d}{\lambda}\frac{-\frac{c_p \rho_a}{r_a}q_s}{\Delta + \gamma\left(1 + \frac{r_s}{r_a}\right)}$$

(S22)

Which is equivalent to

$$\frac{\partial E_P}{\partial R_h} = \frac{t_d}{\lambda} \frac{-1}{1-R_h} \frac{\frac{c_p \rho_a}{r_a} q_s (1-R_h)}{\Delta + \gamma \left(1 + \frac{r_s}{r_a}\right)} \tag{S23}$$

Which can be rewritten

$$\frac{\partial E_P}{\partial R_h} = \frac{E_{PA}}{R_h - 1} \tag{S24}$$

**Air temperature when using relative humidity**

In this case, the derivative of PET with respect to air temperature is

$$\frac{\partial E_P}{\partial T_a} = \frac{t_d}{\lambda} \left( \frac{\Delta \frac{\partial A}{\partial T_a} + A \frac{\partial \Delta}{\partial T_a} + \frac{c_p}{r_a}(1-R_h)\left(\rho_a \frac{\partial q_s}{\partial T_a} + q_s \frac{\partial \rho_a}{\partial T_a}\right)}{\Delta + \gamma \left(1 + \frac{r_s}{r_a}\right)} \right.$$
$$\left. - \frac{\left(\Delta A + \frac{c_p \rho_a}{r_a} q_s(1-R_h)\right) \frac{\partial \Delta}{\partial T_a}}{\left(\Delta + \gamma \left(1 + \frac{r_s}{r_a}\right)\right)^2} \right) \tag{S25}$$

We use the same substitutions as before for the derivatives of specific humidity (Eq S18) and its derivative (Eq S19), available energy (Eq S16) and air density (Eq S17), and rearrange, substituting $E_{PA}$ and $E_{PR}$ as necessary, to get

[revised manuscript text omitted]

---

## Author Response (AR4)

Dear Dr Schymanski,

Thank you for your quick response. We haven't made any changes, as detailed below, but the system wouldn't allow uploading the author's response without uploading a manuscript, so the newly uploaded version of the manuscript is the same as the one uploaded on 31$^{st}$ January.

Regarding Fig B3b, this is indeed now correct. It was changed because a comment from the reviewer (attached in the table below) revealed an error, and so we have updated the figure accordingly. It is now consistent with the area averages which have non-significant trends. We also updated the trend maps in Figures B1 and B2, although the differences are much less between the old and new versions.

Regarding the assets (and the citation of the datasets in the paper). This is the result of having recently extended the data sets to have coverage up to 2015. Since, as you note, the DOI refers to a static dataset, once we had the extra data we had to publish the full 1961-2015 datasets with new DOIs. Unfortunately, the procedures used by the data centre are such that the original data download links are superseded. However, the original 1961-2012 files are unchanged and are still available through the download links associated with the new DOIs. We appreciate this is not ideal but for now we are constrained by the procedures imposed by the data centre.

Because the analysis was performed with the 1961-2012 data, and since the update to 2015 happened very recently, we kept the original DOIs in the references and in the assets. We think this is most explicit about exactly which data were used, while also allowing for readers to follow through to download the files from the newest version of the data.

Finally, we would still be interested in giving Reviewer #2 a named acknowledgement, if they would be happy for us to know who they are.

Regards,

Emma Robinson on behalf of the authors

| Reviewer comment | Author response |
|---|---|
| Fig B3b: A bit odd that the local Rh trend is significant over almost all of GB, but the trend in the area-average back at Fig 14c (leftmost symbol) is not significant. Are these significances defined consistently? | Thank you for noticing this – the maps were in fact plotted using the wrong CIs. As part of the process of analysis we calculated CIs based on a simple linear regression, as well as the CIs allowing for the non-zero lag-1 autocorrelation. The latter are the ones that have been used throughout the analysis, but we mistakenly used the former to calculate the significant regions of the trend maps (Figures B1, B2 and B3b). The simple linear regression CIs are less conservative and more restrictive, so make it appear that more of the area of the UK has significant trends. This was the case for all the variables, although most noticeable for the relative humidity. Now that we have remade the plots using the correct CIs, they are consistent with the rest of the analysis, and show smaller regions for which the trends are significant. For the relative humidity this leaves only a few small regions with significant trends, and this is consistent with the area means having non-significant trends as well. Note that the trends shown in the maps are unchanged, just the regions that are grey (for non-significant trends) have increased. (Figures B1, B2 and B3b) |

---

## Author Response (AR5)

Dear Stan,

Thank you, this is excellent news! We have edited the acknowledgements to specifically acknowledge the contribution of Jack Scheff to the review process.

Regards,

Emma Robinson on behalf of the authors